# Reimagining Synthetic Tabular Data Generation through Data-Centric AI: A Comprehensive Benchmark

**Lasse Hansen**[*1,2]     **Nabeel Seedat**[*3]     **Mihaela van der Schaar**[3]     **Andrija Petrovic**[4]

[1]Department of Affective Disorders, Aarhus University Hospital - Psychiatry, Aarhus, Denmark
[2]Department of Clinical Medicine, Aarhus University, Denmark
[3]Department of Applied Mathematics and Theoretical Physics, University of Cambridge, UK
[4]Faculty of Organisational Sciences, University of Belgrade, Serbia
`lasse.hansen@clin.au.dk`
`ns741@cam.ac.uk`

## Abstract

Synthetic data serves as an alternative in training machine learning models, particularly when real-world data is limited or inaccessible. However, ensuring that synthetic data mirrors the complex nuances of real-world data is a challenging task. This paper addresses this issue by exploring the potential of integrating data-centric AI techniques which profile the data to guide the synthetic data generation process. Moreover, we shed light on the often ignored consequences of neglecting these data profiles during synthetic data generation — despite seemingly high statistical fidelity. Subsequently, we propose a novel framework to evaluate the integration of data profiles to guide the creation of more representative synthetic data. In an empirical study, we evaluate the performance of five state-of-the-art models for tabular data generation on eleven distinct tabular datasets. The findings offer critical insights into the successes and limitations of current synthetic data generation techniques. Finally, we provide practical recommendations for integrating data-centric insights into the synthetic data generation process, with a specific focus on classification performance, model selection, and feature selection. This study aims to reevaluate conventional approaches to synthetic data generation and promote the application of data-centric AI techniques in improving the quality and effectiveness of synthetic data.

## 1   Introduction

Machine learning has become an essential tool across various industries, with high-quality data representative of the real world being a crucial component for training accurate models that generalize [1, 2, 3]. In cases where data access is restricted or insufficient synthetic data has emerged as a viable alternative [4, 5]. The purpose of synthetic data is to generate training data that closely mirrors real-world data, enabling the effective use of models trained on synthetic data on real data. Moreover, synthetic data is used for a variety of different uses, including privacy (i.e. to enable data sharing, [6, 7]), competitions [8] fairness [9, 10], and improving downstream models [11, 12, 13, 14].

---

[*]Equal Contributions

37th Conference on Neural Information Processing Systems (NeurIPS 2023) Track on Datasets and Benchmarks.

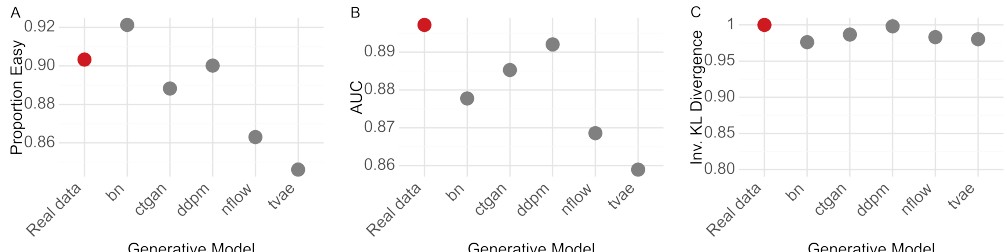

Figure 1: Measures of data-centric profiling (A) better reflect the downstream performance of generative models (B) than measures of statistical fidelity (C). Assessed on the Adult dataset [16] using five different generative models A) Proportion easy examples in the generated datasets identified by Cleanlab, B) Supervised classification performance when training on synthetic, testing on real data, C) Inverse KL-divergence. (bn=`bayesian_network`)

However, generating high-quality synthetic data that adequately captures the nuances of real-world data, remains a challenging task. Despite significant strides in synthetic data with generative models, they sometimes fall short in replicating the complex subtleties of real-world data, particularly when dealing with messy, mislabeled or biased data. For instance, regarding fairness, [15] have shown that such gaps can lead to flawed conclusions and unreliable predictions on subpopulations, thereby restricting the practical usage of synthetic data.

The ability of synthetic data to capture the subtle complexities of real-world data is crucial, particularly in contexts where these issues might surface during deployment. Inaccurate synthetic data can not only hamper predictive performance but also result in improper model selection and distorted assessments of feature importance, thereby undermining the overall analysis. These challenges underscore the need to improve the synthetic data generation process.

One might wonder, surely, assessing fidelity via statistical divergence metrics [17, 5] such as MMD or KL-divergence is sufficient? We argue that such high-level metrics tell one aspect of the story. An overlooked dimension is the characterization of data profiles. In this approach, samples are assigned to profiles that reflect their usefulness for an ML task. Specifically, samples are typically categorized as easy to learn, ambiguous, or hard, which are proxies for data issues like mislabeling, data shift, or under-represented samples. In methods such as Data-IQ [18]and Data Maps [19] this is referred to as "groups of the data", however, we use "data profiles" for clarity.

While this issue has been well-studied for supervised tasks, it has not been explored in the generative setting. We highlight the issues of overlooking such data profiling in Figure 1, where despite near-perfect statistical fidelity (inverse KLD), we show the differing proportion of 'easy' examples identified in synthetic data generated by different generative models trained on the Adult dataset [16]. On the other hand, this data profile correlates with downstream classification performance.

To address this challenge of the representativeness of synthetic data, we explore the potential of integrating data-centric AI techniques and their insights to improve synthetic data generation. Specifically, we propose characterizing individual samples in the data and subsequently using the different data profiles to guide synthetic data generation in a way that better reflects the real world. While our work is applicable across modalities, our primary focus is tabular data given the ubiquity of tabular data in real-world applications [20, 21], with approximately 79% of data scientists working with it on a daily basis, vastly surpassing other modalities [22].

**Contributions:**
① *Conceptually*, we delve into the understanding of fundamental properties of data with respect to synthetic data generation, casting light on the impact of overlooking data characteristics and profiles when generating synthetic data.
② *Technically*, we bring the idea of data profiles in data-centric AI to the generative setting and explore its role in guiding synthetic data generation. We introduce a comprehensive framework to facilitate this evaluation across various generative models.
③ *Empirically*, we benchmark the performance of five state-of-the-art models for tabular data generation on eleven distinct tabular datasets and investigate the practical integration of data-centric profiles to guide synthetic data generation. We provide practical recommendations for enhancing

synthetic data generation, particularly with respect to the 3 categories of synthetic data utility (i) predictive performance, (ii) model selection and (iii) feature selection.

We hope the insights of this paper spur the reconsideration of the conventional approaches to synthetic data generation and encourage experimentation on how data-centric AI could help synthetic data generation deliver on its promises.

## 2    Related work

This work engages with synthetic data generation and data characterization in data-centric AI.

**Synthetic Tabular Data Generation** uses generative models to create artificial data that mimics the structure and statistical properties of real data, and is particularly useful when real data is scarce or inaccessible [23, 24, 4]. In the following, we describe the broad classes of synthetic data generators applicable to the tabular domain. *Bayesian networks* [25] are a traditional approach for synthetic data generation, that represent probabilistic relationships using graphical models. *Conditional Tabular Generative Adversarial Network* (CTGAN) [26] is a deep learning method for modeling tabular data. It uses a conditional GAN to capture complex non-linear relationships. *The Tabular Variational Autoencoder* (TVAE) is a specialized Variational Autoencoder, designed for the tabular setting [26].

*Normalizing flow models* [27, 28] provide an invertible mapping between data and a known distribution, and offer a flexible approach for generative modeling. Diffusion models, which have gained recent popularity, offer a different paradigm for generative modeling. *TabDDPM* [29] is a diffusion model proposed for the tabular data domain. In this work, we evaluate these classes of generative models, considering various aspects of synthetic data evaluation.

**Evaluation of Synthetic Data** is a multifaceted task [17, 30], involving various dimensions such as data utility with respect to a downstream task, statistical fidelity, and privacy preservation [17, 30]. In this work, we focus on dimensions that impact model performance and hence, while important, we do not consider privacy aspects.

*(1) Data Utility:* refers to how well the synthetic data can be used in place of the real data for a given task. Typically, utility is assessed by training predictive models on synthetic data and testing them on real data [4, 17, 31, 32, 5]. We posit that beyond matching predictive performance, we also desire to retain both *model ranking* and *feature importance* rankings. We empirically assess these aspects in Sec. 5.

*(2) Statistical Fidelity:* measures the degree of similarity between synthetic data and the original data in terms of statistical properties, including the marginal and joint distributions of variables [17]. Statistical tests like the Kolmogorov-Smirnov test or divergence measures like Maximum Mean Discrepancy, KL-divergence or Wasserstein distance are commonly used for evaluation[17, 5].

Beyond statistical measures, the concept of data characterization and profiles of easy and hard examples has emerged in data-centric AI. These profiles serve as proxies for understanding real-world data, which is often not "perfect" due to mislabeling, noise, etc.The impact of these profiles on supervised models has been demonstrated in the data-centric literature [33, 34, 18]. In Figure 1, we show that data profiles are similarly important in the generative setting. Despite having almost perfect statistical fidelity, different generative models capture different data profiles (e.g. proportion of easy examples), leading to varying data utility as reflected in different performances. Consequently, we propose considering data profiles as an important dimension when creating synthetic data. We describe current data-centric methods that can facilitate this next.

**Data profiling** is a growing field in Data-Centric AI that aims to evaluate the characteristics of data samples for specific tasks [35, 36]. In the supervised learning setting, various methods have been developed to assign samples to groups, which we refer to as data profiles. These profiles, such as easy, ambiguous, or hard, often reveal issues such as mislabeling, data shifts, or under-represented groups [34, 18, 33, 37, 19, 38]. Various mechanisms are used in different methods for data characterization. For example, Cleanlab [34] models relationships between instances based on confidence, while Data Maps and Data-IQ [18] assess uncertainty through training dynamics. However, many existing methods are designed for neural networks and are unsuitable for non-differentiable models like XGBoost, which are commonly used in tabular data settings. Consequently, we focus on data characterization approaches such as Cleanlab, Data-IQ, and Data Maps which are more applicable to tabular data.

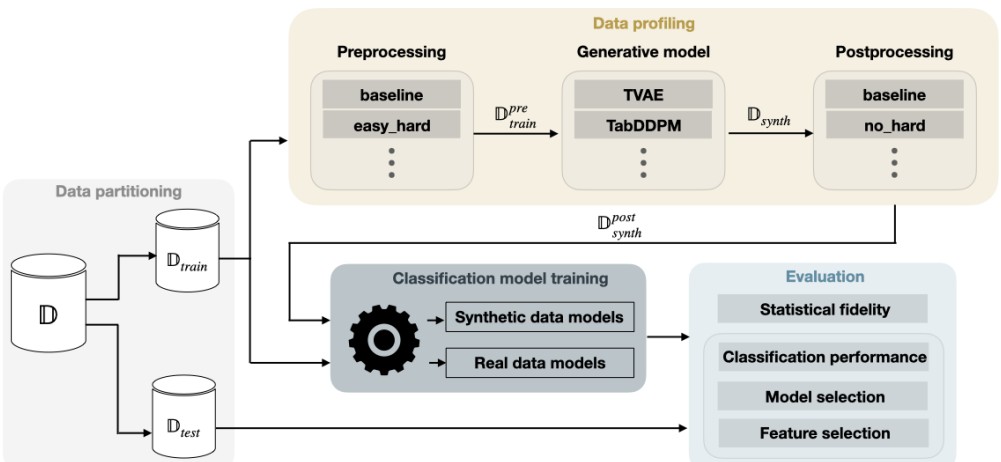

Figure 2: Illustration of the framework's process flow. *Data partitioning*: the dataset is divided into a training set, $\mathbb{D}_{\text{train}}$, and a testing set, $\mathbb{D}_{\text{test}}$. *Data profiling*: a data-centric preprocessing approach is employed on a duplicate of $\mathbb{D}_{\text{train}}$ to produce $\mathbb{D}_{\text{train}}^{\text{pre}}$. A generative model, trained on $\mathbb{D}_{\text{train}}^{\text{pre}}$, is then utilized to synthesize a dataset, $\mathbb{D}_{\text{synth}}$, which is further processed using a data-centric postprocessing method to achieve the final synthetic dataset, $\mathbb{D}_{\text{synth}}^{\text{post}}$. *Classification model training*: various classification models are separately trained on $\mathbb{D}_{\text{train}}$ and $\mathbb{D}_{\text{synth}}^{\text{post}}$ and applied to $\mathbb{D}_{\text{test}}$. *Evaluation*: the generative and supervised models are appraised for their statistical fidelity and utility, focusing on classification accuracy, model selection, and feature selection.

# 3 Framework

We propose a unified framework that enables a thorough assessment of generative models and the synthetic data they produce. The framework encompasses the evaluation of the synthetic data based on established statistical fidelity metrics as well as three distinct tasks encompassing *data utility*.

At a high level, the framework proceeds as visualized in 2. The dataset is first divided into a training set, denoted as $\mathbb{D}_{\text{train}}$, and a testing set, denoted as $\mathbb{D}_{\text{test}}$. A duplicate of the training set ($\mathbb{D}_{\text{train}}$) undergoes a data-centric preprocessing approach to produce a preprocessed version of the training set, referred to as $\mathbb{D}_{\text{train}}^{\text{pre}}$. A generative model is then trained on $\mathbb{D}_{\text{train}}^{\text{pre}}$. This model is used to synthesize a new dataset, denoted as $\mathbb{D}_{\text{synth}}$. The synthetic dataset is further processed using a data-centric postprocessing method to create the final synthetic dataset, denoted as $\mathbb{D}_{\text{synth}}^{\text{post}}$. Various classification models $\mathcal{M}$ are then trained separately on the original training set $\mathbb{D}_{\text{train}}$ and the synthetic dataset $\mathbb{D}_{\text{synth}}^{\text{post}}$. These models are then applied to the testing set $\mathbb{D}_{\text{test}}$ for evaluation. The generative and supervised models are evaluated for their statistical fidelity and data utility. The focus is on classification performance, model selection, and feature selection. Further details on each process within the framework can be found in the following subsections.

## 3.1 Data profiling

Assume we have a dataset $\mathcal{D} = \{(x^n, y^n) \mid n \in [N]\}$. Data profiling aims to assign a score $S$ to samples in $\mathcal{D}$. On the basis of the score, a threshold $\tau$ is typically used to assign a specific profile group $p^n \in \mathcal{P}$, where $\mathcal{P} = \{Easy, Ambigious, Hard\}$ to each sample $x^n$.

Our framework supports three recent data characterization methods applicable to tabular data: Cleanlab [34], Data-IQ [18], and Data Maps [33]. They primarily differ based on their scoring mechanism $S$. For instance, Cleanlab [34] uses the predicted probabilities as $S$ to estimate a noise matrix, Data-IQ [18] uses confidence and aleatoric uncertainty as $S$, and Data Maps uses confidence and variability (epistemic uncertainty) as $S$. Moreover, they differ in the categories in the data profiles derived from their scores. Data-IQ and Data Maps provide three categories of data profiles: *easy*; samples that are easy for the model to predict, *ambiguous*; samples with high uncertainty, and *hard*;

samples that are wrongly predicted with high certainty. Cleanlab provides two profiles: *easy* and *hard* examples.

We create data profiles with these three data-centric methods to evaluate the value of data-centric methods to improve synthetic data generation, both *ex-ante* and *post hoc*. We use the profiles in multiple preprocessing and postprocessing strategies applied to the original and synthetic data.

### 3.1.1 Preprocessing

Preprocessing strategies are applied to the original data $\mathbb{D}_{\text{train}}$ i.e., before feeding to a generative model. We investigate three preprocessing strategies: (1) `baseline`, which applies no processing, and simply feeds the $\mathbb{D}_{\text{train}}$ to the generative model. (2) `easy_hard`: Let $S_c : \mathbb{D}_{\text{train}} \rightarrow [0, 1]$ denote the scoring function for data-centric method $c$. We partition $\mathbb{D}_{\text{train}}$ into $\mathbb{D}_{\text{train}}^{\text{easy}}$ and $\mathbb{D}_{\text{train}}^{\text{hard}}$ data profiles using a threshold $\tau$, such that $\mathbb{D}_{\text{train}}^{\text{easy}} = \{x^n \mid S_c(x^n) \leq \tau\}$ and $\mathbb{D}_{\text{train}}^{\text{hard}} = \{x^n \mid S_c(x^n) > \tau\}$. (3) Analogously, `easy_ambiguous_hard` [2] splits the $\mathbb{D}_{\text{train}}$ on the easy, ambiguous, and hard examples. Further details are provided in Appendix A.

### 3.1.2 Generative model

We utilize the data profiles identified in the preprocessing step to train a specific generative model for each data segment, e.g. easy and hard examples separately. Let $G : \mathbb{D}_{\text{train}} \rightarrow \mathbb{D}_{\text{synth}}$ denote the generative model trained on a dataset $\mathbb{D}_{\text{train}}$, which produces synthetic dataset $\mathbb{D}_{\text{synth}}$. In our framework, for each data profile in preprocessed dataset $\mathbb{D}_{\text{train}}^{\text{pre}}$, we train a separate generative model. We generate data using each generative model and the combined synthetic data is then $\mathbb{D}_{\text{synth}} = G_{\text{easy}}(\mathbb{D}_{\text{train}}^{\text{easy}}) \cup G_{\text{hard}}(\mathbb{D}_{\text{train}}^{\text{hard}})$, with generation preserving the ratio of the data segments, to reflect their distribution in the initial dataset.

### 3.1.3 Postprocessing

We define postprocessing strategies as processing applied to the synthetic data after data generation but before supervised model training and task evaluation. We denote the set of postprocessing strategies as $\mathcal{H}$. Given the synthetic dataset $\mathbb{D}_{\text{synth}}$, each postprocessing strategy $h \in \mathcal{H}$ maps $\mathbb{D}_{\text{synth}}$ to a processed dataset $\mathbb{D}_{\text{synth}}^{\text{post}} = h(\mathbb{D}_{\text{synth}})$. Two different postprocessing strategies were used: `baseline`: This is the identity function $h_{\text{baseline}}(\mathbb{D}_{\text{synth}}) = \mathbb{D}_{\text{synth}}$. `no_hard`: We remove the hard examples from the synthetic data, $\mathbb{D}_{\text{synth}}^{\text{post}} = \mathbb{D}_{\text{synth}} \setminus \{x_{\text{synth}}^n \mid S_c(x_{\text{synth}}^n) > \tau\}$, where $x_{\text{synth}}^n$ is generated synthetic data.

## 3.2 Classification model training

The training procedure of the supervised classification models $\mathcal{M}$ comprises two steps, each minimizing a cost function $\mathcal{L}$. (1) Train on the real data, i.e., $\mathcal{M}_{\text{real}} = \arg\min \mathcal{L}(\mathcal{M}(\mathbb{D}_{\text{train}}))$. (2) Train on synthetic data, i.e. $\mathcal{M}_{\text{syn}} = \arg\min \mathcal{L}(\mathcal{M}(\mathbb{D}_{\text{synth}}^{\text{post}}))$ We then compare utility of $\mathcal{M}_{\text{real}}$ and $\mathcal{M}_{\text{syn}}$ in the evaluation procedure. Our framework supports any machine learning model $\mathcal{M}$ compatible with the Scikit-Learn API.

## 3.3 Evaluation

Finally, the framework includes automated evaluation tools for the generated synthetic data to evaluate the effect of pre- and postprocessing strategies, across datasets, random seeds, and generative models. To thoroughly assess our framework, we establish evaluation metrics that extend beyond statistical fidelity, encapsulating data utility through the inclusion of three tasks.

### 3.3.1 Statistical fidelity

The quality of synthetic data is commonly assessed using divergence measures between the real and synthetic data [5, 30]. Our framework allows for this assessment using widely adopted methods including inverse KL-Divergence [5], Maximum Mean Discrepancy [39], Wasserstein distance, as

---

[2] Only defined for data-centric methods that identify ambiguous examples, i.e. Data-IQ and Data Maps.

well as Alpha-precision and Beta-Recall [30]. However, as shown in Figure 1, such measures can only tell one aspect of the story. Indeed, despite all generative models providing near-perfect statistical fidelity based on divergence measures, the synthetic data captures the nuances of real data differently, as reflected in the varying data profiles (e.g. proportion easy examples). This motivates us to also assess the data utility and the potential implications of this variability.

### 3.3.2 Data utility

Three specific metrics were employed to assess data utility: classification performance, model selection, and feature selection.

**Classification performance** To explore the usefulness of the generated synthetic data for model training, we use the train-on-synthetic, test-on-real paradigm to fit a set of machine learning models $\mathcal{M}$ on the synthetic data, $\mathbb{D}_{\text{synth}}$, and subsequently evaluate their performance on a real, held-out test dataset, $\mathbb{D}_{\text{test}}$. By using $\mathbb{D}_{\text{test}}$ we avoid potential issues from data leakage that might occur from an evaluation on the real training sets, $\mathbb{D}_{\text{train}}$.

**Model selection** When using synthetic data for model selection, it is imperative that the ranking of classification models $\mathcal{M}$ trained on synthetic data aligns closely with the ranking of classification models trained on the original data. To evaluate this, we first train a set of $\mathcal{M}_{\text{real}}$ on $\mathbb{D}_{\text{train}}$ and evaluate their classification performance on $\mathbb{D}_{\text{test}}$. Next, we fit the same set of $\mathcal{M}_{\text{synth}}$ on $\mathbb{D}_{\text{synth}}^{\text{post}}$ and evaluate their classification performance on $\mathbb{D}_{\text{test}}$. The rank-ordering of the $\mathcal{M}_{\text{real}}$ in terms of a performance metric (e.g. AUROC) is compared with the ranking-order of the $\mathcal{M}_{\text{synth}}$ using Spearman's Rank Correlation.

**Feature selection** Feature selection is a crucial task in data analysis and machine learning, aiming to identify the most relevant and informative features that contribute to a model's predictive power. To evaluate the utility of using synthetic data for feature selection, a similar approach is followed as for model selection. First, a model $\mathcal{M}_{\text{real}}$ with inherent feature importance (e.g. random forest) is trained on $\mathbb{D}_{train}$ and the rank-ordering of the most important features is determined. This ranking is then compared to the rank ordering of the most important features obtained from the same model type $\mathcal{M}_{\text{synth}}$ trained on $\mathbb{D}_{\text{synth}}^{\text{post}}$ using Spearman's Rank Correlation.

### 3.4 Extending the framework

The framework presented in this paper is intentionally designed to be modular and highly adaptable, allowing for seamless integration of various generative models, pre- and postprocessing strategies, and diverse tasks. This flexibility enables researchers and practitioners to explore and evaluate e.g. different combinations of generative models alongside various pre- and post-processing strategies. Further, the framework is extensible, allowing for the incorporation of additional generative models, novel processing methods, and emerging tasks, ensuring that it remains up-to-date and capable of accommodating future advancements in the field of synthetic data generation.

## 4 Experiments

To demonstrate the framework, we conduct multiple experiments, aiming to answer the following subquestions in order to investigate: **Can data-centric ML improve synthetic data generation?**:

**Q1:** Is statistical fidelity sufficient to quantify the utility of synthetic data?

**Q2:** Can we trust results from supervised classification models trained on synthetic data to generalize to real data?

**Q3**: Can data-centric approaches be integrated with synthetic data generation to create more realistic synthetic data?

**Q4**: Does the level of label noise influence the effect of data-centric processing for synthetic data generation?

All code for running the analysis and creating tables and graphs can be found at the following links: `https://github.com/HLasse/data-centric-synthetic-data` or `https://github.com/vanderschaarlab/data-centric-synthetic-data`.

Table 1: Summarised performance for the baseline condition (no data-centric processing) across all datasets. Classification is measured by AUROC, model selection and feature selection by Spearman's Rank Correlation, and statistical fidelity by inverse KL divergence. Numbers show bootstrapped mean and 95% CI. The best-performing model by task is in bold.

| Generative Model | Classification | Model Selection | Feature Selection | Statistical fidelity |
|---|---|---|---|---|
| Real data | 0.866 (0.855, 0.877) | 1.0 | 1.0 | 1.0 |
| bayesian_network | 0.622 (0.588, 0.656) | 0.155 (0.055, 0.264) | 0.091 (-0.001, 0.188) | 0.998 (0.998, 0.999) |
| ctgan | 0.797 (0.769, 0.823) | **0.519** (0.457, 0.579) | 0.63 (0.557, 0.691) | 0.979 (0.967, 0.987) |
| ddpm | **0.813** (0.781, 0.844) | 0.508 (0.446, 0.573) | 0.635 (0.546, 0.718) | 0.846 (0.668, 0.972) |
| nflow | 0.737 (0.713, 0.761) | 0.354 (0.288, 0.427) | 0.415 (0.34, 0.485) | 0.975 (0.968, 0.981) |
| tvae | 0.792 (0.764, 0.818) | 0.506 (0.436, 0.565) | **0.675** (0.63, 0.722) | 0.966 (0.953, 0.978) |

## 4.1 Data

We assess our framework on a filtered version of the Tabular Classification from Numerical features benchmark suite from [40]. To reduce computational costs, we filter the benchmark suite to only include datasets with less than 100.000 samples and less than 50 features which reduced the number of datasets from 16 to 11. The datasets span several domains and contain a highly varied number of samples and features (see B for more details.). Notably, the datasets have been preprocessed to meet a series of criteria to ensure their suitability for benchmarking tasks. For instance, the datasets have at least 5 features and 3000 samples, are not too easy to classify, have missing values removed, have balanced classes, and only contain low cardinality features.

## 4.2 Generative models

To cover a representative sample of the space of generative models, we evaluate 5 different models with different architectures as reviewed in 2: bayesian networks (`bayesian_network`), conditional tabular generative adversarial network (`ctgan`), tabular variational autoencoder (`tvae`), normalizing flow (`nflow`), diffusion model for tabular data (`ddpm`).

## 4.3 Supervised classification model training

The variety of models employed in our study includes: extreme gradient boosting (xgboost), random forest, logistic regression, decision tree, k-nearest neighbors, support vector classifier, gaussian naive bayes, and multi-layer perception. It is the ranking of these models that is evaluated for the model selection task. Given the large number of models, we restrict the classification results in the main paper to be from the xgboost model. Feature selection results are reported for xgboost models. Classification and feature selection results for the other classifiers can be found in Appendix C.

## 4.4 Experimental procedure

**Main study** The experimental process followed the structure outlined in Sec. 3 and Figure 2, repeated for each of the 11 datasets, 5 generative models, 10 random seeds, and all permutations of pre- and postprocessing methods for each of the three data-centric methods (Cleanlab, Data-IQ, and Data Maps). We comprehensively evaluate the results across classification performance, model selection, feature selection, and statistical fidelity. In total, we fit more than **8000** generative models.

**Impact of label noise** To assess the impact of label noise on the effect of data-centric pre- and postprocessing, we carried out an analogous experiment to the main study, on the Covid mortality dataset [41]. Here, we introduce label noise to $\mathbb{D}_{\text{train}}$ before applying any processing. We study the impact of adding [0, 2, 4, 6, 8, 10] percent label noise.

All results reported in the main paper use Cleanlab as the data-centric method for both pre- and postprocessing. This decision was made to ensure clarity in the reported results and because Cleanlab was found to outperform Data-IQ and Data Maps in a simulated benchmark. For the benchmark of the data-centric methods as well as results using Data-IQ and Data Maps, we refer to Appendix C.

Table 2: Percentage increase in performance from baseline, i.e., no data-centric pre- or postprocessing (as seen in Table 1), per generative model for each pre- and postprocessing strategy, averaged across all datasets and seeds.

| Generative Model | Preprocessing Strategy | Postprocessing Strategy | Classification | Model Selection | Feature Selection | Statistical Fidelity |
|---|---|---|---|---|---|---|
| bayesian_network | baseline | no_hard | 0.31 (-4.89, 5.98) ↑ | -27.73 (-86.26, 32.7) ↓ | -52.26 (-166.02, 47.61) ↓ | -0.007 (-0.054, 0.046) ↓ |
| | easy_hard | baseline | 0.35 (-4.98, 6.03) ↑ | 92.18 (31.99, 151.42) ↑ | 9.79 (-110.47, 129.68) ↑ | -0.013 (-0.054, 0.027) ↓ |
| | | no_hard | 0.8 (-4.4, 6.34) ↑ | 27.25 (-31.99, 90.28) ↑ | -9.39 (-132.22, 107.39) ↓ | -0.023 (-0.067, 0.021) ↓ |
| ctgan | baseline | no_hard | 1.23 (-2.03, 4.44) ↑ | 11.47 (-1.92, 24.28) ↑ | 3.66 (-6.48, 12.61) ↑ | -0.054 (-1.423, 0.805) ↓ |
| | easy_hard | baseline | -0.78 (-4.14, 2.88) ↓ | -1.12 (-13.34, 11.47) ↓ | 0.15 (-10.66, 10.13) ↑ | -0.006 (-1.061, 0.804) ↓ |
| | | no_hard | 0.37 (-2.96, 3.8) ↑ | 14.73 (2.13, 26.68) ↑ | 0.18 (-10.04, 9.43) ↑ | -0.119 (-1.218, 0.701) ↓ |
| ddpm | baseline | no_hard | 0.63 (-3.55, 4.24) ↑ | -6.35 (-20.41, 7.86) ↓ | 1.7 (-10.67, 13.37) ↑ | -0.166 (-19.424, 15.964) ↓ |
| | easy_hard | baseline | 0.65 (-2.84, 4.06) ↑ | 7.0 (-5.87, 20.19) ↑ | 1.58 (-10.92, 13.92) ↑ | 0.356 (-16.583, 13.893) ↑ |
| | | no_hard | 1.32 (-2.06, 4.68) ↑ | 5.98 (-7.27, 18.63) ↑ | 4.19 (-6.95, 16.16) ↑ | 0.284 (-16.721, 14.567) ↑ |
| nflow | baseline | no_hard | 1.05 (-2.45, 3.97) ↑ | 1.22 (-16.88, 18.41) ↑ | 5.82 (-11.28, 21.68) ↑ | 0.035 (-0.743, 0.688) ↑ |
| | easy_hard | baseline | 0.7 (-2.64, 3.81) ↑ | 2.37 (-15.28, 20.63) ↑ | 9.63 (-8.01, 25.65) ↑ | 0.022 (-0.752, 0.625) ↑ |
| | | no_hard | 1.64 (-1.76, 4.81) ↑ | 4.66 (-13.67, 22.84) ↑ | 7.28 (-11.54, 25.04) ↑ | 0.052 (-0.705, 0.654) ↑ |
| tvae | baseline | no_hard | 1.1 (-2.29, 4.38) ↑ | -3.58 (-18.44, 11.01) ↓ | -0.13 (-6.84, 6.38) ↓ | -0.053 (-1.418, 1.113) ↓ |
| | easy_hard | baseline | -0.25 (-3.46, 3.16) ↓ | -5.83 (-19.45, 6.73) ↓ | 1.67 (-6.03, 8.17) ↑ | 0.24 (-0.941, 1.296) ↑ |
| | | no_hard | 0.71 (-2.59, 3.95) ↑ | 0.11 (-13.79, 14.38) ↑ | 3.41 (-3.38, 9.89) ↑ | 0.199 (-0.929, 1.285) ↑ |

### 4.4.1 Evaluation metrics

The classification performance is evaluated in terms of the area under the receiver operating characteristic curve (AUROC), model selection performance as Spearman's Rank Correlation between the ranking of the supervised classification models trained on the original data and the supervised classification models trained on the synthetic data, and feature selection performance as Spearman's Rank Correlation between the ranking of features in an xgboost model trained on the original data and an xgboost model trained on the synthetic data.

## 5 Results

**Statistical fidelity is insufficient for evaluating generative models.** Measures of statistical fidelity fail to capture variability in performance on downstream tasks, as shown in Table 1. Surprisingly, the worst performing model across all tasks (bayesian network), has the highest inverse KL-divergence of all synthetic datasets, which should indicate a strong resemblance to the original data. Conversely, the lowest inverse KL-divergence is found for ddpm which is one of the consistently best performing models.

**Practical guidance:** The benchmarking results illustrate that when selecting a generative model, even if the statistical fidelity appears similar, different generative models may perform differently on the 3 downstream tasks (classification, model selection, feature selection). Hence, beyond statistical fidelity, practitioners should understand which aspect is most crucial for their purpose to guide selection of the generative model.

**Different generative models for different tasks.** As shown in Table 1, training on synthetic data leads to a marked decline in classification performance compared to real data, as well as highly differing model and feature rankings. The effect differs largely by generative model, where CTGAN, TabDDPM, and TVAE most closely retain the characteristics of the real data. No one model is superior across all tasks. Specifically, TabDPPM achieves the highest classification performance, CTGAN performs best in model selection, and TVAE excels in feature selection. These findings indicate that one should test a range of generative models and consider the trade-offs in data utility before publishing synthetic data. Additionally, Appendix C reveals that although there are slight differences in performance based on the supervised model type, the overall pattern of results remains consistent across generative models.

**Practical guidance:** No generative model reigns supreme (highlighting the inherent challenge of synthetic tabular data). However, over tabular data sets, we show that *CTGAN* and *TVAE* offer the best trade-off between high statistical fidelity and strong performance on the three downstream tasks.

**Data-centric methods can improve the utility of synthetic data.** The addition of data-centric pre- and postprocessing strategies has a generally positive effect across all tasks as seen in Table 2 and Figure 3, despite resulting in lower statistical fidelity. In terms of classification performance, 13 out of 15 evaluations showed a net improvement, with gains up to 1.64% better classification

performance compared to not processing the data. Model selection exhibited more pronounced effects, particularly for bayesian networks, which demonstrated the greatest variability overall. While the model selection results for TVAE decreased following data-centric processing, the other generative models saw positive effects, with performance improvements ranging from 4.66% to 92%. Regarding feature selection, 12 out of 15 evaluations demonstrated a net benefit of data-centric processing, resulting in improvements of 3.41% to 9.79% in Spearman's rank correlation. The benefit of data-centric processing was found to be statistically significant for classification and feature selection (see Appendix D for details).

**Practical guidance:** Before releasing a synthetic dataset, practitioners are advised to apply the data-centric methods studied in this paper as an add-on. This will ensure enhanced utility of the synthetic data in terms of classification performance, model selection, and feature selection.

**Data-centric processing provides benefits across levels of label noise.** Data-centric pre- and postprocessing lead to consistently higher performance across tasks for all augmented datasets. As shown in Figure 4, the magnitude of the effect of data-centric processing decreases with higher levels of label noise, particularly above 8%, although this effect is not statistically significant. Even though the level of statistical fidelity decreased marginally by applying data-centric processing, data-centric processing led to statistically significant increases in performance on all three tasks.

**Practical guidance:** Fitting generative models on noisy "real-world" data can lead to sub-optimal downstream performance despite seemingly high statistical fidelity. Data-centric methods are especially useful at reasonable levels of label noise, typically below 8%. Therefore, we recommend their application when fitting generative models on real-world datasets.

## 5.1 Limitations and future work

Our work delves into the performance-driven aspects of synthetic data generation, focusing primarily on data utility and statistical fidelity, particularly within the realm of tabular data. While tabular data is highly diverse and contains many intricacies, we also recognize several directions for further exploration. Our current framework, while rooted in tabular data, hints at the broader applicability to other data types such as text and images. Accommodating our framework to these modalities would require further work on modality-specific tasks. For instance, images or text do not possess a direct analog to feature selection. Such disparities underscore the need for a bespoke benchmarking methodology tailored to each specific data type.

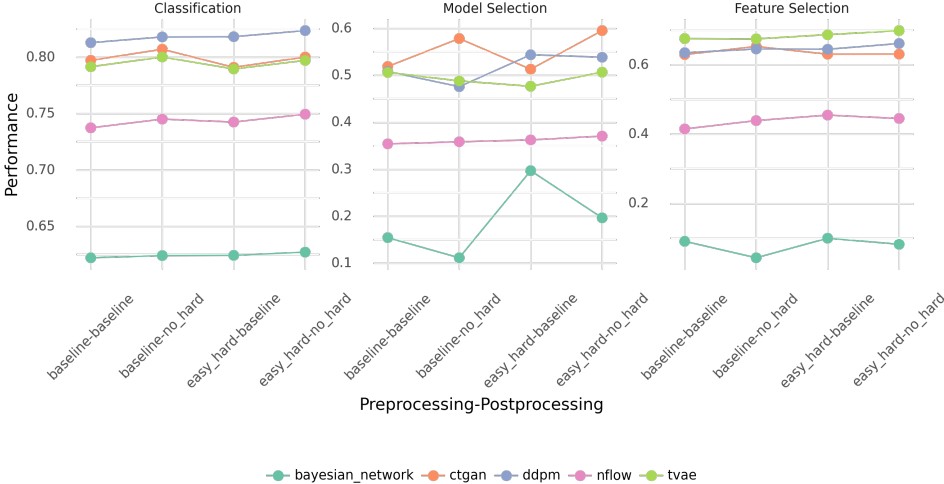

Figure 3: Average performance across all datasets for each generative model by pre- and postprocessing method.

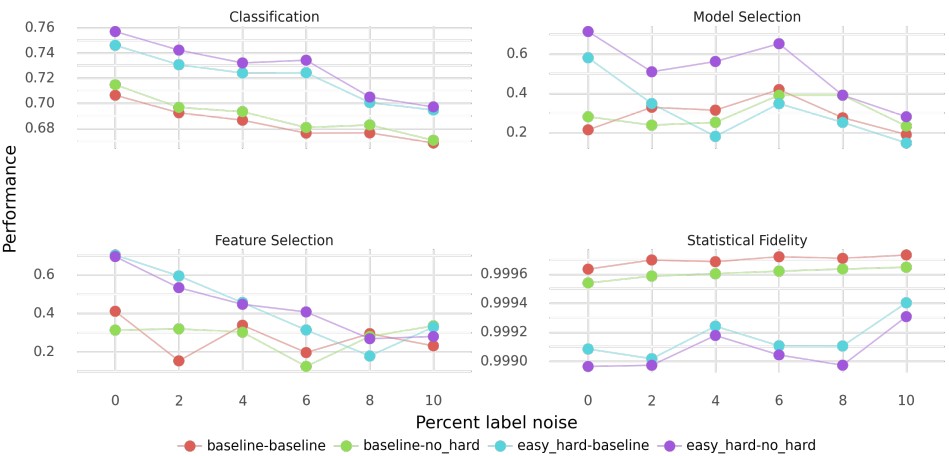

Figure 4: Performance of a single generative model (TabDDPM) on the Covid mortality dataset with varying levels of label noise across the pre- and postprocessing conditions.

# 6 Conclusion

This research provides novel insights into integrating data-centric AI techniques into synthetic tabular data generation. First, we introduce a framework to evaluate the integration of data profiles for creating more representative synthetic data. Second, we confirm that statistical fidelity alone is insufficient for assessing synthetic data's utility, as it may overlook important nuances impacting downstream tasks. Third, the choice of generative model significantly influences synthetic data quality and utility. Last, incorporating data-centric methods consistently improves the utility of synthetic data across varying levels of label noise. Our study demonstrates the potential of data-centric AI techniques to enhance synthetic data's representation of real-world complexities, opening avenues for further exploration at their intersection.

## Acknowledgements

This work was partially supported by DeiC National HPC (g.a. 2022-H2-10). NS is supported by the Cystic Fibrosis Trust. LH was supported by a travel grant from A.P. Møller Fonden til Lægevidenskabens Fremme and is supported by grants from the Lundbeck Foundation (grant number: R344-2020-1073), the Central Denmark Region Fund for Strengthening of Health Science (grant number: 1-36-72-4-20), and The Danish Agency for Digitisation Investment Fund for New Technologies (grant number: 2020–6720).

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

# Appendix

## A   Additional details

This appendix provides additional details on the data-centric methods used in our study, as well as outlining limitations and future work.

### A.1   Data-centric AI methods

Our paper considers different data-centric AI methods to perform data characterization, which forms the basis of our data profiles. As discussed in the main manuscript, the primary difference between these methods is their scoring mechanism $S$. The three approaches considered in this paper next, i.e. Cleanlab, Data-IQ and Data Maps will be described next. Importantly, these methods are applicable to tabular data models like XGBoost, unlike several other data-centric methods which are only applicable to differentiable models. Recall that the data characterization methods apply a threshold $\tau$ to $S$. The implementation of $\tau$ in this work is described in Appendix C.3.

**Cleanlab [34].**   Cleanlab estimates the joint distribution of noisy and true labels, thereby characterizing data into profiles easy and hard, where hard data points might indicate mislabeling. It operates on the output of classification models and can be applied to any modality. We focus on Cleanlab in the main manuscript, however, our framework could be applied with any of the following data-centric tools.

**Data-IQ [18].**   Data-IQ is a training dynamics-based method, which characterizes data based on the aleatoric uncertainty (inherent data uncertainty), i.e. $\mathbb{E}[\mathcal{P}(x,\vartheta)(1 - \mathcal{P}(x,\vartheta))]$. We are able to extract three data profiles: easy, ambiguous and hard. Typically, these "ambiguous" and "hard" examples are harmful to model performance and might be mislabeled or "dirty".

**Data Maps [33].**   Data Maps is a training dynamics-based method, which characterizes data based on the variability/epistemic uncertainty (model uncertainty), i.e. $\mathbb{V}[\mathcal{P}(x,\vartheta)]$. We are able to extract three data profiles: easy, ambiguous and hard. Typically, these "ambiguous" and "hard" examples are harmful to model performance and might be mislabeled or "dirty".

## B   Experimental details

By accident, the results reported in the main paper were based on data from 8 random seeds instead of the intended 10, as stated in the paper. To maintain consistency in the results, the results reported in the Appendix are also presented with 8 seeds.

### B.1   Computational resources

All experiments were conducted using NVIDIA T4 16GB GPUs. The total number of GPU hours spent across all experiments is approximately 1000. All experiments were performed on the UCloud platform.

### B.2   Hyperparameter tuning of generative models

In order to reduce the computational costs of running the main experiment, we conducted a hyperparameter search on the Adult dataset instead of tuning the parameters for each dataset in our benchmark. For each model (`bayesian_network`, `ctgan`, `ddpm`, `nflow`, `tvae`), we conducted a search over the hyperparameter space defined for each model in the synthcity [5] implementation. We used Optuna [42] to conduct 20 trials for each model. The best hyperparameters for each model are listed in Table 3.

### B.3   Classification model set

Throughout the experiments, we fit a set of models from the scikit-learn library, namely, `XGBClassifier`, `RandomForestClassifier`, `LogisticRegression`,

Table 3: Hyperparameters used for the generative models.

| Model | Parameters |
|---|---|
| bayesian_network | struct_learning_search_method: hillclimb, struct_learning_score: bic |
| ctgan | generator_n_layers_hidden: 2, generator_n_units_hidden: 50, generator_nonlin: tanh,     n_iter: 1000, generator_dropout: 0.0575, discriminator_n_layers_hidden: 4, discriminator_n_units_hidden: 150, discriminator_nonlin: relu |
| ddpm | lr: 0.0009375080542687667, batch_size: 2929, num_timesteps: 998, n_iter: 1051, is_classification: True |
| nflow | n_iter: 1000, n_layers_hidden: 10, n_units_hidden: 98, dropout: 0.11496088236749386, batch_norm: True, lr: 0.0001, linear_transform_type: permutation, base_transform_type: rq-autoregressive, batch_size: 512, |
| tvae | n_iter: 300, lr: 0.0002, decoder_n_layers_hidden: 4, weight_decay: 0.001, batch_size: 256, n_units_embedding: 200, decoder_n_units_hidden: 300, decoder_nonlin: elu, decoder_dropout: 0.194325119117226, encoder_n_layers_hidden: 1, encoder_n_units_hidden: 450, encoder_nonlin: leaky_relu, encoder_dropout: 0.04288563703094718, |

DecisionTreeClassifier, KNeighborsClassifier, SVC, GaussianNB, and MLPClassifier. All hyperparameters were kept at their default value.

## B.4  Datasets

**Main experiments** As described in Sec 4.1. we used the "Tabular benchmark numerical classification" downloaded from OpenML (suite id 337), filtered to only include datasets with less than 100.000 samples and less than 50 features for our main experiments. The number of samples, features, and links to the datasets are provided in Table 4.

**Noise dataset** The dataset used to investigate the impact of label noise was the Covid mortality dataset from [41]. The data includes data on patients with Covid with a label for whether the patient will die within 14 days. The dataset contains 6882 samples and 21 features.

**Adult dataset** The adult dataset is a classic machine learning dataset, containing 48842 and 14 samples, with the task of predicting whether an individual's income exceeds $50k per year based on census data.

Table 4: Datasets used in the benchmark.

| Task id | Dataset name | n_features | n_samples | URL |
|---|---|---|---|---|
| 361055 | credit | 10 | 16714 | https://openml.org/search?type=task&collections.id=337&sort=runs&id=361055 |
| 361060 | electricity | 7 | 38474 | https://openml.org/search?type=task&collections.id=337&sort=runs&id=361060 |
| 361062 | pol | 26 | 10082 | https://openml.org/search?type=task&collections.id=337&sort=runs&id=361062 |
| 361063 | house_16H | 16 | 13488 | https://openml.org/search?type=task&collections.id=337&sort=runs&id=361063 |
| 361065 | MagicTelescope | 10 | 13376 | https://openml.org/search?type=task&collections.id=337&sort=runs&id=361065 |
| 361066 | bank-marketing | 7 | 10578 | https://openml.org/search?type=task&collections.id=337&sort=runs&id=361066 |
| 361070 | eye_movements | 20 | 7608 | https://openml.org/search?type=task&collections.id=337&sort=runs&id=361070 |
| 361273 | Diabetes130US | 7 | 71090 | https://openml.org/search?type=task&collections.id=337&sort=runs&id=361273 |
| 361275 | default-of-credit-card-clients | 20 | 13272 | https://openml.org/search?type=task&collections.id=337&sort=runs&id=361275 |
| 361277 | california | 8 | 20634 | https://openml.org/search?type=task&collections.id=337&sort=runs&id=361277 |
| 361278 | heloc | 22 | 10000 | https://openml.org/search?type=task&collections.id=337&sort=runs&id=361278 |

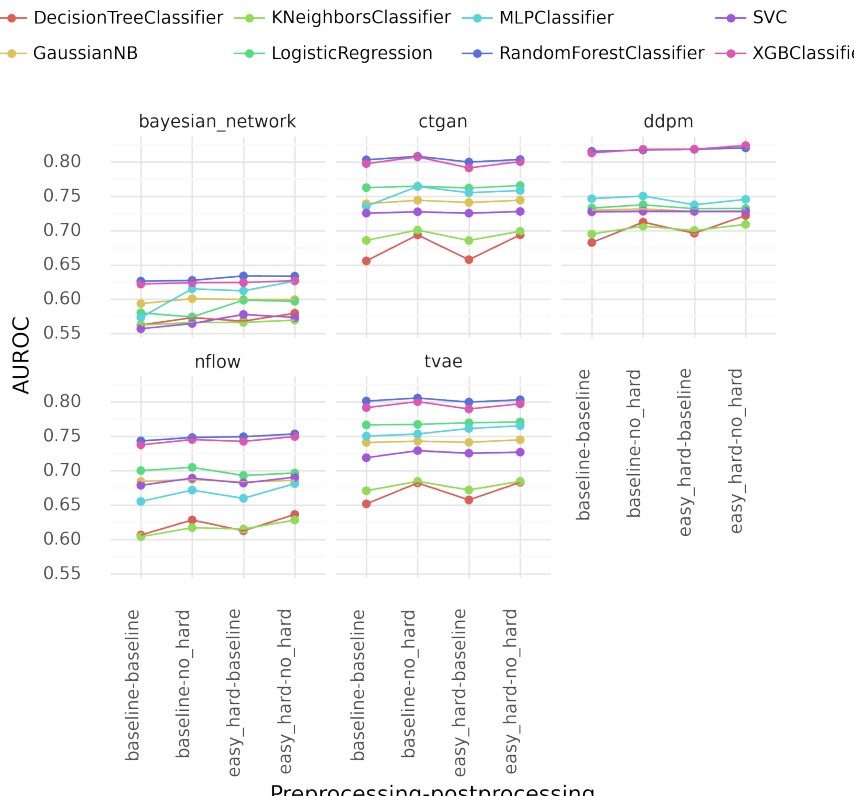

Figure 5: Average classification performance across all datasets and generative models by pre- and postprocessing strategy and classification model type, using Cleanlab as the data-centric method.

## C   Additional experiments

### C.1   Performance on other models

Table 5 shows classification performance across all classification models, averaged over all datasets. Random Forest and XGBoost performs the best across all combinations of pre- and postprocessing, hence, we chose to focus our results in the main manuscript on the XGBoost model. As evident from Table 5 and Figure 5, the benefit of data-centric pre- and postprocessing is consistent across model types.

Table 5: Average classification performance across all datasets and generative models by pre- and postprocessing strategy and classification model type, using Cleanlab as the data-centric method. Numbers in parentheses indicate 95% bootstrapped CI.

| Preprocessing Strategy | Postprocessing Strategy | Classification model | AUROC |
|---|---|---|---|
| baseline | baseline | DecisionTreeClassifier | 0.64 (0.63, 0.64) |
| | | GaussianNB | 0.7 (0.69, 0.71) |
| | | KNeighborsClassifier | 0.65 (0.64, 0.66) |
| | | LogisticRegression | 0.72 (0.7, 0.73) |
| | | MLPClassifier | 0.7 (0.68, 0.71) |
| | | RandomForestClassifier | **0.76** (0.75, 0.78) |
| | | SVC | 0.69 (0.67, 0.7) |
| | | XGBClassifier | **0.76** (0.75, 0.77) |
| | no_hard | DecisionTreeClassifier | 0.66 (0.65, 0.67) |
| | | GaussianNB | 0.71 (0.69, 0.72) |
| | | KNeighborsClassifier | 0.66 (0.65, 0.67) |
| | | LogisticRegression | 0.72 (0.7, 0.73) |
| | | MLPClassifier | 0.72 (0.7, 0.73) |
| | | RandomForestClassifier | **0.77** (0.75, 0.78) |
| | | SVC | 0.69 (0.68, 0.71) |
| | | XGBClassifier | **0.77** (0.75, 0.78) |
| easy_hard | baseline | DecisionTreeClassifier | 0.64 (0.63, 0.65) |
| | | GaussianNB | 0.7 (0.69, 0.72) |
| | | KNeighborsClassifier | 0.65 (0.64, 0.66) |
| | | LogisticRegression | 0.72 (0.7, 0.73) |
| | | MLPClassifier | 0.71 (0.69, 0.72) |
| | | RandomForestClassifier | **0.77** (0.75, 0.78) |
| | | SVC | 0.69 (0.68, 0.71) |
| | | XGBClassifier | 0.76 (0.75, 0.77) |
| | no_hard | DecisionTreeClassifier | 0.67 (0.66, 0.68) |
| | | GaussianNB | 0.71 (0.69, 0.72) |
| | | KNeighborsClassifier | 0.66 (0.65, 0.67) |
| | | LogisticRegression | 0.72 (0.71, 0.73) |
| | | MLPClassifier | 0.72 (0.71, 0.73) |
| | | RandomForestClassifier | **0.77** (0.76, 0.78) |
| | | SVC | 0.7 (0.68, 0.71) |
| | | XGBClassifier | **0.77** (0.75, 0.78) |
| Real data | | DecisionTreeClassifier | 0.74 (0.73, 0.75) |
| | | GaussianNB | 0.73 (0.72, 0.74) |
| | | KNeighborsClassifier | 0.73 (0.71, 0.74) |
| | | LogisticRegression | 0.78 (0.77, 0.79) |
| | | MLPClassifier | 0.77 (0.76, 0.79) |
| | | RandomForestClassifier | 0.86 (0.85, 0.87) |
| | | SVC | 0.74 (0.72, 0.75) |
| | | XGBClassifier | **0.87** (0.86, 0.88) |

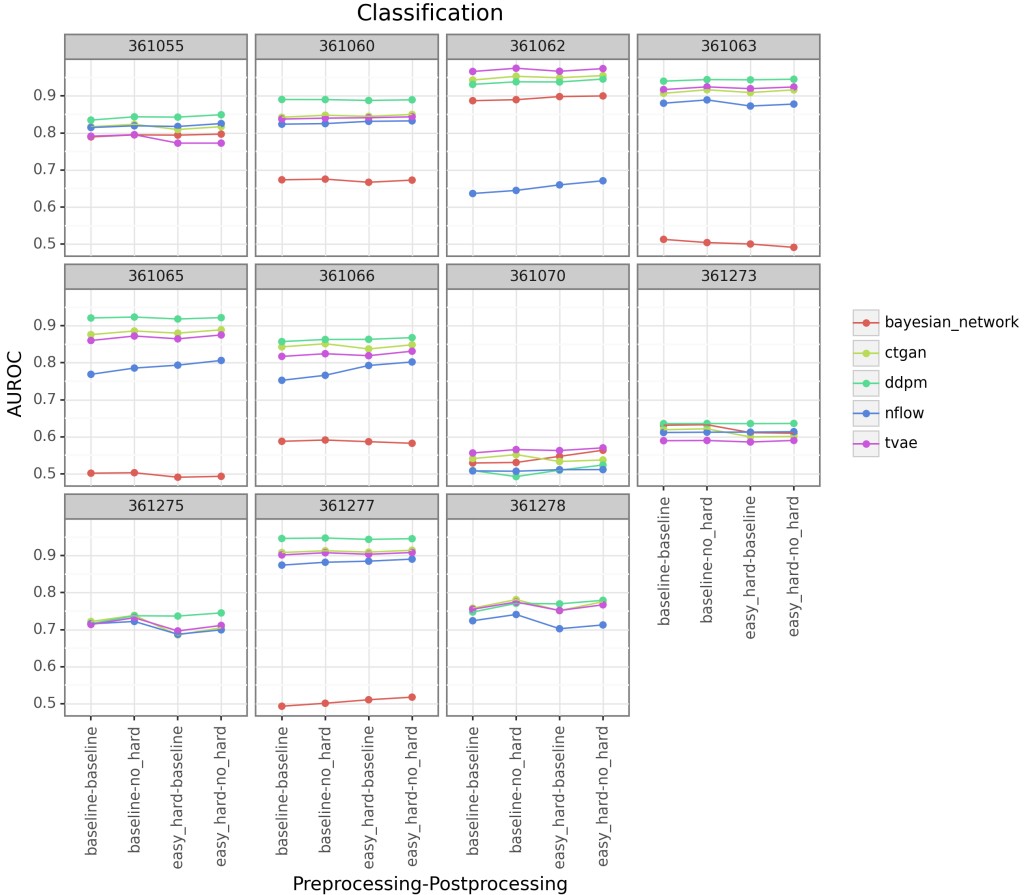

Figure 6: Classification performance by pre- and postprocessing strategy for XGBoost across all datasets using Cleanlab as the data-centric method.

## C.2 Performance by dataset

As shown in Figures 6 to 8 the relative ranking of the five generative models remains largely consistent across datasets. Specifically, the classification performance results shown in Figure 6 indicate that `ddpm` produces synthetic data that best retains the ability to train classification models.

The model selection results shown in Figure 7 display higher variance across datasets and pre- and postprocessing methods. This variance can likely be attributed to the findings presented in Table 5, which demonstrate that the performance of several classification models is relatively similar. Consequently, minor performance variations caused by data-centric processing can significantly change the models' relative ranking.

As shown in Figure 8, data generated with the `ddpm` model tends to lead to the best ranking of important features. Consistently with the findings in classification performance and model selection, the `bayesian_network` model tends to perform the worst across datasets.

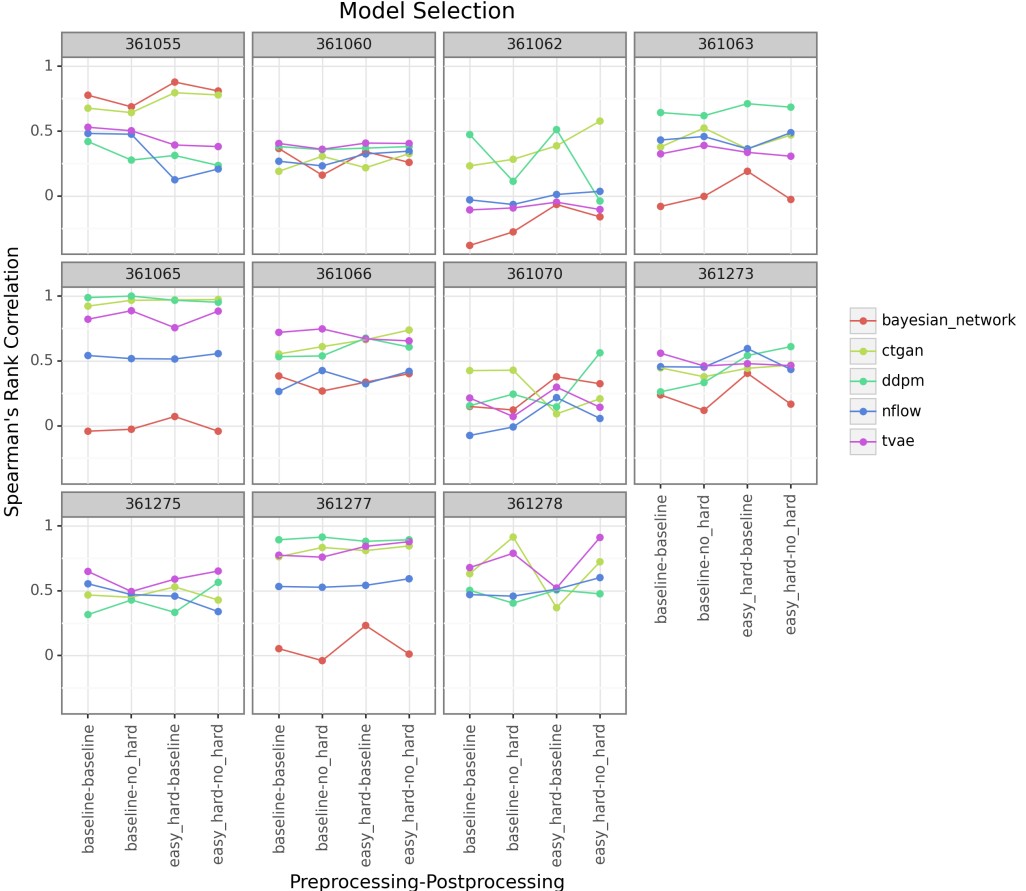

Figure 7: Model selection performance by pre- and postprocessing strategy across all datasets using Cleanlab as the data-centric method.

### C.3 Identifying optimal thresholds for the data-centric methods

To identify the optimal threshold $\tau$ for the scoring function $S$ for each of the data-centric methods, we constructed four simulated datasets with varying levels of label noise. In particular, we generated datasets with 10 features and 10 samples, 10 features and 50 samples, 50 features and 10 samples, and 50 features and 50 samples. The correlation of each feature with the outcome label was sampled from a uniform distribution ranging from -0.7 to 0.7.

For each dataset, we randomly flipped a proportion (0, 0.02, 0.04, 0.06, 0.08, 0.1) of the outcome labels and recorded the indices of the flipped indices.

**Data-IQ**. Following the Data-IQ implementation, we explored two methods for setting $\tau$: one based on whether the aleatoric uncertainty of the sample was below a certain percentile, and the other based on whether the aleatoric uncertainty was below a certain value. In both cases, to categorize a sample as 'hard', the average predicted probability of the correct label (throughout the model training) had to be $\leq 0.25$ and the aleatoric uncertainty had to be below $\tau$. 'Easy' samples were defined as samples where the average predicted probability of the correct label was $\geq 0.75$ and the aleatoric uncertainty was below $\tau$. Any remaining samples were categorized as 'ambiguous'. We tested setting $\tau$ at the [20, 30, 40, 50, 60, 70, 80] percentile, and using raw values of [0.1, 0.125, 0.15, 0.175, 0.2].

**Data Maps**. The data-profiling procedure for Data Maps followed the same approach as Data-IQ, with the exception that the epistemic uncertainty was used instead of the aleatoric uncertainty. We tested the same values of $\tau$ as used for Data-IQ.

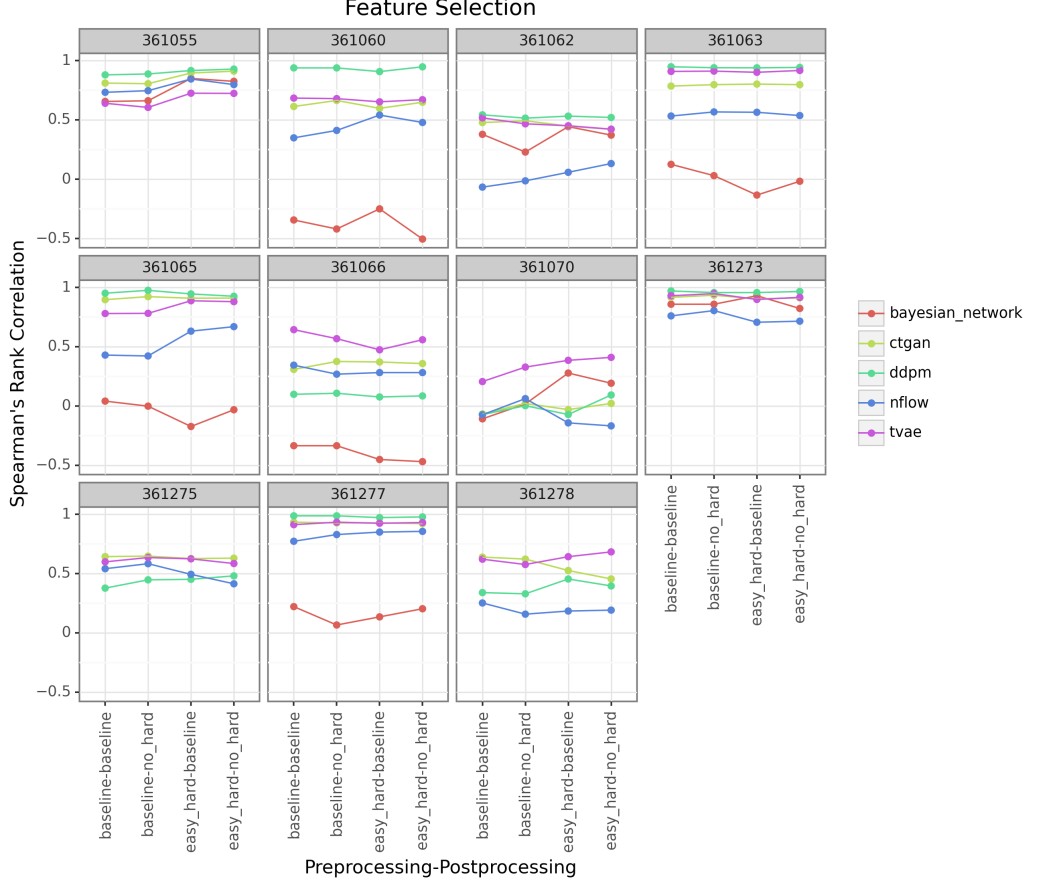

Figure 8: Feature selection performance by pre- and postprocessing strategy for XGBoost across all datasets using Cleanlab as the data-centric method.

**Cleanlab**. The Cleanlab package provides a 'label quality' score which was used to set $\tau$ to categorize samples as 'easy' or 'hard'. We tested using the default score as well as the values [0.1, 0.125, 0.15, 0.175, 0.2]. For details of the theoretical basis of Cleanlab's thresholds, we refer to [34], Section 3.1.

**Experiment**. For each augmented dataset (varying levels of label noise), we applied each of the data-centric methods and tested all the specified values of $\tau$. We recorded the number of samples identified as 'hard', $n_{hard}$ as well as the number of samples that were correctly identified as hard (i.e., a flipped label) $n_{hard}^{correct}$. Recall was calculated as $\frac{n_{hard}^{correct}}{n_{flipped}}$, precision as $\frac{n_{hard}^{correct}}{n_{hard}}$ and the F1 score as $2 \cdot \frac{\text{precision} \cdot \text{recall}}{\text{recall} + \text{precision}}$

**Results**. As shown in Table 6, Cleanlab achieves markedly higher F1 scores and recall compared to Data Maps and Data-IQ, albeit with slightly lower precision. Given the substantially higher F1 and recall scores, we chose to focus the main analysis on Cleanlab. We selected the threshold that maximised F1 for the main experiments, resulting in a $\tau$ of 0.2 for all methods.

Table 6: Average results across all simulated datasets and levels of noise. Numbers in parentheses indicate the standard deviation. The highest value per data-centric method and metric is highlighted in bold. The value in bold for $\tau$ indicates our chosen threshold for the main experiments.

| Method | Type | $\tau$ | F1 | Recall | Precision |
|---|---|---|---|---|---|
| Cleanlab | cutoff | Default | 0.407 (0.29) | **0.633** (0.30) | 0.332 (0.28) |
| | | 0.100 | 0.384 (0.30) | 0.354 (0.30) | **0.5** (0.33) |
| | | 0.125 | 0.403 (0.30) | 0.393 (0.30) | 0.478 (0.33) |
| | | 0.150 | 0.41 (0.30) | 0.42 (0.31) | 0.45 (0.32) |
| | | 0.175 | 0.417 (0.30) | 0.45 (0.32) | 0.428 (0.31) |
| | | **0.200** | **0.42** (0.29) | 0.478 (0.32) | 0.408 (0.30) |
| Data-IQ | cutoff | 0.100 | 0.083 (0.15) | 0.051 (0.10) | 0.507 (0.42) |
| | | 0.125 | 0.118 (0.19) | 0.076 (0.13) | **0.562** (0.40) |
| | | 0.150 | 0.16 (0.22) | 0.11 (0.16) | 0.547 (0.40) |
| | | 0.175 | 0.206 (0.24) | 0.153 (0.18) | 0.523 (0.40) |
| | | **0.200** | **0.221** (0.24) | **0.17** (0.19) | 0.513 (0.40) |
| | percentile | 20 | 0.033 (0.05) | 0.019 (0.03) | 0.279 (0.41) |
| | | 30 | 0.057 (0.07) | 0.038 (0.05) | 0.433 (0.43) |
| | | 40 | 0.075 (0.08) | 0.052 (0.06) | 0.371 (0.39) |
| | | 50 | 0.099 (0.10) | 0.067 (0.07) | 0.373 (0.39) |
| | | 60 | 0.138 (0.12) | 0.093 (0.08) | 0.407 (0.40) |
| | | 70 | 0.17 (0.16) | 0.12 (0.11) | 0.4 (0.40) |
| | | 80 | 0.199 (0.21) | 0.146 (0.15) | 0.438 (0.40) |
| Data Maps | cutoff | 0.100 | 0.21 (0.24) | 0.16 (0.18) | 0.426 (0.39) |
| | | 0.125 | 0.216 (0.24) | 0.166 (0.19) | 0.47 (0.40) |
| | | 0.150 | 0.219 (0.24) | 0.169 (0.19) | **0.513** (0.40) |
| | | 0.175 | **0.221** (0.24) | **0.17** (0.19) | **0.513** (0.40) |
| | | **0.200** | **0.221** (0.24) | **0.17** (0.19) | **0.513** (0.40) |
| | percentile | 20 | 0.058 (0.07) | 0.033 (0.04) | 0.327 (0.37) |
| | | 30 | 0.082 (0.10) | 0.048 (0.06) | 0.331 (0.37) |
| | | 40 | 0.103 (0.12) | 0.064 (0.07) | 0.334 (0.37) |
| | | 50 | 0.124 (0.14) | 0.08 (0.09) | 0.336 (0.37) |
| | | 60 | 0.144 (0.17) | 0.097 (0.11) | 0.376 (0.38) |
| | | 70 | 0.163 (0.19) | 0.114 (0.13) | 0.419 (0.39) |
| | | 80 | 0.183 (0.21) | 0.133 (0.15) | 0.423 (0.39) |

## C.4 Development of Figure 1

The results shown in Figure 1 were obtained performing the following steps: First, the Adult dataset was split into an 80% training set and a 20% test set. Second, each of the five generative models (with the hyperparameters specified in Table 3) was trained on the training set. Third, A synthetic dataset, $D_{synth}$, with the same dimensions as the training set was generated using each generative model. Fourth, the inverse KL divergence between $D_{synth}$ and the training data was calculated, an XGBoost model was trained on $D_{synth}$ and evaluated on the test set to measure the performance, and Cleanlab was applied to $D_{synth}$ to extract the proportion of easy and hard examples from each synthetic dataset.

Table 7: Percentage increase in performance from baseline, i.e., no data-centric pre- or postprocessing (as seen in Table 1), per generative model for each pre- and postprocessing strategy, averaged across all datasets and seeds, using Data-IQ as the data-centric method.

| Generative Model | Preprocessing Strategy | Postprocessing Strategy | Classification | Model Selection | Feature Selection |
|---|---|---|---|---|---|
| bayesian_network | baseline | no_hard | 0.27 (-6.25, 6.66)↑ | -3.49 (-98.06, 92.25)↓ | -9.71 (-61.47, 44.94)↓ |
| | easy_ambi_hard | baseline | -0.92 (-7.92, 6.3)↓ | 2.33 (-91.86, 115.12)↑ | -27.86 (-90.12, 33.36)↓ |
| | | no_hard | -0.64 (-7.8, 7.64)↓ | 63.18 (-35.66, 164.73)↑ | -20.85 (-76.47, 35.22)↓ |
| | easy_hard | baseline | -0.33 (-7.34, 6.76)↓ | 61.24 (-9.69, 131.4)↑ | -64.55 (-128.52, -4.18)↓ |
| | | no_hard | -0.1 (-6.77, 6.99)↓ | 29.84 (-33.33, 93.02)↑ | -81.7 (-135.04, -23.46)↓ |
| ctgan | baseline | no_hard | 0.29 (-2.26, 2.75)↑ | 2.13 (-9.95, 13.86)↑ | 0.43 (-7.16, 7.78)↑ |
| | easy_ambi_hard | baseline | -0.3 (-3.24, 2.37)↓ | 8.51 (-5.0, 21.51)↑ | -0.56 (-9.67, 8.99)↓ |
| | | no_hard | -0.17 (-2.7, 2.38)↓ | 6.96 (-5.12, 18.29)↑ | 0.56 (-9.08, 10.0)↑ |
| | easy_hard | baseline | -0.08 (-2.83, 2.44)↓ | -12.02 (-25.19, 1.38)↓ | -4.02 (-10.59, 2.31)↓ |
| | | no_hard | 0.13 (-2.51, 2.83)↑ | -13.11 (-25.82, 0.12)↓ | -3.97 (-11.06, 2.72)↓ |
| ddpm | baseline | no_hard | 0.2 (-3.48, 4.02)↑ | -0.23 (-16.27, 15.02)↓ | 1.39 (-11.55, 13.66)↑ |
| | easy_ambi_hard | baseline | -5.08 (-9.82, 0.18)↓ | -28.29 (-47.34, -9.98)↓ | -4.45 (-18.54, 9.27)↓ |
| | | no_hard | -5.06 (-9.84, -0.29)↓ | -18.2 (-34.64, -0.06)↓ | -5.43 (-20.07, 8.01)↓ |
| | easy_hard | baseline | -3.19 (-7.5, 0.98)↓ | -11.51 (-26.59, 3.12)↓ | -10.6 (-22.83, 0.19)↓ |
| | | no_hard | -2.99 (-7.3, 1.19)↓ | -17.46 (-34.07, -1.47)↓ | -11.44 (-23.96, 0.79)↓ |
| nflow | baseline | no_hard | 0.35 (-2.39, 3.07)↑ | 1.6 (-15.26, 19.65)↑ | 4.66 (-10.09, 19.15)↑ |
| | easy_ambi_hard | baseline | 0.16 (-2.58, 2.79)↑ | 6.32 (-15.85, 27.4)↑ | -1.57 (-17.11, 15.53)↓ |
| | | no_hard | 0.5 (-2.14, 3.24)↑ | 6.16 (-14.67, 25.8)↑ | 0.21 (-15.45, 16.06)↑ |
| | easy_hard | baseline | 1.02 (-1.91, 3.62)↑ | 5.65 (-13.41, 25.13)↑ | 6.18 (-9.79, 21.52)↑ |
| | | no_hard | 1.03 (-2.06, 3.67)↑ | 3.46 (-15.85, 22.85)↑ | 4.82 (-10.25, 18.61)↑ |
| tvae | baseline | no_hard | 0.19 (-2.63, 3.11)↑ | 3.27 (-11.38, 16.1)↑ | 0.97 (-4.51, 6.27)↑ |
| | easy_ambi_hard | baseline | -0.43 (-3.47, 2.35)↓ | -6.48 (-22.11, 7.53)↓ | -1.28 (-9.16, 6.16)↓ |
| | | no_hard | -0.27 (-3.37, 2.71)↓ | -4.03 (-20.01, 10.97)↓ | -0.06 (-7.57, 6.89)↓ |
| | easy_hard | baseline | 1.02 (-1.93, 3.63)↑ | -18.03 (-32.56, -3.15)↓ | -0.8 (-5.49, 3.83)↓ |
| | | no_hard | 1.07 (-1.86, 3.97)↑ | -19.84 (-34.66, -5.19)↓ | 0.9 (-4.16, 5.73)↑ |

## C.5   Results with Data-IQ and Datamaps

The following subsections show the main results using Data-IQ and Data Maps as the data-centric method. Note, that statistical fidelity metrics for Data-IQ and Data Maps are not included in the table to reduce the complexity of the table, as they do not demonstrate any correspondence with task-related performance.

### C.5.1   Data-IQ

As shown in Table 7 and Figure 9, the results from using Data-IQ are relatively similar to what's reported in Table 2. However, the impact of data-centric processing is weaker and more variable when using Data-IQ. This effect be attributed to Data-IQ's tendency to identify fewer 'hard' examples (as shown in Table 6 by markedly lower recall) and thereby being a less strong intervention than Cleanlab.

### C.5.2   Data Maps

The main results using Data Maps as the data-centric method are presented in Table 8 and Figure 10. Similarly to Data-IQ, the variation across performance metrics is generally higher when using Data Maps compared to Cleanlab. The benefits of Data Maps compared to no processing varies to some extent, with models like nflow seeming to draw some benefit from it, while other models do not see major gains. This might again be attributed to the relatively low recall and the small number of identified 'hard' samples with Data Maps, which reduces the strength of the intervention.

Table 8: Percentage increase in performance from baseline, i.e., no data-centric pre- or postprocessing (as seen in Table 1), per generative model for each pre- and postprocessing strategy, averaged across all datasets and seeds, using Data-IQ as the data-centric method.

| Generative Model | Preprocessing Strategy | Postprocessing Strategy | Classification | Model Selection | Feature Selection |
|---|---|---|---|---|---|
| bayesian_network | baseline | no_hard | 0.14 (-5.08, 6.41)↑ | -14.67 (-179.33, 148.67)↓ | -70.0 (-243.76, 116.63)↓ |
| | easy_ambi_hard | baseline | 0.87 (-5.18, 7.4)↑ | 190.67 (39.33, 340.67)↑ | 135.71 (-33.95, 301.53)↑ |
| | | no_hard | 1.2 (-4.46, 8.79)↑ | 199.33 (52.67, 341.33)↑ | 104.23 (-63.28, 272.74)↑ |
| | easy_hard | baseline | 2.18 (-3.58, 8.95)↑ | 262.0 (74.67, 443.33)↑ | -51.7 (-223.73, 137.24)↓ |
| | | no_hard | 2.46 (-3.46, 9.15)↑ | 213.33 (26.0, 400.0)↑ | -80.31 (-249.14, 118.38)↓ |
| ctgan | baseline | no_hard | 0.17 (-2.67, 2.72)↑ | 3.63 (-7.26, 14.19)↑ | 0.33 (-6.6, 6.91)↑ |
| | easy_ambi_hard | baseline | -0.57 (-3.39, 2.17)↓ | 7.54 (-4.08, 18.38)↑ | -2.52 (-11.64, 5.99)↓ |
| | | no_hard | -0.32 (-3.11, 2.47)↓ | 13.13 (1.34, 24.25)↑ | -3.6 (-13.19, 5.62)↓ |
| | easy_hard | baseline | -0.75 (-3.8, 2.14)↓ | -15.2 (-28.21, -1.96)↓ | -2.47 (-9.4, 4.63)↓ |
| | | no_hard | -0.65 (-3.62, 2.23)↓ | -10.34 (-22.79, 1.9)↓ | -3.13 (-9.47, 3.35)↓ |
| ddpm | baseline | no_hard | 0.13 (-3.82, 3.73)↑ | -10.19 (-24.66, 4.66)↓ | 1.34 (-12.26, 13.9)↑ |
| | easy_ambi_hard | baseline | -3.98 (-8.9, 0.5)↓ | -11.78 (-25.59, 1.7)↓ | -4.52 (-18.29, 8.67)↓ |
| | | no_hard | -4.03 (-8.96, 0.77)↓ | -19.23 (-35.29, -4.05)↓ | -4.14 (-17.14, 9.47)↓ |
| | easy_hard | baseline | -2.53 (-7.05, 1.58)↓ | -20.93 (-36.11, -5.92)↓ | -4.42 (-16.55, 6.67)↓ |
| | | no_hard | -2.82 (-7.49, 1.12)↓ | -16.27 (-28.99, -3.18)↓ | -1.53 (-12.95, 9.81)↓ |
| nflow | baseline | no_hard | 0.22 (-2.48, 2.75)↑ | 3.83 (-13.8, 21.77)↑ | -0.4 (-13.37, 13.22)↓ |
| | easy_ambi_hard | baseline | 0.77 (-1.89, 3.22)↑ | 8.13 (-10.21, 25.68)↑ | 4.43 (-10.51, 19.46)↑ |
| | | no_hard | 1.13 (-1.53, 3.75)↑ | -0.72 (-16.59, 14.75)↓ | 5.48 (-9.09, 19.53)↑ |
| | easy_hard | baseline | 1.23 (-1.5, 4.15)↑ | -3.03 (-18.74, 12.84)↓ | 9.02 (-5.26, 22.42)↑ |
| | | no_hard | 1.46 (-1.02, 4.2)↑ | 6.7 (-10.93, 22.49)↑ | 9.53 (-4.33, 21.96)↑ |
| tvae | baseline | no_hard | 0.21 (-2.65, 2.92)↑ | 4.5 (-10.54, 18.35)↑ | -1.23 (-6.94, 4.55)↓ |
| | easy_ambi_hard | baseline | -0.45 (-3.26, 2.43)↓ | -14.39 (-28.0, 0.3)↓ | -1.25 (-7.76, 5.62)↓ |
| | | no_hard | -0.26 (-3.14, 2.68)↓ | -14.68 (-30.55, 0.71)↓ | -0.67 (-8.08, 5.57)↓ |
| | easy_hard | baseline | 0.8 (-2.13, 3.5)↑ | -9.47 (-24.33, 6.34)↓ | -0.61 (-5.74, 4.4)↓ |
| | | no_hard | 0.88 (-1.97, 3.55)↑ | -10.3 (-24.33, 4.32)↓ | 0.31 (-4.84, 5.71)↑ |

## C.6 Performance on data with added label noise across all tasks and generative models

The performance of all generative models across all tasks (of which a subset is presented in Figure 4) is shown in Figure 11. The overall trend is consistent across generative models, although not all models are affected equally hard by varying levels of noise. For instance, tvae and ctgan seem to be more robust to noise than e.g. ddpm.

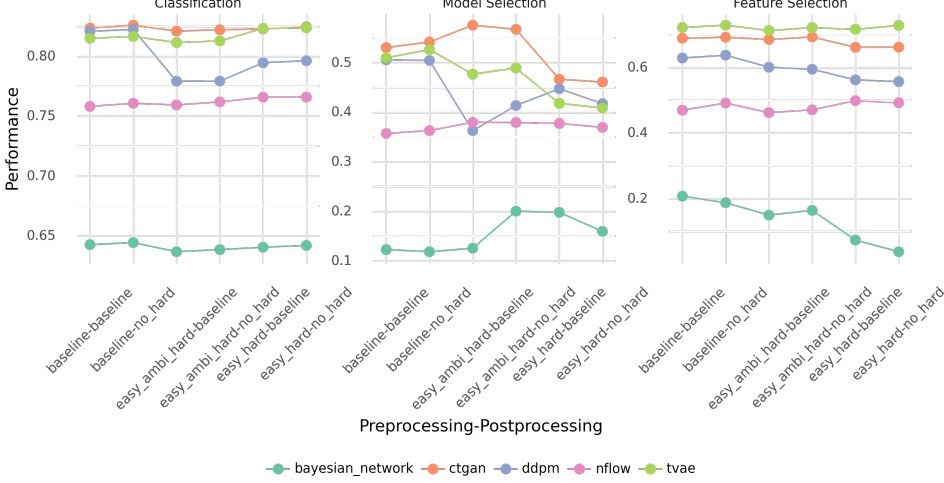

Figure 9: Average performance across all datasets for each generative model by pre- and postprocessing method using Data-IQ as data-centric method.

Table 9: Feature selection performance by classification model, averaged across all datasets and generative models using Cleanlab as the data-centric method. Numbers in parentheses indicate 95% bootstrapped CI.

| Preprocessing Strategy | Postprocessing Strategy | Classification Model | Spearman's Rank Correlation |
| --- | --- | --- | --- |
| baseline | baseline | DecisionTreeClassifier | 0.68 (0.65, 0.71) |
| | | RandomForestClassifier | 0.71 (0.68, 0.74) |
| | | XGBClassifier | 0.51 (0.47, 0.55) |
| | no_hard | DecisionTreeClassifier | 0.67 (0.63, 0.69) |
| | | RandomForestClassifier | 0.69 (0.66, 0.73) |
| | | XGBClassifier | 0.51 (0.48, 0.56) |
| easy_hard | baseline | DecisionTreeClassifier | 0.69 (0.66, 0.72) |
| | | RandomForestClassifier | 0.73 (0.7, 0.76) |
| | | XGBClassifier | 0.52 (0.49, 0.56) |
| | no_hard | DecisionTreeClassifier | 0.67 (0.63, 0.7) |
| | | RandomForestClassifier | 0.71 (0.68, 0.74) |
| | | XGBClassifier | 0.53 (0.49, 0.56) |

## C.7 Feature selection results with other models

As indicated in Table 9 and Figure 12, the feature selection results reported in the main manuscript are stable across generative models and classification models. Across all comparisons, Spearman's rank correlation is calculated between the feature rankings of the same model type trained on $\mathbb{D}_{\text{train}}$ and $\mathbb{D}_{\text{synth}}^{\text{post}}$. In general, random forest and decision trees models tend to display higher concordance in their feature rankings between models trained on synthetic data and models trained on real data compared to xgboost models.

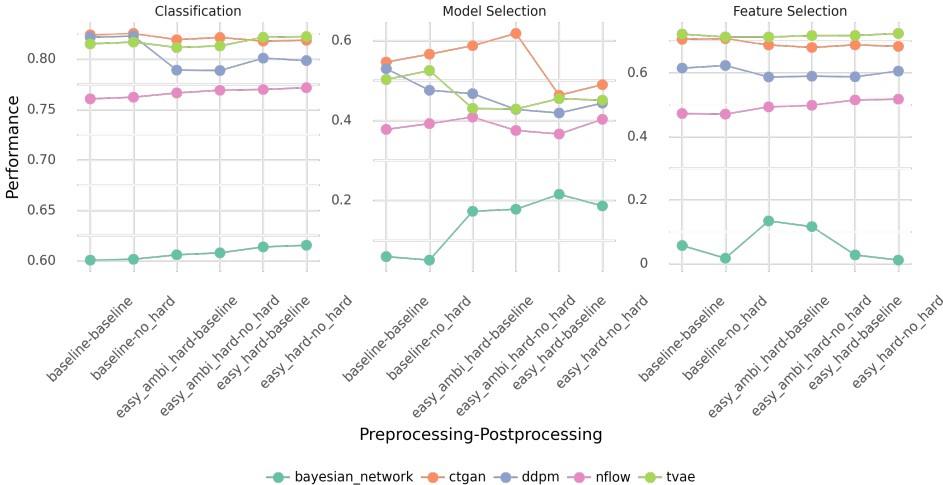

Figure 10: Average performance across all datasets for each generative model by pre- and postprocessing method using Data-IQ as the data-centric method.

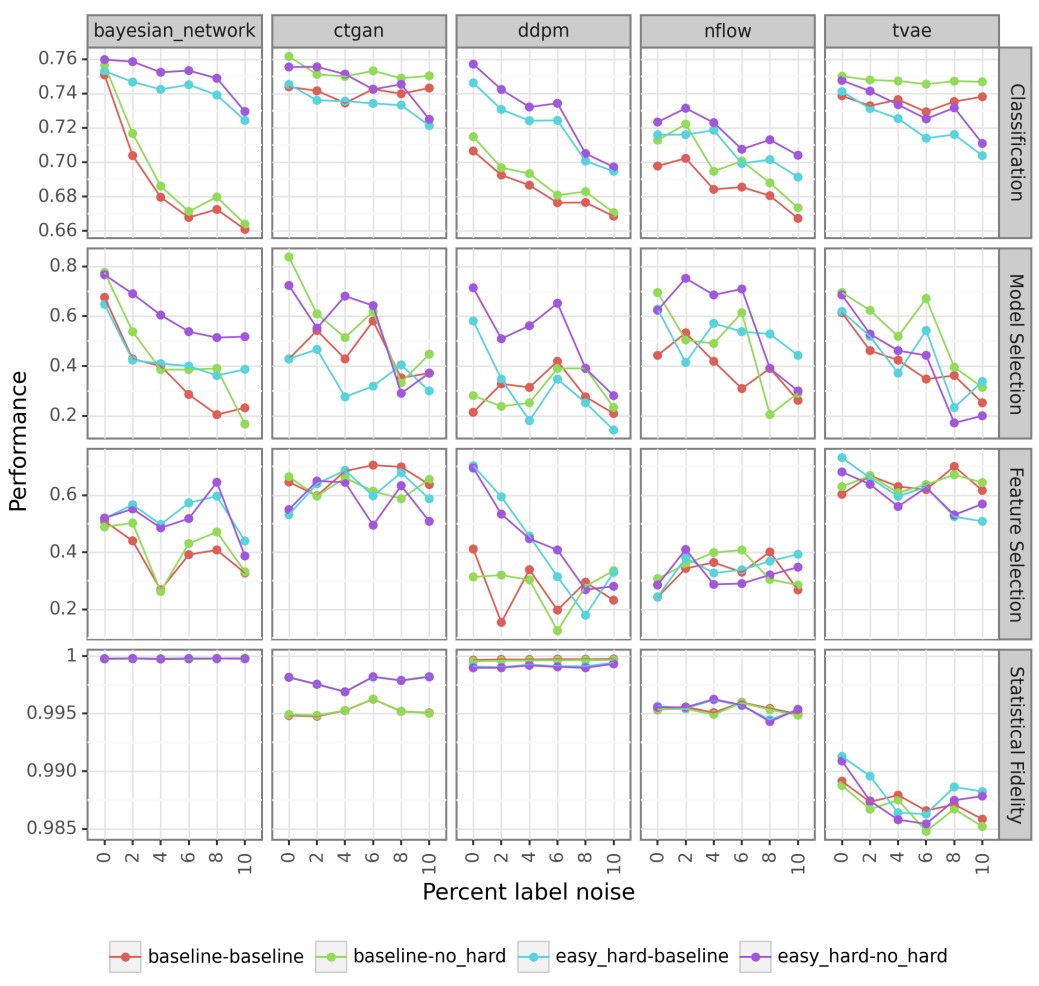

Figure 11: Performance across all models and tasks on the Covid mortality data with added label noise. Using Cleanlab as the data-centric method.

## C.8 Causal Discovery

As a proof of concept, we investigated whether data-centric processing might be useful for causal discovery. For this purpose, we used DAGMA [43] to estimate a causal graph, $DAG_{\text{train}}$ based on $\mathbb{D}_{\text{train}}$. $DAG_{\text{train}}$ was used as the reference graph, and compared to a causal graph estimated from $\mathbb{D}_{\text{synth}}^{\text{post}}$. Results for a single dataset are shown in Figure 13 in terms of Structural Hamming Distance (SHD). The results appear promising in this particular example, however, further work is required.

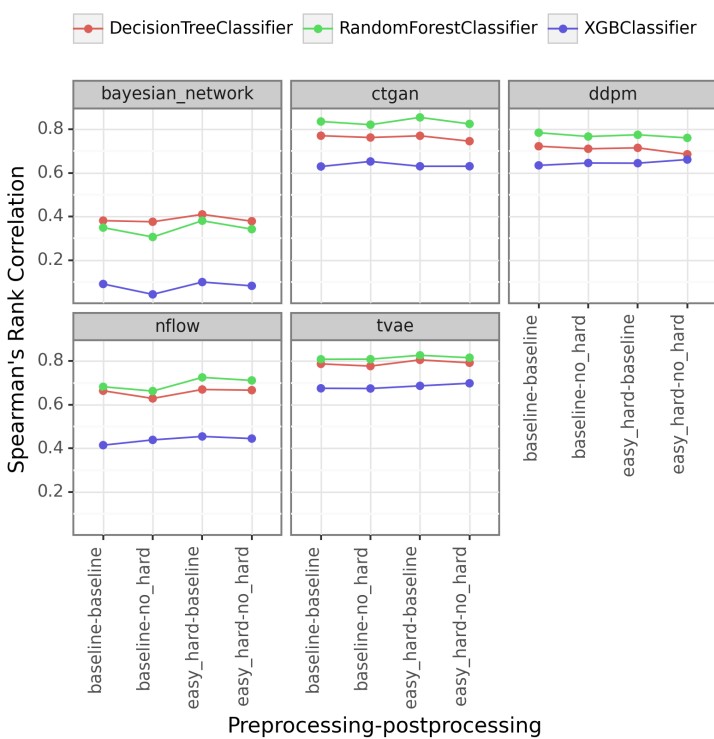

Figure 12: Mean feature selection performance across all datasets by classification model and generative model, using Cleanlab as the data-centric method.

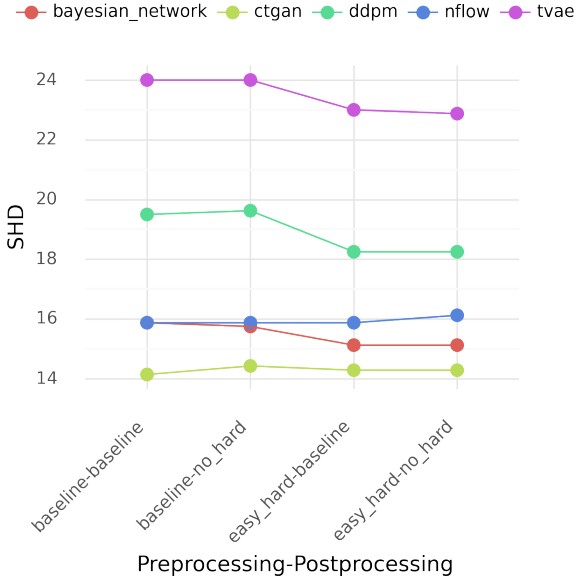

Figure 13: Causal graph discovery performance using Cleanlab as the data-centric method on a single dataset (id=361055).

## C.9 Distribution of Scores

### C.9.1 Main Experiment

To investigate the variability of the performance across the three main tasks, we show the distribution of performance metrics across all generative models, datasets, and processing strategies. The findings for classification performance, feature selection, and model selection are shown in Figure 14, Figure 15, and Figure 16, respectively. All figures are shown using Cleanlab as the data-centric method. Overall, the difficulty of the datasets is highly variable, and the variance by generative model is heterogeneous. In general, the best performing models, such as TabDDPM, exhibit comparatively lower variability across different seeds. In contrast, models with poorer performance, like Bayesian networks and normalizing flows, display higher variability across seeds.

The variance of the performance metrics is higher for the tasks of feature selection and model selection than classification. This discrepancy can likely be partly attributed to the inherent smoothness of the performance metrics. The model ranking score is established by accurately ranking the eight supervised classification models listed in Appendix B.3. Similarly, the feature selection score is determined by correctly ranking the metrics for feature importance across all features within the dataset - a quantity that naturally varies by dataset.

In contrast, classification performance relies on AUROC values computed on the test dataset, which encompasses a relatively large number of samples (1,141 in the smallest dataset). Due to the dissimilarity in the number of samples, models, and features, the metrics for model and feature ranking are inevitably more susceptible to larger variance. This occurs because disparities in a single element can lead to more substantial effects on the performance metrics in comparison to the impact of single elements for classification metric.

Classification

Figure 14: Distribution of classification performance across the random seeds by generative model (rows) and dataset ids (columns).

Feature Selection

Figure 15: Distribution of feature selection performance across the random seeds by generative model (rows) and dataset ids (columns).

## Model Selection

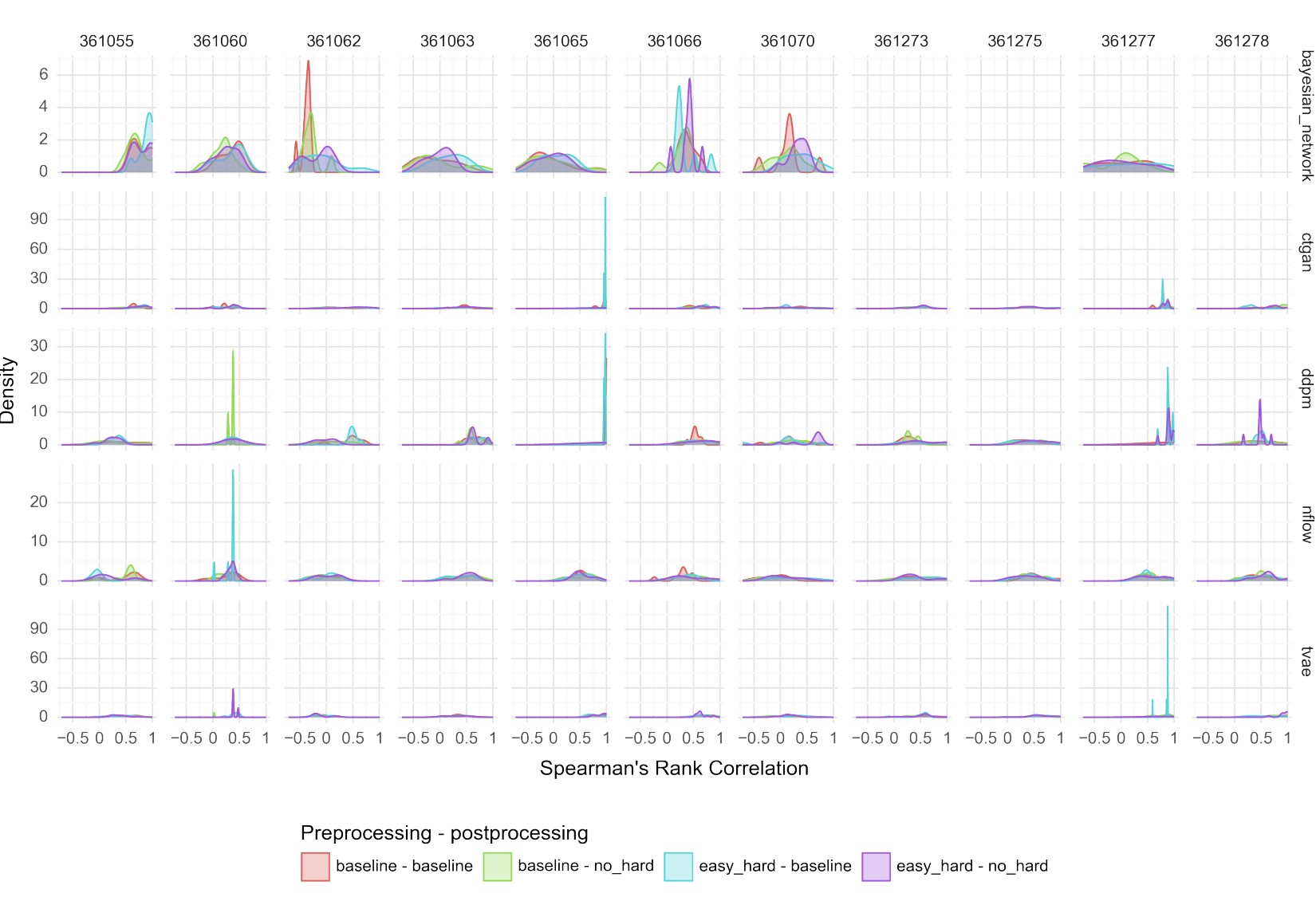

Figure 16: Distribution of model selection performance across the random seeds by generative model (rows) and dataset ids (columns).

### C.9.2 Label noise

To investigate the variability in the label noise experiments, we show the distribution of performance metrics across all generative models, proportions of label noise, and processing strategies. The findings for classification performance, feature selection, and model selection are shown in Figure 17, Figure 18, and Figure 19, respectively. Further, we show the performance across the generative models, summarized over the levels of label noise, in Table 10 and show the performance by the level of label noise, summarized over all generative models, in Table 11. All figures and tables are shown using Cleanlab as the data-centric method.

Classification

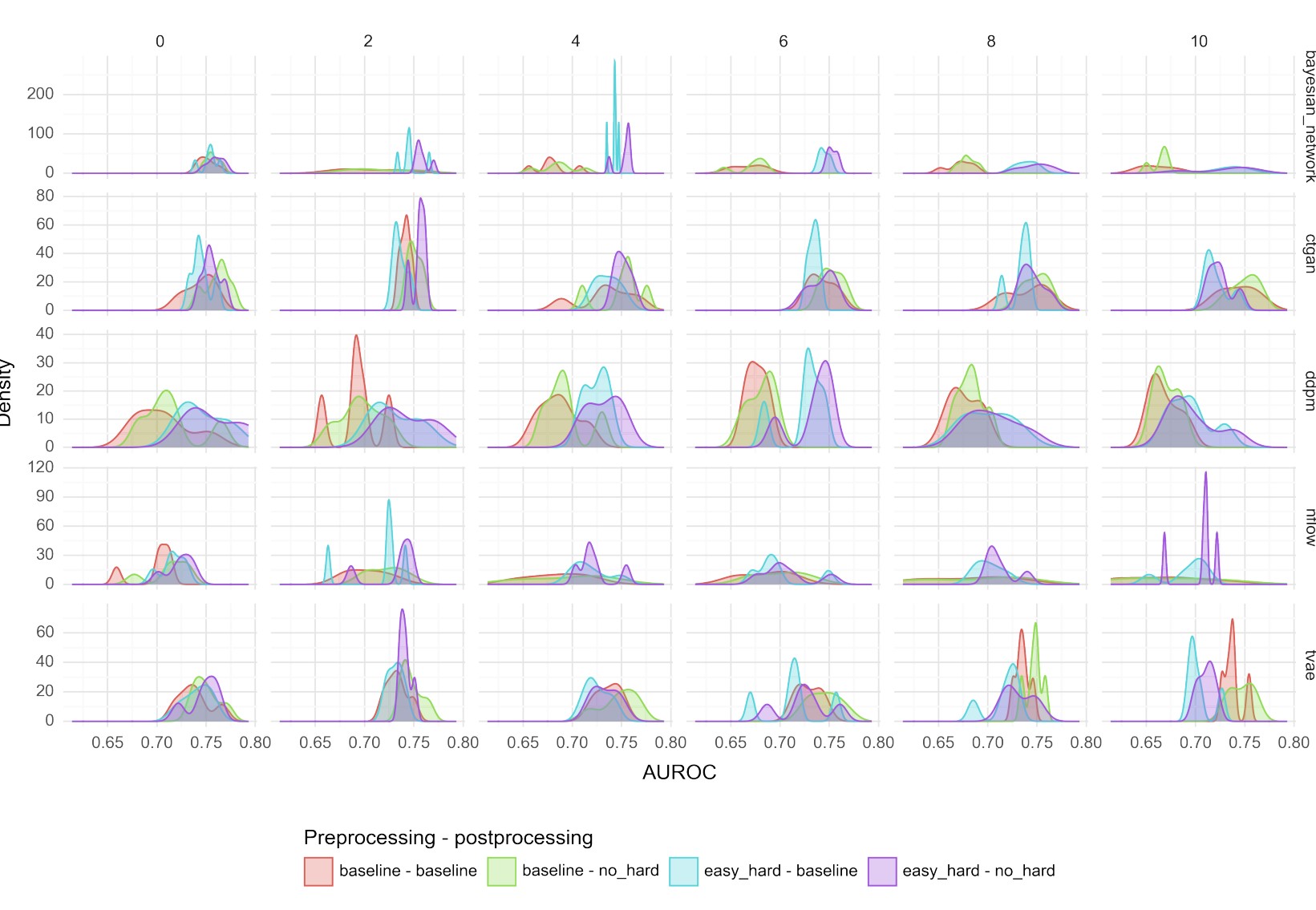

Figure 17: Distribution of classification performance across the random seeds by generative model (rows) and percent label noise (columns).

Feature Selection

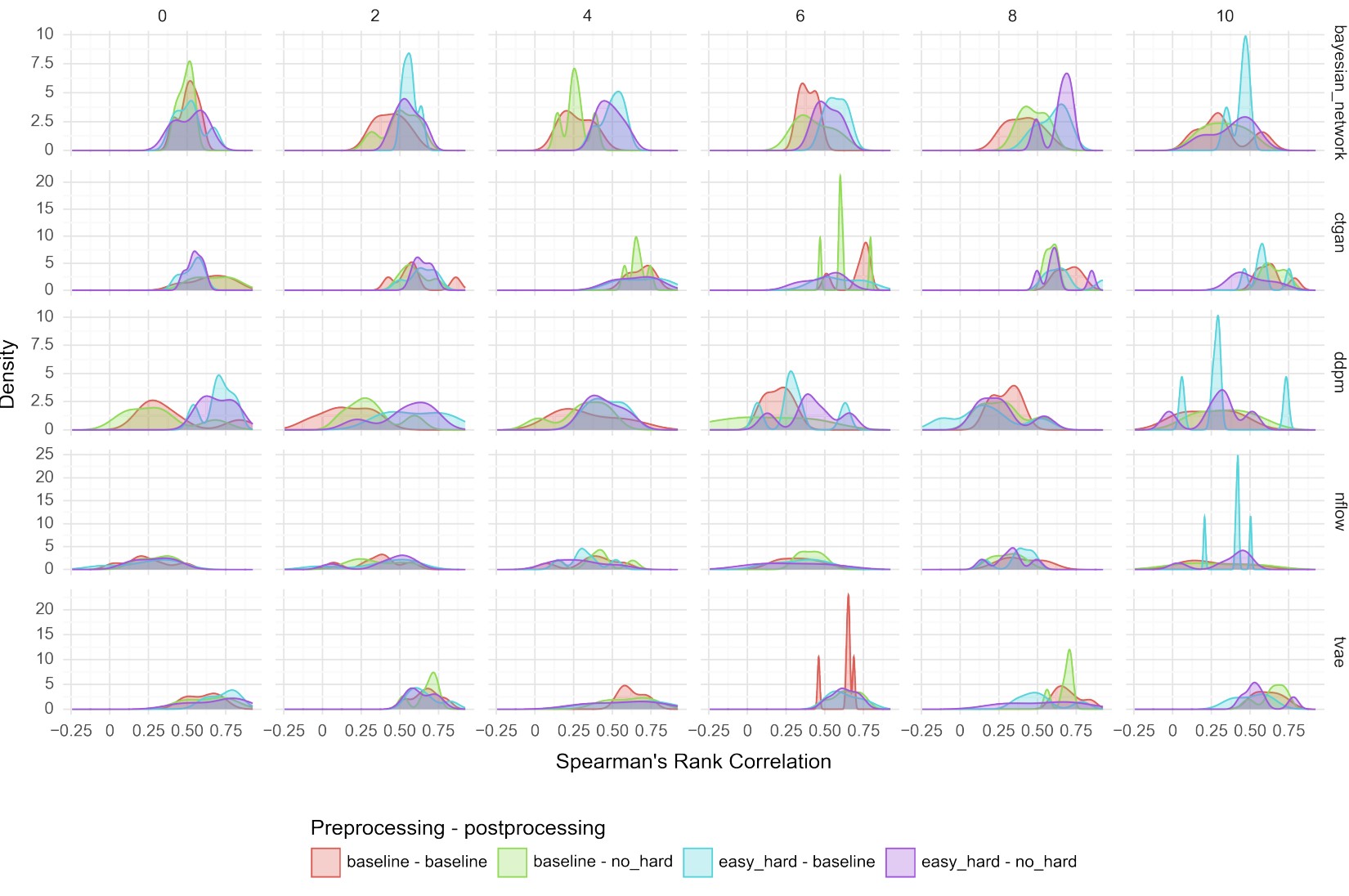

Figure 18: Distribution of feature selection performance across the random seeds by generative model (rows) and percent label noise (columns).

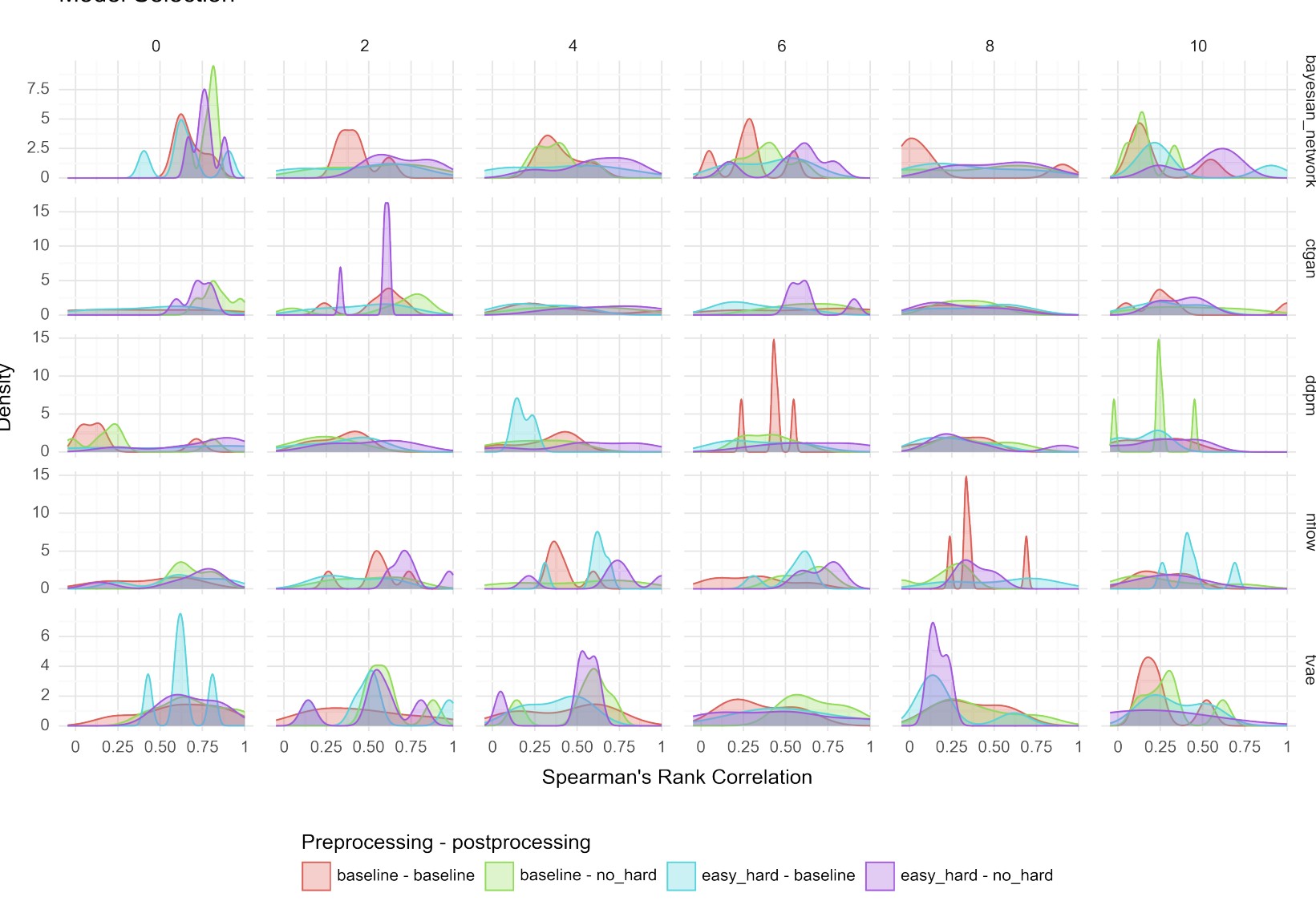

Figure 19: Distribution of model selection performance across the random seeds by generative model (rows) and percent label noise (columns).

Table 10: Summarised performance for the label noise experiments across all levels of label noise. Classification is measured by AUROC, feature selection and model selection by Spearman's Rank Correlation, and statistical fidelity by inverse KL divergence. Numbers show bootstrapped mean and 95% CI.

| Generative Model | Preprocessing Strategy | Postprocessing Strategy | Classification | Feature Selection | Model Selection | Statistical Fidelity |
|---|---|---|---|---|---|---|
| Real data | None | None | 0.766 (0.764, 0.767) | - | - | - |
| bayesian_network | baseline | baseline | 0.69 (0.678, 0.704) | 0.393 (0.349, 0.435) | 0.376 (0.284, 0.46) | 1.0 (1.0, 1.0) |
| | | no_hard | 0.697 (0.684, 0.71) | 0.417 (0.371, 0.462) | 0.45 (0.355, 0.539) | 1.0 (1.0, 1.0) |
| | easy_hard | baseline | 0.742 (0.737, 0.747) | 0.535 (0.505, 0.566) | 0.44 (0.336, 0.526) | 1.0 (1.0, 1.0) |
| | | no_hard | 0.751 (0.744, 0.756) | 0.522 (0.478, 0.563) | 0.608 (0.536, 0.683) | 1.0 (1.0, 1.0) |
| ctgan | baseline | baseline | 0.741 (0.734, 0.746) | 0.662 (0.62, 0.7) | 0.451 (0.346, 0.559) | 0.995 (0.994, 0.996) |
| | | no_hard | 0.752 (0.747, 0.757) | 0.63 (0.601, 0.661) | 0.56 (0.467, 0.651) | 0.995 (0.994, 0.996) |
| | easy_hard | baseline | 0.734 (0.73, 0.738) | 0.62 (0.581, 0.666) | 0.366 (0.292, 0.442) | 0.998 (0.997, 0.998) |
| | | no_hard | 0.746 (0.74, 0.751) | 0.58 (0.54, 0.625) | 0.544 (0.46, 0.626) | 0.998 (0.997, 0.998) |
| ddpm | baseline | baseline | 0.685 (0.677, 0.693) | 0.272 (0.208, 0.338) | 0.294 (0.23, 0.36) | 1.0 (1.0, 1.0) |
| | | no_hard | 0.69 (0.682, 0.699) | 0.28 (0.212, 0.356) | 0.298 (0.229, 0.371) | 1.0 (1.0, 1.0) |
| | easy_hard | baseline | 0.72 (0.71, 0.729) | 0.43 (0.347, 0.511) | 0.309 (0.217, 0.398) | 0.999 (0.999, 0.999) |
| | | no_hard | 0.728 (0.717, 0.739) | 0.439 (0.365, 0.51) | 0.518 (0.421, 0.626) | 0.999 (0.999, 0.999) |
| nflow | baseline | baseline | 0.686 (0.674, 0.697) | 0.325 (0.268, 0.378) | 0.393 (0.328, 0.457) | 0.995 (0.995, 0.996) |
| | | no_hard | 0.699 (0.686, 0.711) | 0.344 (0.294, 0.396) | 0.467 (0.37, 0.556) | 0.995 (0.995, 0.996) |
| | easy_hard | baseline | 0.707 (0.699, 0.716) | 0.342 (0.286, 0.395) | 0.521 (0.45, 0.586) | 0.995 (0.995, 0.996) |
| | | no_hard | 0.717 (0.71, 0.724) | 0.324 (0.26, 0.383) | 0.577 (0.488, 0.663) | 0.995 (0.995, 0.996) |
| tvae | baseline | baseline | 0.735 (0.731, 0.74) | 0.64 (0.605, 0.671) | 0.41 (0.327, 0.498) | 0.987 (0.986, 0.989) |
| | | no_hard | 0.747 (0.743, 0.752) | 0.643 (0.606, 0.679) | 0.537 (0.45, 0.619) | 0.987 (0.985, 0.988) |
| | easy_hard | baseline | 0.722 (0.715, 0.729) | 0.608 (0.558, 0.663) | 0.437 (0.356, 0.519) | 0.988 (0.987, 0.99) |
| | | no_hard | 0.732 (0.725, 0.738) | 0.601 (0.547, 0.658) | 0.415 (0.315, 0.506) | 0.987 (0.986, 0.989) |

## C.10  Other Measures of Statistical Fidelity

Table 12 shows additional measures of statistical fidelity (Wasserstein Distance (WD) and Maximum Mean Discrepancy (MMD)) for the main experiment.

Table 11: Summarised performance for the label noise experiments across all generative models. Classification is measured by AUROC, feature selection and model selection by Spearman's Rank Correlation, and statistical fidelity by inverse KL divergence. Numbers show bootstrapped mean and 95% CI. A value of 'None' in Pre- and postprocessing strategy indicates that the results are for models trained on the real data.

| Prop. label noise | Preprocessing Strategy | Postprocessing Strategy | Classification | Feature Selection | Model Selection | Statistical Fidelity |
|---|---|---|---|---|---|---|
| 0.0 | None | None | 0.775 (0.773, 0.777) | - | - | - |
| | baseline | baseline | 0.728 (0.717, 0.738) | 0.482 (0.398, 0.559) | 0.475 (0.365, 0.588) | 0.996 (0.994, 0.997) |
| | | no_hard | 0.739 (0.728, 0.749) | 0.481 (0.411, 0.558) | 0.657 (0.545, 0.748) | 0.996 (0.994, 0.997) |
| | easy_hard | baseline | 0.74 (0.733, 0.747) | 0.546 (0.46, 0.621) | 0.581 (0.474, 0.688) | 0.997 (0.995, 0.998) |
| | | no_hard | 0.749 (0.741, 0.756) | 0.546 (0.471, 0.62) | 0.703 (0.626, 0.769) | 0.997 (0.995, 0.998) |
| 0.02 | None | None | 0.771 (0.768, 0.774) | - | - | - |
| | baseline | baseline | 0.715 (0.705, 0.725) | 0.441 (0.344, 0.529) | 0.459 (0.387, 0.536) | 0.995 (0.993, 0.997) |
| | | no_hard | 0.727 (0.716, 0.737) | 0.489 (0.411, 0.558) | 0.503 (0.396, 0.604) | 0.995 (0.993, 0.997) |
| | easy_hard | baseline | 0.732 (0.724, 0.739) | 0.568 (0.492, 0.631) | 0.434 (0.346, 0.53) | 0.996 (0.994, 0.998) |
| | | no_hard | 0.746 (0.738, 0.753) | 0.557 (0.498, 0.611) | 0.607 (0.529, 0.68) | 0.996 (0.994, 0.998) |
| 0.04 | None | None | 0.767 (0.765, 0.77) | - | - | - |
| | baseline | baseline | 0.704 (0.691, 0.718) | 0.457 (0.37, 0.54) | 0.397 (0.321, 0.484) | 0.996 (0.994, 0.997) |
| | | no_hard | 0.714 (0.699, 0.728) | 0.447 (0.369, 0.525) | 0.432 (0.34, 0.526) | 0.995 (0.993, 0.997) |
| | easy_hard | baseline | 0.729 (0.723, 0.735) | 0.513 (0.447, 0.581) | 0.362 (0.276, 0.449) | 0.996 (0.993, 0.998) |
| | | no_hard | 0.738 (0.732, 0.745) | 0.485 (0.41, 0.555) | 0.599 (0.507, 0.704) | 0.996 (0.993, 0.998) |
| 0.06 | None | None | 0.765 (0.762, 0.767) | - | - | - |
| | baseline | baseline | 0.7 (0.688, 0.713) | 0.449 (0.367, 0.534) | 0.389 (0.303, 0.481) | 0.996 (0.993, 0.998) |
| | | no_hard | 0.71 (0.697, 0.726) | 0.443 (0.351, 0.526) | 0.536 (0.463, 0.61) | 0.995 (0.993, 0.997) |
| | easy_hard | baseline | 0.723 (0.712, 0.733) | 0.49 (0.418, 0.565) | 0.43 (0.355, 0.519) | 0.996 (0.993, 0.998) |
| | | no_hard | 0.733 (0.723, 0.742) | 0.468 (0.395, 0.535) | 0.597 (0.498, 0.689) | 0.996 (0.993, 0.998) |
| 0.08 | None | None | 0.759 (0.756, 0.762) | - | - | - |
| | baseline | baseline | 0.701 (0.686, 0.715) | 0.501 (0.428, 0.577) | 0.317 (0.232, 0.409) | 0.995 (0.993, 0.997) |
| | | no_hard | 0.709 (0.695, 0.724) | 0.463 (0.389, 0.532) | 0.343 (0.26, 0.434) | 0.995 (0.993, 0.997) |
| | easy_hard | baseline | 0.718 (0.709, 0.726) | 0.47 (0.376, 0.554) | 0.356 (0.262, 0.463) | 0.996 (0.994, 0.998) |
| | | no_hard | 0.729 (0.72, 0.738) | 0.48 (0.39, 0.558) | 0.351 (0.266, 0.441) | 0.996 (0.994, 0.997) |
| 0.1 | None | None | 0.756 (0.753, 0.76) | - | - | - |
| | baseline | baseline | 0.697 (0.68, 0.714) | 0.42 (0.329, 0.504) | 0.267 (0.19, 0.354) | 0.995 (0.993, 0.997) |
| | | no_hard | 0.703 (0.686, 0.721) | 0.455 (0.373, 0.534) | 0.296 (0.212, 0.385) | 0.995 (0.993, 0.997) |
| | easy_hard | baseline | 0.706 (0.697, 0.715) | 0.452 (0.387, 0.513) | 0.319 (0.238, 0.408) | 0.996 (0.994, 0.998) |
| | | no_hard | 0.713 (0.703, 0.722) | 0.42 (0.348, 0.487) | 0.326 (0.255, 0.406) | 0.996 (0.994, 0.998) |

Table 12: Additional measures of statistical fidelity for the main experiment, summarized across all datasets. All numbers have been rounded to 3 decimals. Inv. KL-D = inverse Kullback-Leibler Divergence, WD = Wasserstein Distance, MMD = Maximum Mean Discrepancy.

| Generative Model | Preprocessing Strategy | Postprocessing Strategy | Inv. KL-D | WD | MMD |
|---|---|---|---|---|---|
| bayesian_network | baseline | baseline | 0.998 (0.998, 0.999) | 0.041 (0.018, 0.067) | 0.001 (0.0, 0.003) |
| | | no_hard | 0.998 (0.998, 0.999) | 0.148 (0.034, 0.34) | 0.001 (0.0, 0.003) |
| | easy_hard | baseline | 0.998 (0.998, 0.999) | 0.04 (0.017, 0.068) | 0.001 (0.0, 0.003) |
| | | no_hard | 0.998 (0.998, 0.999) | 0.146 (0.035, 0.332) | 0.001 (0.0, 0.003) |
| ctgan | baseline | baseline | 0.979 (0.967, 0.987) | 0.022 (0.01, 0.037) | 0.001 (0.0, 0.002) |
| | | no_hard | 0.978 (0.967, 0.986) | 0.534 (0.041, 1.481) | 0.001 (0.0, 0.002) |
| | easy_hard | baseline | 0.979 (0.968, 0.986) | 0.022 (0.01, 0.036) | 0.001 (0.0, 0.002) |
| | | no_hard | 0.977 (0.967, 0.985) | 0.546 (0.039, 1.518) | 0.001 (0.0, 0.002) |
| ddpm | baseline | baseline | 0.846 (0.685, 0.979) | 0.717 (0.17, 1.409) | 0.0 (0.0, 0.001) |
| | | no_hard | 0.845 (0.688, 0.971) | 1.085 (0.326, 1.966) | 0.0 (0.0, 0.001) |
| | easy_hard | baseline | 0.849 (0.709, 0.967) | 0.744 (0.137, 1.42) | 0.0 (0.0, 0.001) |
| | | no_hard | 0.848 (0.714, 0.967) | 1.066 (0.363, 1.874) | 0.0 (0.0, 0.001) |
| nflow | baseline | baseline | 0.975 (0.968, 0.981) | 0.074 (0.038, 0.115) | 0.001 (0.0, 0.002) |
| | | no_hard | 0.975 (0.968, 0.981) | 0.519 (0.079, 1.351) | 0.001 (0.0, 0.002) |
| | easy_hard | baseline | 0.975 (0.967, 0.981) | 0.075 (0.04, 0.113) | 0.001 (0.0, 0.003) |
| | | no_hard | 0.975 (0.967, 0.982) | 0.546 (0.076, 1.439) | 0.001 (0.0, 0.003) |
| tvae | baseline | baseline | 0.966 (0.952, 0.977) | 0.026 (0.012, 0.041) | 0.0 (0.0, 0.001) |
| | | no_hard | 0.965 (0.952, 0.977) | 0.531 (0.046, 1.465) | 0.0 (0.0, 0.001) |
| | easy_hard | baseline | 0.968 (0.957, 0.978) | 0.026 (0.012, 0.041) | 0.0 (0.0, 0.001) |
| | | no_hard | 0.968 (0.957, 0.978) | 0.483 (0.047, 1.32) | 0.0 (0.0, 0.001) |

Table 13: Summarised performance across all datasets, with an additional row showing the performance of the real data with data-centric postprocessing.

| Generative Model | Preprocessing Strategy | Postprocessing Strategy | Classification | Feature Selection | Model Selection |
|---|---|---|---|---|---|
| None | None | None | 0.866 (0.856, 0.876) | - | - |
| | | no_hard | 0.859 (0.837, 0.88) | 0.937 (0.926, 0.946) | 0.697 (0.647, 0.747) |
| bayesian_network | baseline | baseline | 0.622 (0.588, 0.656) | 0.091 (-0.004, 0.188) | 0.155 (0.047, 0.263) |
| | | no_hard | 0.624 (0.59, 0.659) | 0.043 (-0.065, 0.143) | 0.112 (0.021, 0.207) |
| | easy_hard | baseline | 0.624 (0.592, 0.66) | 0.1 (-0.015, 0.213) | 0.297 (0.202, 0.387) |
| | | no_hard | 0.627 (0.594, 0.661) | 0.082 (-0.033, 0.195) | 0.197 (0.104, 0.291) |
| ctgan | baseline | baseline | 0.797 (0.771, 0.822) | 0.63 (0.568, 0.699) | 0.519 (0.457, 0.581) |
| | | no_hard | 0.807 (0.782, 0.829) | 0.653 (0.592, 0.714) | 0.578 (0.512, 0.648) |
| | easy_hard | baseline | 0.791 (0.763, 0.818) | 0.63 (0.566, 0.692) | 0.513 (0.441, 0.576) |
| | | no_hard | 0.8 (0.772, 0.828) | 0.631 (0.566, 0.691) | 0.594 (0.531, 0.655) |
| ddpm | baseline | baseline | 0.813 (0.784, 0.842) | 0.635 (0.552, 0.718) | 0.508 (0.442, 0.571) |
| | | no_hard | 0.818 (0.787, 0.849) | 0.645 (0.569, 0.726) | 0.476 (0.406, 0.543) |
| | easy_hard | baseline | 0.818 (0.788, 0.844) | 0.645 (0.564, 0.723) | 0.544 (0.476, 0.61) |
| | | no_hard | 0.824 (0.796, 0.85) | 0.661 (0.584, 0.738) | 0.539 (0.466, 0.604) |
| nflow | baseline | baseline | 0.737 (0.713, 0.761) | 0.415 (0.342, 0.486) | 0.348 (0.275, 0.415) |
| | | no_hard | 0.745 (0.718, 0.768) | 0.439 (0.372, 0.503) | 0.357 (0.292, 0.425) |
| | easy_hard | baseline | 0.742 (0.718, 0.768) | 0.455 (0.384, 0.527) | 0.362 (0.297, 0.428) |
| | | no_hard | 0.749 (0.725, 0.772) | 0.445 (0.365, 0.511) | 0.371 (0.31, 0.436) |
| tvae | baseline | baseline | 0.792 (0.766, 0.816) | 0.675 (0.623, 0.723) | 0.506 (0.438, 0.575) |
| | | no_hard | 0.8 (0.775, 0.828) | 0.675 (0.63, 0.721) | 0.488 (0.419, 0.561) |
| | easy_hard | baseline | 0.79 (0.763, 0.814) | 0.687 (0.642, 0.73) | 0.477 (0.416, 0.538) |
| | | no_hard | 0.797 (0.771, 0.822) | 0.699 (0.653, 0.742) | 0.507 (0.437, 0.578) |

## C.11 Postprocessing the real data

Tables 13 and 14 investigate the effect of postprocessing the original data (i.e. removing hard examples) on the main experimental benchmark data and on the datasets with added label noise, respectively. The performances for feature and model selection are obtained by comparing the feature and model rankings against the non-postprocessed version of the real data, in a similar manner as the performance of the generative models. The results indicate that postprocessing has a minor impact on classification: a slight negative effect on the purportedly clean benchmark datasets, and a slight positive effect on the datasets with added label noise. Spearman's Rank Correlation for feature selection remains high across both the main benchmark datasets (0.94) and the datasets with label noise (0.88), whereas the ranking of models is lower at approximately 0.7 for both types of data.

Table 14: Summarised performance across all datasets with added label noise, with an additional row showing the performance of the real data with data-centric postprocessing.

| Generative Model | Preprocessing Strategy | Postprocessing Strategy | Classification | Feature Selection | Model Selection |
|---|---|---|---|---|---|
| None | None | None | 0.766 (0.764, 0.767) | - | - |
| | | no_hard | 0.776 (0.773, 0.779) | 0.883 (0.868, 0.896) | 0.704 (0.662, 0.744) |
| bayesian_network | baseline | baseline | 0.69 (0.678, 0.703) | 0.393 (0.348, 0.439) | 0.376 (0.284, 0.47) |
| | | no_hard | 0.697 (0.685, 0.71) | 0.417 (0.371, 0.462) | 0.45 (0.359, 0.55) |
| | easy_hard | baseline | 0.742 (0.736, 0.747) | 0.535 (0.503, 0.565) | 0.44 (0.346, 0.54) |
| | | no_hard | 0.751 (0.745, 0.756) | 0.522 (0.485, 0.562) | 0.608 (0.533, 0.685) |
| ctgan | baseline | baseline | 0.741 (0.734, 0.746) | 0.662 (0.621, 0.7) | 0.451 (0.352, 0.565) |
| | | no_hard | 0.752 (0.747, 0.757) | 0.63 (0.6, 0.658) | 0.56 (0.463, 0.646) |
| | easy_hard | baseline | 0.734 (0.73, 0.738) | 0.62 (0.58, 0.665) | 0.366 (0.29, 0.443) |
| | | no_hard | 0.746 (0.74, 0.751) | 0.58 (0.539, 0.624) | 0.544 (0.46, 0.624) |
| ddpm | baseline | baseline | 0.685 (0.677, 0.693) | 0.272 (0.21, 0.333) | 0.294 (0.232, 0.357) |
| | | no_hard | 0.69 (0.682, 0.699) | 0.28 (0.203, 0.35) | 0.298 (0.228, 0.367) |
| | easy_hard | baseline | 0.72 (0.711, 0.73) | 0.43 (0.345, 0.518) | 0.309 (0.222, 0.404) |
| | | no_hard | 0.728 (0.717, 0.738) | 0.439 (0.365, 0.508) | 0.518 (0.417, 0.635) |
| nflow | baseline | baseline | 0.686 (0.675, 0.698) | 0.325 (0.266, 0.384) | 0.393 (0.328, 0.457) |
| | | no_hard | 0.699 (0.685, 0.711) | 0.344 (0.295, 0.396) | 0.467 (0.363, 0.56) |
| | easy_hard | baseline | 0.707 (0.699, 0.715) | 0.342 (0.279, 0.395) | 0.521 (0.449, 0.594) |
| | | no_hard | 0.717 (0.71, 0.725) | 0.324 (0.261, 0.385) | 0.577 (0.491, 0.667) |
| tvae | baseline | baseline | 0.735 (0.731, 0.739) | 0.64 (0.609, 0.673) | 0.41 (0.331, 0.494) |
| | | no_hard | 0.747 (0.743, 0.752) | 0.643 (0.608, 0.676) | 0.537 (0.459, 0.621) |
| | easy_hard | baseline | 0.722 (0.715, 0.729) | 0.608 (0.56, 0.66) | 0.437 (0.355, 0.523) |
| | | no_hard | 0.732 (0.725, 0.738) | 0.601 (0.546, 0.653) | 0.415 (0.325, 0.521) |

# D  Statistical tests

This appendix provides details on the statistical significance of the effect of data-centric processing on both the datasets in the main experimental benchmark and on the datasets with added noise. Subsection D.1 provides a summary of the insights gained from the tests, and subsections D.2 and D.3 provide details on the main experiment and label noise experiment, respectively.

## D.1  Summary

On the label noise experiments, the data-centric processing strategies have a beneficial effect across all tasks (see Table 16), with significant effects of pre- and postprocessing strategies across all tasks, except postprocessing for feature selection. The effect of data-centric processing is not significantly different across varying levels of label noise. There is a slight tendency towards an inverted V shape effect of the processing strategies, i.e., larger benefits with moderate levels of label noise, however, this is not significant.

For the main experimental benchmark data, data-centric processing led to significant improvements in performance for classification (postprocessing only), and model selection (preprocessing only), as shown in Table 15. The diminished effect of data-centric processing in the main experimental benchmark compared to the datasets with added label noise might be caused by the extensive preprocessing of the datasets in the benchmark as mentioned in Sec 4.1. The datasets could therefore be speculated to be too "clean" to receive substantive benefits from data-centric processing.

Measures of model fit ($R^2$) of the models in the main benchmark data indicate that the variance in task performance is best explained in the classification task ($R^2 = 0.98$), slightly worse in the feature selection task ($R^2 = 0.85$), and substantially lower in the model selection task ($R^2 = 0.57$). This might be due to the smoothness of the performance metrics. The model ranking score is established by accurately ranking the eight supervised classification models listed in Appendix B.3. Similarly, the feature selection score is determined by correctly ranking the metrics for feature importance across all features within the dataset – a quantity that naturally varies by dataset. In contrast, classification performance relies on AUROC values computed on the test dataset, which encompasses a relatively large number of samples (1,141 in the smallest dataset). Due to the dissimilarity in the number of samples, models, and features, the metrics for model and feature ranking are inevitably more susceptible to larger variance. This occurs because disparities in a single element can lead to more

Table 15: Main effect of the pre- and postprocessing strategies of the main experiment, controlling for the interaction between dataset and generative model. The estimates represent the difference in the corresponding performance metric (AUROC or Spearman's Rank Correlation depending on the task), compared to the baseline strategy, i.e. no processing.

| Task | Processing | Estimate | Std. Error | p-value | Significant |
|------|-----------|----------|-----------|---------|-------------|
| Classification | preprocessing_strategy[T.easy_hard] | 0.001 | 0.001 | 0.526 | 0 |
| | postprocessing_strategy[T.no_hard] | 0.007 | 0.001 | <0.000 | 1 |
| Feature Selection | preprocessing_strategy[T.easy_hard] | 0.013 | 0.008 | 0.091 | 0 |
| | postprocessing_strategy[T.no_hard] | 0.003 | 0.008 | 0.706 | 0 |
| Model Selection | preprocessing_strategy[T.easy_hard] | 0.031 | 0.012 | 0.008 | 1 |
| | postprocessing_strategy[T.no_hard] | 0.003 | 0.012 | 0.822 | 0 |

substantial effects on the performance metrics in comparison to the impact on classification. In summary, this means that the performance metrics for feature and model selection are more sensitive to minor deviations or random noise than classification.

## D.2  Main experiment

Three linear models were constructed to statistically assess the effect of the data-centric processing strategies on the performance of each task (classification, feature selection, model selection). The models controlled for the main effects of the datasets and generative models, as well as the interaction between the two, as the different datasets might have characteristics that make them more or less difficult for specific generative models. The models took the following formula:

$$metric \sim dataset\_id * generative\_model + preprocessing\_strategy + postprocessing\_strategy$$

Where *metric* represents the performance metric corresponding to the task, i.e., AUROC for classification or Spearman's Rank Correlation for feature and model selection.

Table 15 shows the main effect of the processing strategies for each of the tasks. Tables 17, 18, and 19 show the full output of the models.

## D.3  Label noise

Similarly to the main experiment, three linear models were constructed to statistically assess the effect of the data-centric processing strategies on each task's performance with different label noise levels (see Figure 4). The models controlled for the main effects of level of label noise and generative model, and took the following formula:

$$metric \sim prop\_label\_noise + generative\_model + preprocessing\_strategy + postprocessing\_strategy$$

Where *metric* represents the performance metric corresponding to the task, i.e., AUROC for classification or Spearman's Rank Correlation for feature and model selection.

Table 16 shows the main effect of the processing strategies for each of the tasks. Tables 20, 21, and 22 show the full output of the models.

To assess whether the effectiveness of the data-centric processing strategies is modulated by the level of label noise in the data, we constructed a linear model similar to the one above, but with an interaction effect between the proportion of label noise and the data-centric processing strategies. The models took the following form:

$$metric \sim generative\_model + prop\_label\_noise * (preprocessing\_strategy + postprocessing\_strategy)$$

The output of these models is shown in Tables 23, 24, and 25.

Notably, the model with interaction effects had slightly higher Akaike's Information Criteria than the model without, indicating a worse fit to the data.

Table 16: Main effect of the pre- and postprocessing strategies of the label noise experiment, controlling for the effect of the proportion of label noise and generative model. The estimates represent the difference in the corresponding performance metric (AUROC or Spearman's Rank Correlation depending on the task), compared to the baseline strategy, i.e. no processing.

| Task | Processing | Estimate | Std. Error | p-value | Significant |
|---|---|---|---|---|---|
| Classification | preprocessing_strategy[T.easy_hard] | 0.017 | 0.002 | <0.000 | 1 |
| | postprocessing_strategy[T.no_hard] | 0.010 | 0.002 | <0.000 | 1 |
| Feature Selection | preprocessing_strategy[T.easy_hard] | 0.039 | 0.013 | 0.002 | 1 |
| | postprocessing_strategy[T.no_hard] | -0.005 | 0.013 | 0.710 | 0 |
| Model Selection | preprocessing_strategy[T.easy_hard] | 0.050 | 0.019 | 0.008 | 1 |
| | postprocessing_strategy[T.no_hard] | 0.098 | 0.019 | <0.000 | 1 |

Table 17: Full output of the linear model assessing the impact of data-centric processing on the classification task. The model has the following formula: auroc ~ dataset_id * generative_model + preprocessing_strategy + postprocessing_strategy.

| | coef | std err | t | P>|t| | [0.025 | 0.975] |
|---|---|---|---|---|---|---|
| Intercept | 0.7902 | 0.004 | 204.799 | 0.000 | 0.783 | 0.798 |
| dataset_id[T.361060] | -0.1217 | 0.005 | -22.730 | 0.000 | -0.132 | -0.111 |
| dataset_id[T.361062] | 0.1000 | 0.005 | 18.669 | 0.000 | 0.089 | 0.110 |
| dataset_id[T.361063] | -0.2918 | 0.005 | -54.508 | 0.000 | -0.302 | -0.281 |
| dataset_id[T.361065] | -0.2965 | 0.005 | -55.379 | 0.000 | -0.307 | -0.286 |
| dataset_id[T.361066] | -0.2066 | 0.005 | -38.587 | 0.000 | -0.217 | -0.196 |
| dataset_id[T.361070] | -0.2510 | 0.005 | -46.876 | 0.000 | -0.261 | -0.240 |
| dataset_id[T.361273] | -0.1725 | 0.011 | -15.192 | 0.000 | -0.195 | -0.150 |
| dataset_id[T.361275] | -0.0790 | 0.002 | -36.264 | 0.000 | -0.083 | -0.075 |
| dataset_id[T.361277] | -0.2882 | 0.005 | -53.822 | 0.000 | -0.299 | -0.278 |
| dataset_id[T.361278] | -0.0492 | 0.002 | -22.787 | 0.000 | -0.053 | -0.045 |
| generative_model[T.ctgan] | 0.0228 | 0.006 | 4.107 | 0.000 | 0.012 | 0.034 |
| generative_model[T.ddpm] | 0.0488 | 0.005 | 9.109 | 0.000 | 0.038 | 0.059 |
| generative_model[T.nflow] | 0.0254 | 0.005 | 4.746 | 0.000 | 0.015 | 0.036 |
| generative_model[T.tvae] | -0.0109 | 0.005 | -2.034 | 0.042 | -0.021 | -0.000 |
| preprocessing_strategy[T.easy_hard] | 0.0007 | 0.001 | 0.635 | 0.526 | -0.001 | 0.003 |
| postprocessing_strategy[T.no_hard] | 0.0067 | 0.001 | 6.393 | 0.000 | 0.005 | 0.009 |
| dataset_id[T.361060]:generative_model[T.ctgan] | 0.1514 | 0.008 | 19.314 | 0.000 | 0.136 | 0.167 |
| dataset_id[T.361062]:generative_model[T.ctgan] | 0.0334 | 0.008 | 4.340 | 0.000 | 0.018 | 0.049 |
| dataset_id[T.361063]:generative_model[T.ctgan] | 0.3876 | 0.008 | 50.293 | 0.000 | 0.372 | 0.403 |
| dataset_id[T.361065]:generative_model[T.ctgan] | 0.3626 | 0.008 | 47.056 | 0.000 | 0.347 | 0.378 |
| dataset_id[T.361066]:generative_model[T.ctgan] | 0.2351 | 0.008 | 30.504 | 0.000 | 0.220 | 0.250 |
| dataset_id[T.361070]:generative_model[T.ctgan] | -0.0246 | 0.008 | -3.191 | 0.001 | -0.040 | -0.009 |
| dataset_id[T.361273]:generative_model[T.ctgan] | -0.0336 | 0.013 | -2.658 | 0.008 | -0.058 | -0.009 |
| dataset_id[T.361275]:generative_model[T.ctgan] | -0.0248 | 0.005 | -5.147 | 0.000 | -0.034 | -0.015 |
| dataset_id[T.361277]:generative_model[T.ctgan] | 0.3828 | 0.008 | 49.671 | 0.000 | 0.368 | 0.398 |
| dataset_id[T.361278]:generative_model[T.ctgan] | -0.0010 | 0.005 | -0.204 | 0.838 | -0.010 | 0.008 |
| dataset_id[T.361060]:generative_model[T.ddpm] | 0.1686 | 0.008 | 22.261 | 0.000 | 0.154 | 0.183 |
| dataset_id[T.361062]:generative_model[T.ddpm] | -0.0044 | 0.008 | -0.583 | 0.560 | -0.019 | 0.010 |
| dataset_id[T.361063]:generative_model[T.ddpm] | 0.3924 | 0.008 | 51.823 | 0.000 | 0.378 | 0.407 |
| dataset_id[T.361065]:generative_model[T.ddpm] | 0.3751 | 0.008 | 49.535 | 0.000 | 0.360 | 0.390 |
| dataset_id[T.361066]:generative_model[T.ddpm] | 0.2268 | 0.008 | 29.955 | 0.000 | 0.212 | 0.242 |
| dataset_id[T.361070]:generative_model[T.ddpm] | -0.0828 | 0.008 | -10.929 | 0.000 | -0.098 | -0.068 |
| dataset_id[T.361273]:generative_model[T.ddpm] | -0.0342 | 0.013 | -2.721 | 0.007 | -0.059 | -0.010 |
| dataset_id[T.361275]:generative_model[T.ddpm] | -0.0297 | 0.005 | -6.172 | 0.000 | -0.039 | -0.020 |
| dataset_id[T.361277]:generative_model[T.ddpm] | 0.3912 | 0.008 | 51.667 | 0.000 | 0.376 | 0.406 |
| dataset_id[T.361278]:generative_model[T.ddpm] | -0.0266 | 0.005 | -5.681 | 0.000 | -0.036 | -0.017 |
| dataset_id[T.361060]:generative_model[T.nflow] | 0.1307 | 0.008 | 17.264 | 0.000 | 0.116 | 0.146 |
| dataset_id[T.361062]:generative_model[T.nflow] | -0.2662 | 0.008 | -35.153 | 0.000 | -0.281 | -0.251 |
| dataset_id[T.361063]:generative_model[T.nflow] | 0.3527 | 0.008 | 46.577 | 0.000 | 0.338 | 0.368 |
| dataset_id[T.361065]:generative_model[T.nflow] | 0.2659 | 0.008 | 35.114 | 0.000 | 0.251 | 0.281 |
| dataset_id[T.361066]:generative_model[T.nflow] | 0.1658 | 0.008 | 21.896 | 0.000 | 0.151 | 0.181 |
| dataset_id[T.361070]:generative_model[T.nflow] | -0.0587 | 0.008 | -7.755 | 0.000 | -0.074 | -0.044 |
| dataset_id[T.361273]:generative_model[T.nflow] | -0.0340 | 0.013 | -2.706 | 0.007 | -0.059 | -0.009 |
| dataset_id[T.361275]:generative_model[T.nflow] | -0.0341 | 0.005 | -7.274 | 0.000 | -0.043 | -0.025 |
| dataset_id[T.361277]:generative_model[T.nflow] | 0.3517 | 0.008 | 46.451 | 0.000 | 0.337 | 0.367 |
| dataset_id[T.361278]:generative_model[T.nflow] | -0.0500 | 0.005 | -10.700 | 0.000 | -0.059 | -0.041 |
| dataset_id[T.361060]:generative_model[T.tvae] | 0.1793 | 0.008 | 23.684 | 0.000 | 0.164 | 0.194 |
| dataset_id[T.361062]:generative_model[T.tvae] | 0.0874 | 0.008 | 11.540 | 0.000 | 0.073 | 0.102 |
| dataset_id[T.361063]:generative_model[T.tvae] | 0.4303 | 0.008 | 56.827 | 0.000 | 0.415 | 0.445 |
| dataset_id[T.361065]:generative_model[T.tvae] | 0.3816 | 0.008 | 50.397 | 0.000 | 0.367 | 0.396 |
| dataset_id[T.361066]:generative_model[T.tvae] | 0.2467 | 0.008 | 32.578 | 0.000 | 0.232 | 0.262 |
| dataset_id[T.361070]:generative_model[T.tvae] | 0.0319 | 0.008 | 4.210 | 0.000 | 0.017 | 0.047 |
| dataset_id[T.361273]:generative_model[T.tvae] | -0.0214 | 0.013 | -1.701 | 0.089 | -0.046 | 0.003 |
| dataset_id[T.361275]:generative_model[T.tvae] | 0.0095 | 0.005 | 2.036 | 0.042 | 0.000 | 0.019 |
| dataset_id[T.361277]:generative_model[T.tvae] | 0.4105 | 0.008 | 54.211 | 0.000 | 0.396 | 0.425 |
| dataset_id[T.361278]:generative_model[T.tvae] | 0.0283 | 0.005 | 6.062 | 0.000 | 0.019 | 0.038 |

| Dep. Variable: | auroc | R-squared: | 0.979 |
|---|---|---|---|
| Model: | OLS | Adj. R-squared: | 0.978 |
| Method: | Least Squares | F-statistic: | 1363. |
| Date: | Thu, 10 Aug 2023 | Prob (F-statistic): | 0.00 |
| Time: | 13:47:12 | Log-Likelihood: | 4043.2 |
| No. Observations: | 1656 | AIC: | -7976. |
| Df Residuals: | 1601 | BIC: | -7679. |
| Df Model: | 54 | | |
| Covariance Type: | nonrobust | | |

| Omnibus: | 368.744 | Durbin-Watson: | 1.231 |
|---|---|---|---|
| Prob(Omnibus): | 0.000 | Jarque-Bera (JB): | 5945.102 |
| Skew: | -0.586 | Prob(JB): | 0.00 |
| Kurtosis: | 12.208 | Cond. No. | 3.28e+15 |

Table 18: Full output of the linear model assessing the impact of data-centric processing on the feature selection task. The model has the following formula: spearmans_r ∼ dataset_id ∗ generative_model + preprocessing_strategy + postprocessing_strategy.

| | coef | std err | t | P>|t| | [0.025 | 0.975] |
|---|---|---|---|---|---|---|
| Intercept | 0.7391 | 0.028 | 26.413 | 0.000 | 0.684 | 0.794 |
| dataset_id[T.361060] | -1.1264 | 0.039 | -29.011 | 0.000 | -1.203 | -1.050 |
| dataset_id[T.361062] | -0.3920 | 0.039 | -10.095 | 0.000 | -0.468 | -0.316 |
| dataset_id[T.361063] | -0.7461 | 0.039 | -19.217 | 0.000 | -0.822 | -0.670 |
| dataset_id[T.361065] | -0.7883 | 0.039 | -20.301 | 0.000 | -0.864 | -0.712 |
| dataset_id[T.361066] | -1.1443 | 0.039 | -29.471 | 0.000 | -1.220 | -1.068 |
| dataset_id[T.361070] | -0.6526 | 0.039 | -16.806 | 0.000 | -0.729 | -0.576 |
| dataset_id[T.361273] | 0.1191 | 0.082 | 1.446 | 0.148 | -0.042 | 0.281 |
| dataset_id[T.361275] | -0.2040 | 0.016 | -12.903 | 0.000 | -0.235 | -0.173 |
| dataset_id[T.361277] | -0.5915 | 0.039 | -15.233 | 0.000 | -0.668 | -0.515 |
| dataset_id[T.361278] | -0.2893 | 0.016 | -18.462 | 0.000 | -0.320 | -0.259 |
| generative_model[T.ctgan] | 0.1076 | 0.040 | 2.677 | 0.008 | 0.029 | 0.186 |
| generative_model[T.ddpm] | 0.1549 | 0.039 | 3.990 | 0.000 | 0.079 | 0.231 |
| generative_model[T.nflow] | 0.0326 | 0.039 | 0.839 | 0.402 | -0.044 | 0.109 |
| generative_model[T.tvae] | -0.0742 | 0.039 | -1.912 | 0.056 | -0.150 | 0.002 |
| preprocessing_strategy[T.easy_hard] | 0.0129 | 0.008 | 1.693 | 0.091 | -0.002 | 0.028 |
| postprocessing_strategy[T.no_hard] | 0.0029 | 0.008 | 0.377 | 0.706 | -0.012 | 0.018 |
| dataset_id[T.361060]:generative_model[T.ctgan] | 0.9020 | 0.057 | 15.869 | 0.000 | 0.791 | 1.013 |
| dataset_id[T.361062]:generative_model[T.ctgan] | -0.0035 | 0.056 | -0.062 | 0.950 | -0.113 | 0.106 |
| dataset_id[T.361063]:generative_model[T.ctgan] | 0.6861 | 0.056 | 12.277 | 0.000 | 0.576 | 0.796 |
| dataset_id[T.361065]:generative_model[T.ctgan] | 0.8420 | 0.056 | 15.068 | 0.000 | 0.732 | 0.952 |
| dataset_id[T.361066]:generative_model[T.ctgan] | 0.6424 | 0.056 | 11.496 | 0.000 | 0.533 | 0.752 |
| dataset_id[T.361070]:generative_model[T.ctgan] | -0.2157 | 0.056 | -3.859 | 0.000 | -0.325 | -0.106 |
| dataset_id[T.361273]:generative_model[T.ctgan] | -0.0585 | 0.092 | -0.638 | 0.524 | -0.238 | 0.121 |
| dataset_id[T.361275]:generative_model[T.ctgan] | -0.0154 | 0.035 | -0.441 | 0.659 | -0.084 | 0.053 |
| dataset_id[T.361277]:generative_model[T.ctgan] | 0.6625 | 0.056 | 11.855 | 0.000 | 0.553 | 0.772 |
| dataset_id[T.361278]:generative_model[T.ctgan] | -0.0062 | 0.035 | -0.178 | 0.859 | -0.075 | 0.062 |
| dataset_id[T.361060]:generative_model[T.ddpm] | 1.1565 | 0.055 | 21.061 | 0.000 | 1.049 | 1.264 |
| dataset_id[T.361062]:generative_model[T.ddpm] | 0.0169 | 0.055 | 0.308 | 0.758 | -0.091 | 0.125 |
| dataset_id[T.361063]:generative_model[T.ddpm] | 0.7857 | 0.055 | 14.309 | 0.000 | 0.678 | 0.893 |
| dataset_id[T.361065]:generative_model[T.ddpm] | 0.8345 | 0.055 | 15.197 | 0.000 | 0.727 | 0.942 |
| dataset_id[T.361066]:generative_model[T.ddpm] | 0.3339 | 0.055 | 6.081 | 0.000 | 0.226 | 0.442 |
| dataset_id[T.361070]:generative_model[T.ddpm] | -0.2618 | 0.055 | -4.768 | 0.000 | -0.370 | -0.154 |
| dataset_id[T.361273]:generative_model[T.ddpm] | -0.0601 | 0.091 | -0.660 | 0.510 | -0.239 | 0.119 |
| dataset_id[T.361275]:generative_model[T.ddpm] | -0.2600 | 0.035 | -7.448 | 0.000 | -0.329 | -0.192 |
| dataset_id[T.361277]:generative_model[T.ddpm] | 0.6687 | 0.055 | 12.179 | 0.000 | 0.561 | 0.776 |
| dataset_id[T.361278]:generative_model[T.ddpm] | -0.2341 | 0.034 | -6.902 | 0.000 | -0.301 | -0.168 |
| dataset_id[T.361060]:generative_model[T.nflow] | 0.7911 | 0.055 | 14.407 | 0.000 | 0.683 | 0.899 |
| dataset_id[T.361062]:generative_model[T.nflow] | -0.3605 | 0.055 | -6.564 | 0.000 | -0.468 | -0.253 |
| dataset_id[T.361063]:generative_model[T.nflow] | 0.5161 | 0.055 | 9.400 | 0.000 | 0.408 | 0.624 |
| dataset_id[T.361065]:generative_model[T.nflow] | 0.5458 | 0.055 | 9.940 | 0.000 | 0.438 | 0.654 |
| dataset_id[T.361066]:generative_model[T.nflow] | 0.6583 | 0.055 | 11.988 | 0.000 | 0.551 | 0.766 |
| dataset_id[T.361070]:generative_model[T.nflow] | -0.2076 | 0.055 | -3.780 | 0.000 | -0.315 | -0.100 |
| dataset_id[T.361273]:generative_model[T.nflow] | -0.1531 | 0.091 | -1.681 | 0.093 | -0.332 | 0.025 |
| dataset_id[T.361275]:generative_model[T.nflow] | -0.0690 | 0.034 | -2.031 | 0.042 | -0.136 | -0.002 |
| dataset_id[T.361277]:generative_model[T.nflow] | 0.6371 | 0.055 | 11.602 | 0.000 | 0.529 | 0.745 |
| dataset_id[T.361278]:generative_model[T.nflow] | -0.2948 | 0.034 | -8.694 | 0.000 | -0.361 | -0.228 |
| dataset_id[T.361060]:generative_model[T.tvae] | 1.1245 | 0.055 | 20.478 | 0.000 | 1.017 | 1.232 |
| dataset_id[T.361062]:generative_model[T.tvae] | 0.1828 | 0.055 | 3.329 | 0.001 | 0.075 | 0.291 |
| dataset_id[T.361063]:generative_model[T.tvae] | 0.9820 | 0.055 | 17.883 | 0.000 | 0.874 | 1.090 |
| dataset_id[T.361065]:generative_model[T.tvae] | 0.9466 | 0.055 | 17.239 | 0.000 | 0.839 | 1.054 |
| dataset_id[T.361066]:generative_model[T.tvae] | 1.0318 | 0.055 | 18.791 | 0.000 | 0.924 | 1.140 |
| dataset_id[T.361070]:generative_model[T.tvae] | 0.3116 | 0.055 | 5.675 | 0.000 | 0.204 | 0.419 |
| dataset_id[T.361273]:generative_model[T.tvae] | 0.1300 | 0.091 | 1.428 | 0.153 | -0.049 | 0.309 |
| dataset_id[T.361275]:generative_model[T.tvae] | 0.1405 | 0.034 | 4.135 | 0.000 | 0.074 | 0.207 |
| dataset_id[T.361277]:generative_model[T.tvae] | 0.8421 | 0.055 | 15.336 | 0.000 | 0.734 | 0.950 |
| dataset_id[T.361278]:generative_model[T.tvae] | 0.2458 | 0.034 | 7.249 | 0.000 | 0.179 | 0.312 |

| Dep. Variable: | spearmans_r | R-squared: | 0.850 | | Omnibus: | 172.268 | Durbin-Watson: | 1.460 |
|---|---|---|---|---|---|---|---|---|
| Model: | OLS | Adj. R-squared: | 0.845 | | Prob(Omnibus): | 0.000 | Jarque-Bera (JB): | 953.150 |
| Method: | Least Squares | F-statistic: | 168.6 | | Skew: | -0.312 | Prob(JB): | 1.06e-207 |
| Date: | Thu, 10 Aug 2023 | Prob (F-statistic): | 0.00 | | Kurtosis: | 6.664 | Cond. No. | 3.28e+15 |
| Time: | 13:47:43 | Log-Likelihood: | 762.21 | | | | | |
| No. Observations: | 1656 | AIC: | -1414. | | | | | |
| Df Residuals: | 1601 | BIC: | -1117. | | | | | |
| Df Model: | 54 | | | | | | | |
| Covariance Type: | nonrobust | | | | | | | |

Table 19: Full output of the linear model assessing the impact of data-centric processing on the model selection task. The model has the following formula: spearmans_r ∼ dataset_id ∗ generative_model + preprocessing_strategy + postprocessing_strategy.

| | coef | std err | t | P>|t| | [0.025 | 0.975] |
|---|---|---|---|---|---|---|
| Intercept | 0.7711 | 0.043 | 17.939 | 0.000 | 0.687 | 0.855 |
| dataset_id[T.361060] | -0.5060 | 0.060 | -8.482 | 0.000 | -0.623 | -0.389 |
| dataset_id[T.361062] | -1.0089 | 0.060 | -16.914 | 0.000 | -1.126 | -0.892 |
| dataset_id[T.361063] | -0.7679 | 0.060 | -12.873 | 0.000 | -0.885 | -0.651 |
| dataset_id[T.361065] | -0.7976 | 0.060 | -13.372 | 0.000 | -0.915 | -0.681 |
| dataset_id[T.361066] | -0.4405 | 0.060 | -7.384 | 0.000 | -0.557 | -0.323 |
| dataset_id[T.361070] | -0.5446 | 0.060 | -9.131 | 0.000 | -0.662 | -0.428 |
| dataset_id[T.361273] | -0.5558 | 0.127 | -4.392 | 0.000 | -0.804 | -0.308 |
| dataset_id[T.361275] | 0.0239 | 0.024 | 0.983 | 0.326 | -0.024 | 0.072 |
| dataset_id[T.361277] | -0.7232 | 0.060 | -12.124 | 0.000 | -0.840 | -0.606 |
| dataset_id[T.361278] | 0.1115 | 0.024 | 4.633 | 0.000 | 0.064 | 0.159 |
| generative_model[T.ctgan] | -0.0643 | 0.062 | -1.042 | 0.298 | -0.185 | 0.057 |
| generative_model[T.ddpm] | -0.4769 | 0.060 | -7.996 | 0.000 | -0.594 | -0.360 |
| generative_model[T.nflow] | -0.4650 | 0.060 | -7.796 | 0.000 | -0.582 | -0.348 |
| generative_model[T.tvae] | -0.3363 | 0.060 | -5.638 | 0.000 | -0.453 | -0.219 |
| preprocessing_strategy[T.easy_hard] | 0.0310 | 0.012 | 2.643 | 0.008 | 0.008 | 0.054 |
| postprocessing_strategy[T.no_hard] | 0.0026 | 0.012 | 0.226 | 0.822 | -0.020 | 0.026 |
| dataset_id[T.361060]:generative_model[T.ctgan] | 0.0425 | 0.087 | 0.487 | 0.626 | -0.129 | 0.214 |
| dataset_id[T.361062]:generative_model[T.ctgan] | 0.6551 | 0.086 | 7.630 | 0.000 | 0.487 | 0.823 |
| dataset_id[T.361063]:generative_model[T.ctgan] | 0.4750 | 0.086 | 5.533 | 0.000 | 0.307 | 0.643 |
| dataset_id[T.361065]:generative_model[T.ctgan] | 1.0323 | 0.086 | 12.024 | 0.000 | 0.864 | 1.201 |
| dataset_id[T.361066]:generative_model[T.ctgan] | 0.3582 | 0.086 | 4.172 | 0.000 | 0.190 | 0.527 |
| dataset_id[T.361070]:generative_model[T.ctgan] | 0.1097 | 0.086 | 1.278 | 0.202 | -0.059 | 0.278 |
| dataset_id[T.361273]:generative_model[T.ctgan] | 0.2659 | 0.141 | 1.889 | 0.059 | -0.010 | 0.542 |
| dataset_id[T.361275]:generative_model[T.ctgan] | -0.2788 | 0.054 | -5.197 | 0.000 | -0.384 | -0.174 |
| dataset_id[T.361277]:generative_model[T.ctgan] | 0.8121 | 0.086 | 9.459 | 0.000 | 0.644 | 0.980 |
| dataset_id[T.361278]:generative_model[T.ctgan] | -0.1759 | 0.054 | -3.286 | 0.001 | -0.281 | -0.071 |
| dataset_id[T.361060]:generative_model[T.ddpm] | 0.5670 | 0.084 | 6.721 | 0.000 | 0.402 | 0.732 |
| dataset_id[T.361062]:generative_model[T.ddpm] | 0.9628 | 0.084 | 11.413 | 0.000 | 0.797 | 1.128 |
| dataset_id[T.361063]:generative_model[T.ddpm] | 1.1213 | 0.084 | 13.292 | 0.000 | 0.956 | 1.287 |
| dataset_id[T.361065]:generative_model[T.ddpm] | 1.4635 | 0.084 | 17.349 | 0.000 | 1.298 | 1.629 |
| dataset_id[T.361066]:generative_model[T.ddpm] | 0.7180 | 0.084 | 8.511 | 0.000 | 0.553 | 0.883 |
| dataset_id[T.361070]:generative_model[T.ddpm] | 0.5104 | 0.084 | 6.051 | 0.000 | 0.345 | 0.676 |
| dataset_id[T.361273]:generative_model[T.ddpm] | 0.6815 | 0.140 | 4.872 | 0.000 | 0.407 | 0.956 |
| dataset_id[T.361275]:generative_model[T.ddpm] | 0.0758 | 0.054 | 1.414 | 0.158 | -0.029 | 0.181 |
| dataset_id[T.361277]:generative_model[T.ddpm] | 1.3073 | 0.084 | 15.497 | 0.000 | 1.142 | 1.473 |
| dataset_id[T.361278]:generative_model[T.ddpm] | 0.0499 | 0.052 | 0.958 | 0.338 | -0.052 | 0.152 |
| dataset_id[T.361060]:generative_model[T.nflow] | 0.4754 | 0.084 | 5.636 | 0.000 | 0.310 | 0.641 |
| dataset_id[T.361062]:generative_model[T.nflow] | 0.6741 | 0.084 | 7.991 | 0.000 | 0.509 | 0.840 |
| dataset_id[T.361063]:generative_model[T.nflow] | 0.8802 | 0.084 | 10.434 | 0.000 | 0.715 | 1.046 |
| dataset_id[T.361065]:generative_model[T.nflow] | 1.0074 | 0.084 | 11.942 | 0.000 | 0.842 | 1.173 |
| dataset_id[T.361066]:generative_model[T.nflow] | 0.4762 | 0.084 | 5.645 | 0.000 | 0.311 | 0.642 |
| dataset_id[T.361070]:generative_model[T.nflow] | 0.2515 | 0.084 | 2.981 | 0.003 | 0.086 | 0.417 |
| dataset_id[T.361273]:generative_model[T.nflow] | 0.7135 | 0.140 | 5.101 | 0.000 | 0.439 | 0.988 |
| dataset_id[T.361275]:generative_model[T.nflow] | 0.1063 | 0.052 | 2.037 | 0.042 | 0.004 | 0.209 |
| dataset_id[T.361277]:generative_model[T.nflow] | 0.9487 | 0.084 | 11.246 | 0.000 | 0.783 | 1.114 |
| dataset_id[T.361278]:generative_model[T.nflow] | 0.0760 | 0.052 | 1.458 | 0.145 | -0.026 | 0.178 |
| dataset_id[T.361060]:generative_model[T.tvae] | 0.4487 | 0.084 | 5.319 | 0.000 | 0.283 | 0.614 |
| dataset_id[T.361062]:generative_model[T.tvae] | 0.4695 | 0.084 | 5.566 | 0.000 | 0.304 | 0.635 |
| dataset_id[T.361063]:generative_model[T.tvae] | 0.6555 | 0.084 | 7.771 | 0.000 | 0.490 | 0.821 |
| dataset_id[T.361065]:generative_model[T.tvae] | 1.1830 | 0.084 | 14.024 | 0.000 | 1.018 | 1.348 |
| dataset_id[T.361066]:generative_model[T.tvae] | 0.6868 | 0.084 | 8.141 | 0.000 | 0.521 | 0.852 |
| dataset_id[T.361070]:generative_model[T.tvae] | 0.2746 | 0.084 | 3.255 | 0.001 | 0.109 | 0.440 |
| dataset_id[T.361273]:generative_model[T.tvae] | 0.5952 | 0.140 | 4.255 | 0.000 | 0.321 | 0.870 |
| dataset_id[T.361275]:generative_model[T.tvae] | 0.1205 | 0.052 | 2.308 | 0.021 | 0.018 | 0.223 |
| dataset_id[T.361277]:generative_model[T.tvae] | 1.0848 | 0.084 | 12.860 | 0.000 | 0.919 | 1.250 |
| dataset_id[T.361278]:generative_model[T.tvae] | 0.1615 | 0.052 | 3.101 | 0.002 | 0.059 | 0.264 |

| Dep. Variable: | spearmans_r | R-squared: | 0.566 | | Omnibus: | 15.550 | Durbin-Watson: | 1.779 |
|---|---|---|---|---|---|---|---|---|
| Model: | OLS | Adj. R-squared: | 0.551 | | Prob(Omnibus): | 0.000 | Jarque-Bera (JB): | 23.156 |
| Method: | Least Squares | F-statistic: | 38.60 | | Skew: | 0.055 | Prob(JB): | 9.37e-06 |
| Date: | Thu, 10 Aug 2023 | Prob (F-statistic): | 6.00e-248 | | Kurtosis: | 3.569 | Cond. No. | 3.28e+15 |
| Time: | 13:47:27 | Log-Likelihood: | 51.201 | | | | | |
| No. Observations: | 1656 | AIC: | 7.598 | | | | | |
| Df Residuals: | 1601 | BIC: | 305.3 | | | | | |
| Df Model: | 54 | | | | | | | |
| Covariance Type: | nonrobust | | | | | | | |

Table 20: Full output of the linear model assessing the impact of data-centric processing on the classification task on the label noise data. The model has the following formula: auroc ~ perc_label_noise + generative_model + preprocessing_strategy + postprocessing_strategy.

| | | | |
|---|---|---|---|
| Dep. Variable: | mean | R-squared: | 0.450 |
| Model: | OLS | Adj. R-squared: | 0.440 |
| Method: | Least Squares | F-statistic: | 43.48 |
| Date: | Fri, 11 Aug 2023 | Prob (F-statistic): | 9.47e-69 |
| Time: | 09:19:44 | Log-Likelihood: | 1376.0 |
| No. Observations: | 596 | AIC: | -2728. |
| Df Residuals: | 584 | BIC: | -2675. |
| Df Model: | 11 | | |
| Covariance Type: | nonrobust | | |

| | coef | std err | t | P>|t| | [0.025 | 0.975] |
|---|---|---|---|---|---|---|
| Intercept | 0.7239 | 0.003 | 209.577 | 0.000 | 0.717 | 0.731 |
| perc_label_noise[T.2] | -0.0090 | 0.003 | -2.630 | 0.009 | -0.016 | -0.002 |
| perc_label_noise[T.4] | -0.0174 | 0.003 | -5.050 | 0.000 | -0.024 | -0.011 |
| perc_label_noise[T.6] | -0.0223 | 0.003 | -6.482 | 0.000 | -0.029 | -0.016 |
| perc_label_noise[T.8] | -0.0246 | 0.003 | -7.163 | 0.000 | -0.031 | -0.018 |
| perc_label_noise[T.10] | -0.0343 | 0.003 | -9.877 | 0.000 | -0.041 | -0.027 |
| synthetic_model_type[T.ctgan] | 0.0238 | 0.003 | 7.520 | 0.000 | 0.018 | 0.030 |
| synthetic_model_type[T.ddpm] | -0.0139 | 0.003 | -4.398 | 0.000 | -0.020 | -0.008 |
| synthetic_model_type[T.nflow] | -0.0173 | 0.003 | -5.463 | 0.000 | -0.024 | -0.011 |
| synthetic_model_type[T.tvae] | 0.0145 | 0.003 | 4.576 | 0.000 | 0.008 | 0.021 |
| preprocessing_strategy[T.easy_hard] | 0.0175 | 0.002 | 8.782 | 0.000 | 0.014 | 0.021 |
| postprocessing_strategy[T.no_hard] | 0.0096 | 0.002 | 4.834 | 0.000 | 0.006 | 0.014 |

| | | | |
|---|---|---|---|
| Omnibus: | 9.482 | Durbin-Watson: | 1.202 |
| Prob(Omnibus): | 0.009 | Jarque-Bera (JB): | 6.880 |
| Skew: | -0.144 | Prob(JB): | 0.0321 |
| Kurtosis: | 2.559 | Cond. No. | 8.98 |

Table 21: Full output of the linear model assessing the impact of data-centric processing on the feature selection task on the label noise data. The model has the following formula: spearmans_r ~ perc_label_noise + generative_model + preprocessing_strategy + postprocessing_strategy.

| | | | |
|---|---|---|---|
| Dep. Variable: | spearmans_r | R-squared: | 0.424 |
| Model: | OLS | Adj. R-squared: | 0.413 |
| Method: | Least Squares | F-statistic: | 39.10 |
| Date: | Fri, 11 Aug 2023 | Prob (F-statistic): | 5.47e-63 |
| Time: | 09:20:03 | Log-Likelihood: | 279.90 |
| No. Observations: | 596 | AIC: | -535.8 |
| Df Residuals: | 584 | BIC: | -483.1 |
| Df Model: | 11 | | |
| Covariance Type: | nonrobust | | |

| | coef | std err | t | P>|t| | [0.025 | 0.975] |
|---|---|---|---|---|---|---|
| Intercept | 0.4821 | 0.022 | 22.187 | 0.000 | 0.439 | 0.525 |
| perc_label_noise[T.2] | -0.0002 | 0.022 | -0.008 | 0.993 | -0.043 | 0.042 |
| perc_label_noise[T.4] | -0.0383 | 0.022 | -1.773 | 0.077 | -0.081 | 0.004 |
| perc_label_noise[T.6] | -0.0514 | 0.022 | -2.378 | 0.018 | -0.094 | -0.009 |
| perc_label_noise[T.8] | -0.0355 | 0.022 | -1.643 | 0.101 | -0.078 | 0.007 |
| perc_label_noise[T.10] | -0.0777 | 0.022 | -3.555 | 0.000 | -0.121 | -0.035 |
| synthetic_model_type[T.ctgan] | 0.1575 | 0.020 | 7.912 | 0.000 | 0.118 | 0.197 |
| synthetic_model_type[T.ddpm] | -0.1104 | 0.020 | -5.548 | 0.000 | -0.150 | -0.071 |
| synthetic_model_type[T.nflow] | -0.1318 | 0.020 | -6.623 | 0.000 | -0.171 | -0.093 |
| synthetic_model_type[T.tvae] | 0.1575 | 0.020 | 7.914 | 0.000 | 0.118 | 0.197 |
| preprocessing_strategy[T.easy_hard] | 0.0391 | 0.013 | 3.121 | 0.002 | 0.014 | 0.064 |
| postprocessing_strategy[T.no_hard] | -0.0047 | 0.013 | -0.372 | 0.710 | -0.029 | 0.020 |

| | | | |
|---|---|---|---|
| Omnibus: | 5.130 | Durbin-Watson: | 1.277 |
| Prob(Omnibus): | 0.077 | Jarque-Bera (JB): | 5.145 |
| Skew: | -0.172 | Prob(JB): | 0.0764 |
| Kurtosis: | 3.299 | Cond. No. | 8.98 |

Table 22: Full output of the linear model assessing the impact of data-centric processing on the model selection task on the label noise data. The model has the following formula: spearmans_r ~ perc_label_noise + generative_model + preprocessing_strategy + postprocessing_strategy.

| | | | |
|---|---|---|---|
| Dep. Variable: | spearmans_r | R-squared: | 0.236 |
| Model: | OLS | Adj. R-squared: | 0.221 |
| Method: | Least Squares | F-statistic: | 16.37 |
| Date: | Fri, 11 Aug 2023 | Prob (F-statistic): | 3.17e-28 |
| Time: | 09:19:53 | Log-Likelihood: | 45.755 |
| No. Observations: | 596 | AIC: | -67.51 |
| Df Residuals: | 584 | BIC: | -14.83 |
| Df Model: | 11 | | |
| Covariance Type: | nonrobust | | |

| | coef | std err | t | P>|t| | [0.025 | 0.975] |
|---|---|---|---|---|---|---|
| Intercept | 0.5466 | 0.032 | 16.983 | 0.000 | 0.483 | 0.610 |
| perc_label_noise[T.2] | -0.1033 | 0.032 | -3.228 | 0.001 | -0.166 | -0.040 |
| perc_label_noise[T.4] | -0.1564 | 0.032 | -4.886 | 0.000 | -0.219 | -0.094 |
| perc_label_noise[T.6] | -0.1162 | 0.032 | -3.629 | 0.000 | -0.179 | -0.053 |
| perc_label_noise[T.8] | -0.2621 | 0.032 | -8.188 | 0.000 | -0.325 | -0.199 |
| perc_label_noise[T.10] | -0.3013 | 0.032 | -9.312 | 0.000 | -0.365 | -0.238 |
| synthetic_model_type[T.ctgan] | 0.0166 | 0.029 | 0.561 | 0.575 | -0.041 | 0.074 |
| synthetic_model_type[T.ddpm] | -0.1090 | 0.029 | -3.698 | 0.000 | -0.167 | -0.051 |
| synthetic_model_type[T.nflow] | 0.0257 | 0.029 | 0.871 | 0.384 | -0.032 | 0.084 |
| synthetic_model_type[T.tvae] | -0.0138 | 0.029 | -0.468 | 0.640 | -0.072 | 0.044 |
| preprocessing_strategy[T.easy_hard] | 0.0496 | 0.019 | 2.675 | 0.008 | 0.013 | 0.086 |
| postprocessing_strategy[T.no_hard] | 0.0976 | 0.019 | 5.260 | 0.000 | 0.061 | 0.134 |

| | | | |
|---|---|---|---|
| Omnibus: | 1.523 | Durbin-Watson: | 1.686 |
| Prob(Omnibus): | 0.467 | Jarque-Bera (JB): | 1.515 |
| Skew: | 0.061 | Prob(JB): | 0.469 |
| Kurtosis: | 2.785 | Cond. No. | 8.98 |

Table 23: Full output of the linear model assessing the impact of data-centric processing on the classification task on the label noise data with interaction effects. The model has the following formula: auroc ~ generative_model + perc_label_noise * (preprocessing_strategy + postprocessing_strategy).

| | | | |
|---|---|---|---|
| Dep. Variable: | auroc | R-squared: | 0.458 |
| Model: | OLS | Adj. R-squared: | 0.433 |
| Method: | Least Squares | F-statistic: | 17.81 |
| Date: | Thu, 10 Aug 2023 | Prob (F-statistic): | 5.01e-59 |
| Time: | 14:32:02 | Log-Likelihood: | 1380.5 |
| No. Observations: | 596 | AIC: | -2705. |
| Df Residuals: | 568 | BIC: | -2582. |
| Df Model: | 27 | | |
| Covariance Type: | nonrobust | | |

| | coef | std err | t | P>|t| | [0.025 | 0.975] |
|---|---|---|---|---|---|---|
| Intercept | 0.7261 | 0.005 | 137.188 | 0.000 | 0.716 | 0.736 |
| generative_model[T.ctgan] | 0.0238 | 0.003 | 7.472 | 0.000 | 0.018 | 0.030 |
| generative_model[T.ddpm] | -0.0139 | 0.003 | -4.371 | 0.000 | -0.020 | -0.008 |
| generative_model[T.nflow] | -0.0173 | 0.003 | -5.428 | 0.000 | -0.024 | -0.011 |
| generative_model[T.tvae] | 0.0145 | 0.003 | 4.547 | 0.000 | 0.008 | 0.021 |
| perc_label_noise[T.2] | -0.0129 | 0.007 | -1.868 | 0.062 | -0.027 | 0.001 |
| perc_label_noise[T.4] | -0.0233 | 0.007 | -3.364 | 0.001 | -0.037 | -0.010 |
| perc_label_noise[T.6] | -0.0272 | 0.007 | -3.933 | 0.000 | -0.041 | -0.014 |
| perc_label_noise[T.8] | -0.0266 | 0.007 | -3.843 | 0.000 | -0.040 | -0.013 |
| perc_label_noise[T.10] | -0.0305 | 0.007 | -4.370 | 0.000 | -0.044 | -0.017 |
| pre_post[T.baseline - no_hard] | 0.0116 | 0.007 | 1.683 | 0.093 | -0.002 | 0.025 |
| pre_post[T.easy_hard - baseline] | 0.0128 | 0.007 | 1.848 | 0.065 | -0.001 | 0.026 |
| pre_post[T.easy_hard - no_hard] | 0.0211 | 0.007 | 3.051 | 0.002 | 0.008 | 0.035 |
| perc_label_noise[T.2]:pre_post[T.baseline - no_hard] | 0.0007 | 0.010 | 0.070 | 0.944 | -0.019 | 0.020 |
| perc_label_noise[T.4]:pre_post[T.baseline - no_hard] | -0.0016 | 0.010 | -0.167 | 0.868 | -0.021 | 0.018 |
| perc_label_noise[T.6]:pre_post[T.baseline - no_hard] | -0.0017 | 0.010 | -0.172 | 0.863 | -0.021 | 0.018 |
| perc_label_noise[T.8]:pre_post[T.baseline - no_hard] | -0.0033 | 0.010 | -0.333 | 0.739 | -0.022 | 0.016 |
| perc_label_noise[T.10]:pre_post[T.baseline - no_hard] | -0.0061 | 0.010 | -0.619 | 0.536 | -0.026 | 0.013 |
| perc_label_noise[T.2]:pre_post[T.easy_hard - baseline] | 0.0047 | 0.010 | 0.485 | 0.628 | -0.014 | 0.024 |
| perc_label_noise[T.4]:pre_post[T.easy_hard - baseline] | 0.0122 | 0.010 | 1.246 | 0.213 | -0.007 | 0.031 |
| perc_label_noise[T.6]:pre_post[T.easy_hard - baseline] | 0.0103 | 0.010 | 1.050 | 0.294 | -0.009 | 0.029 |
| perc_label_noise[T.8]:pre_post[T.easy_hard - baseline] | 0.0044 | 0.010 | 0.449 | 0.654 | -0.015 | 0.024 |
| perc_label_noise[T.10]:pre_post[T.easy_hard - baseline] | -0.0034 | 0.010 | -0.349 | 0.727 | -0.023 | 0.016 |
| perc_label_noise[T.2]:pre_post[T.easy_hard - no_hard] | 0.0101 | 0.010 | 1.034 | 0.302 | -0.009 | 0.029 |
| perc_label_noise[T.4]:pre_post[T.easy_hard - no_hard] | 0.0131 | 0.010 | 1.338 | 0.182 | -0.006 | 0.032 |
| perc_label_noise[T.6]:pre_post[T.easy_hard - no_hard] | 0.0111 | 0.010 | 1.137 | 0.256 | -0.008 | 0.030 |
| perc_label_noise[T.8]:pre_post[T.easy_hard - no_hard] | 0.0067 | 0.010 | 0.688 | 0.492 | -0.012 | 0.026 |
| perc_label_noise[T.10]:pre_post[T.easy_hard - no_hard] | -0.0055 | 0.010 | -0.557 | 0.578 | -0.025 | 0.014 |

| | | | |
|---|---|---|---|
| Omnibus: | 8.499 | Durbin-Watson: | 1.185 |
| Prob(Omnibus): | 0.014 | Jarque-Bera (JB): | 6.567 |
| Skew: | -0.155 | Prob(JB): | 0.0375 |
| Kurtosis: | 2.590 | Cond. No. | 34.7 |

Table 24: Full output of the linear model assessing the impact of data-centric processing on the feature selection task on the label noise data with interaction effects. The model has the following formula: spearmans_r ∼ generative_model + perc_label_noise ∗ (preprocessing_strategy + postprocessing_strategy).

| | coef | std err | t | P>|t| | [0.025 | 0.975] |
|---|---|---|---|---|---|---|
| Intercept | 0.4679 | 0.033 | 14.099 | 0.000 | 0.403 | 0.533 |
| generative_model[T.ctgan] | 0.1575 | 0.020 | 7.887 | 0.000 | 0.118 | 0.197 |
| generative_model[T.ddpm] | -0.1104 | 0.020 | -5.530 | 0.000 | -0.150 | -0.071 |
| generative_model[T.nflow] | -0.1318 | 0.020 | -6.602 | 0.000 | -0.171 | -0.093 |
| generative_model[T.tvae] | 0.1575 | 0.020 | 7.889 | 0.000 | 0.118 | 0.197 |
| perc_label_noise[T.2] | -0.0417 | 0.043 | -0.961 | 0.337 | -0.127 | 0.044 |
| perc_label_noise[T.4] | -0.0252 | 0.043 | -0.581 | 0.561 | -0.110 | 0.060 |
| perc_label_noise[T.6] | -0.0337 | 0.043 | -0.777 | 0.437 | -0.119 | 0.051 |
| perc_label_noise[T.8] | 0.0183 | 0.043 | 0.422 | 0.673 | -0.067 | 0.103 |
| perc_label_noise[T.10] | -0.0631 | 0.044 | -1.441 | 0.150 | -0.149 | 0.023 |
| pre_post[T.baseline - no_hard] | -0.0016 | 0.043 | -0.036 | 0.971 | -0.087 | 0.084 |
| pre_post[T.easy_hard - baseline] | 0.0633 | 0.043 | 1.459 | 0.145 | -0.022 | 0.148 |
| pre_post[T.easy_hard - no_hard] | 0.0637 | 0.043 | 1.469 | 0.142 | -0.021 | 0.149 |
| perc_label_noise[T.2]:pre_post[T.baseline - no_hard] | 0.0498 | 0.061 | 0.811 | 0.417 | -0.071 | 0.170 |
| perc_label_noise[T.4]:pre_post[T.baseline - no_hard] | -0.0089 | 0.061 | -0.145 | 0.885 | -0.129 | 0.112 |
| perc_label_noise[T.6]:pre_post[T.baseline - no_hard] | -0.0042 | 0.061 | -0.068 | 0.946 | -0.125 | 0.116 |
| perc_label_noise[T.8]:pre_post[T.baseline - no_hard] | -0.0362 | 0.061 | -0.590 | 0.556 | -0.157 | 0.084 |
| perc_label_noise[T.10]:pre_post[T.baseline - no_hard] | 0.0369 | 0.062 | 0.595 | 0.552 | -0.085 | 0.159 |
| perc_label_noise[T.2]:pre_post[T.easy_hard - baseline] | 0.0641 | 0.061 | 1.044 | 0.297 | -0.056 | 0.185 |
| perc_label_noise[T.4]:pre_post[T.easy_hard - baseline] | -0.0078 | 0.061 | -0.127 | 0.899 | -0.128 | 0.113 |
| perc_label_noise[T.6]:pre_post[T.easy_hard - baseline] | -0.0216 | 0.061 | -0.352 | 0.725 | -0.142 | 0.099 |
| perc_label_noise[T.8]:pre_post[T.easy_hard - baseline] | -0.0944 | 0.061 | -1.539 | 0.124 | -0.215 | 0.026 |
| perc_label_noise[T.10]:pre_post[T.easy_hard - baseline] | -0.0312 | 0.062 | -0.504 | 0.614 | -0.153 | 0.090 |
| perc_label_noise[T.2]:pre_post[T.easy_hard - no_hard] | 0.0521 | 0.061 | 0.850 | 0.396 | -0.068 | 0.173 |
| perc_label_noise[T.4]:pre_post[T.easy_hard - no_hard] | -0.0358 | 0.061 | -0.584 | 0.559 | -0.156 | 0.085 |
| perc_label_noise[T.6]:pre_post[T.easy_hard - no_hard] | -0.0449 | 0.061 | -0.733 | 0.464 | -0.165 | 0.076 |
| perc_label_noise[T.8]:pre_post[T.easy_hard - no_hard] | -0.0846 | 0.061 | -1.380 | 0.168 | -0.205 | 0.036 |
| perc_label_noise[T.10]:pre_post[T.easy_hard - no_hard] | -0.0637 | 0.062 | -1.029 | 0.304 | -0.185 | 0.058 |

| | | | |
|---|---|---|---|
| Dep. Variable: | spearmans_r | R-squared: | 0.436 |
| Model: | OLS | Adj. R-squared: | 0.410 |
| Method: | Least Squares | F-statistic: | 16.28 |
| Date: | Thu, 10 Aug 2023 | Prob (F-statistic): | 2.37e-54 |
| Time: | 14:32:20 | Log-Likelihood: | 286.27 |
| No. Observations: | 596 | AIC: | -516.5 |
| Df Residuals: | 568 | BIC: | -393.6 |
| Df Model: | 27 | | |
| Covariance Type: | nonrobust | | |

| | | | |
|---|---|---|---|
| Omnibus: | 4.136 | Durbin-Watson: | 1.256 |
| Prob(Omnibus): | 0.126 | Jarque-Bera (JB): | 4.063 |
| Skew: | -0.151 | Prob(JB): | 0.131 |
| Kurtosis: | 3.268 | Cond. No. | 34.7 |

Table 25: Full output of the linear model assessing the impact of data-centric processing on the model selection task on the label noise data with interaction effects. The model has the following formula: spearmans_r ∼ generative_model + perc_label_noise ∗ (preprocessing_strategy + postprocessing_strategy).

| | coef | std err | t | P>|t| | [0.025 | 0.975] |
|---|---|---|---|---|---|---|
| Intercept | 0.4914 | 0.049 | 10.055 | 0.000 | 0.395 | 0.587 |
| generative_model[T.ctgan] | 0.0166 | 0.029 | 0.563 | 0.574 | -0.041 | 0.074 |
| generative_model[T.ddpm] | -0.1090 | 0.029 | -3.709 | 0.000 | -0.167 | -0.051 |
| generative_model[T.nflow] | 0.0257 | 0.029 | 0.873 | 0.383 | -0.032 | 0.083 |
| generative_model[T.tvae] | -0.0138 | 0.029 | -0.469 | 0.639 | -0.072 | 0.044 |
| perc_label_noise[T.2] | -0.0162 | 0.064 | -0.254 | 0.800 | -0.142 | 0.109 |
| perc_label_noise[T.4] | -0.0781 | 0.064 | -1.223 | 0.222 | -0.204 | 0.047 |
| perc_label_noise[T.6] | -0.0867 | 0.064 | -1.357 | 0.175 | -0.212 | 0.039 |
| perc_label_noise[T.8] | -0.1581 | 0.064 | -2.476 | 0.014 | -0.284 | -0.033 |
| perc_label_noise[T.10] | -0.2077 | 0.065 | -3.219 | 0.001 | -0.334 | -0.081 |
| pre_post[T.baseline - no_hard] | 0.1819 | 0.064 | 2.849 | 0.005 | 0.056 | 0.307 |
| pre_post[T.easy_hard - baseline] | 0.1057 | 0.064 | 1.656 | 0.098 | -0.020 | 0.231 |
| pre_post[T.easy_hard - no_hard] | 0.2276 | 0.064 | 3.565 | 0.000 | 0.102 | 0.353 |
| perc_label_noise[T.2]:pre_post[T.baseline - no_hard] | -0.1381 | 0.090 | -1.529 | 0.127 | -0.315 | 0.039 |
| perc_label_noise[T.4]:pre_post[T.baseline - no_hard] | -0.1467 | 0.090 | -1.624 | 0.105 | -0.324 | 0.031 |
| perc_label_noise[T.6]:pre_post[T.baseline - no_hard] | -0.0343 | 0.090 | -0.380 | 0.704 | -0.212 | 0.143 |
| perc_label_noise[T.8]:pre_post[T.baseline - no_hard] | -0.1562 | 0.090 | -1.730 | 0.084 | -0.334 | 0.021 |
| perc_label_noise[T.10]:pre_post[T.baseline - no_hard] | -0.1531 | 0.091 | -1.678 | 0.094 | -0.332 | 0.026 |
| perc_label_noise[T.2]:pre_post[T.easy_hard - baseline] | -0.1305 | 0.090 | -1.445 | 0.149 | -0.308 | 0.047 |
| perc_label_noise[T.4]:pre_post[T.easy_hard - baseline] | -0.1410 | 0.090 | -1.561 | 0.119 | -0.318 | 0.036 |
| perc_label_noise[T.6]:pre_post[T.easy_hard - baseline] | -0.0648 | 0.090 | -0.717 | 0.474 | -0.242 | 0.113 |
| perc_label_noise[T.8]:pre_post[T.easy_hard - baseline] | -0.0667 | 0.090 | -0.738 | 0.461 | -0.244 | 0.111 |
| perc_label_noise[T.10]:pre_post[T.easy_hard - baseline] | -0.0531 | 0.091 | -0.582 | 0.561 | -0.232 | 0.126 |
| perc_label_noise[T.2]:pre_post[T.easy_hard - no_hard] | -0.0800 | 0.090 | -0.886 | 0.376 | -0.257 | 0.097 |
| perc_label_noise[T.4]:pre_post[T.easy_hard - no_hard] | -0.0257 | 0.090 | -0.285 | 0.776 | -0.203 | 0.152 |
| perc_label_noise[T.6]:pre_post[T.easy_hard - no_hard] | -0.0190 | 0.090 | -0.211 | 0.833 | -0.196 | 0.158 |
| perc_label_noise[T.8]:pre_post[T.easy_hard - no_hard] | -0.1933 | 0.090 | -2.141 | 0.033 | -0.371 | -0.016 |
| perc_label_noise[T.10]:pre_post[T.easy_hard - no_hard] | -0.1681 | 0.091 | -1.842 | 0.066 | -0.347 | 0.011 |

| | | | |
|---|---|---|---|
| Dep. Variable: | spearmans_r | R-squared: | 0.261 |
| Model: | OLS | Adj. R-squared: | 0.226 |
| Method: | Least Squares | F-statistic: | 7.422 |
| Date: | Thu, 10 Aug 2023 | Prob (F-statistic): | 1.12e-23 |
| Time: | 14:32:11 | Log-Likelihood: | 55.698 |
| No. Observations: | 596 | AIC: | -55.40 |
| Df Residuals: | 568 | BIC: | 67.53 |
| Df Model: | 27 | | |
| Covariance Type: | nonrobust | | |

| | | | |
|---|---|---|---|
| Omnibus: | 1.138 | Durbin-Watson: | 1.634 |
| Prob(Omnibus): | 0.566 | Jarque-Bera (JB): | 1.224 |
| Skew: | 0.077 | Prob(JB): | 0.542 |
| Kurtosis: | 2.840 | Cond. No. | 34.7 |

