# OpenReview forum: "Reimagining Synthetic Tabular Data Generation through Data-Centric AI: A Comprehensive Benchmark"
_NeurIPS.cc/2023/Track/Datasets_and_Benchmarks — NeurIPS 2023 Datasets and Benchmarks Poster_

### Official Review · Reviewer_g5vc · 2023-07-21
**Interesting Topic but Significance Needs to be Improved**

**Rating:** 4
**Confidence:** 4
**Correctness:** yes
**Clarity:** yes

**Strengths:**

1. The motivation of the paper is good, as it addresses a crucial and practical challenge in the field of synthetic data generation by proposing the integration of data-centric AI techniques.

2. The code/data are available and will potentially benefit the community of synthetic data generation.



**Additional Feedback:**

See above.

**Documentation:**

Not enough.

It would be better if the authors could provide a high-level structure of the code to facilitate understanding and navigation for readers and potential users. For example, from the current code, it is not clear where are the implementations for Cleanlab, Data-IQ, and Data Maps.

I tried to install the code but find the following commands fail `git clone https://github.com/HLasse/data-centric-synth`. Does the author's implementation based on this specific repo?

**Limitations:**

yes

**Opportunities For Improvement:**

1. Overclaim. The paper solely conducts experiments on tabular datasets, but tries to generalize the findings to other data types and domains without proper validation. This lack of diversification in the experimental evaluation reduces the paper's credibility.

2. The paper's lack of extensive evaluation across various datasets and domains severely limits its generalizability and raises doubts about the robustness of the proposed framework. From the current paper, I cannot see a strong reason for why only evaluating *tabular* datasets, given there are many more approaches that focus on synthetic data generation for CV/NLP domains.  Without a more comprehensive assessment, it remains unclear how well the approach would perform on different data distributions or task types, undermining the validity and practicality of the paper's contributions.


3. The experiment analysis presented in the paper is disappointingly superficial, lacking any in-depth analysis to substantiate the claims made. The absence of detailed insights and thorough examination of the obtained outcomes raises concerns about the validity and reliability of the findings and leaves the reader questioning the significance and impact of the proposed approach. Further rigorous analysis and interpretation of the results are essential to enhance the credibility and value of the paper's contributions.

**Relation To Prior Work:**

Almost clear.  It would be better to add more works on CV/NLP domains if the author give a general name for this paper.

Borisov, V., Seßler, K., Leemann, T., Pawelczyk, M., & Kasneci, G. (2022). Language models are realistic tabular data generators. arXiv preprint arXiv:2210.06280.



**Summary And Contributions:**

This paper explores the integration of data-centric AI techniques to improve the quality of synthetic data for training machine learning models when real-world data is limited. It proposes a novel framework to evaluate the integration of data profiles and provides practical recommendations for enhancing synthetic data generation. This paper also conducts an evaluation of various tabular datasets and offers critical insights into the strengths and limitations of current synthetic data generation techniques.

---

> ### Author Response · Authors · 2023-08-12
>
> Thank you for your thoughtful comments and suggestions! Please find our answers to each of your points below:
>
> * (A) Generalizability
> * (B) In-depth analysis
> * (C) Documentation
>
> ## (A) Generalizability
> We have strived to incorporate tabular data from a wide range of domains (finance, eye-tracking, medicine, etc.) to ensure the generalizability of the findings. We chose to focus our efforts on the domain domain for a number of reasons as also outlined in the response to reviewer 1:
>
> 1. Data Diversity: Tabular data is ubiquitous across a wide range of domains, including finance, healthcare, e-commerce, and more. By focusing on tabular data we are able to evaluate the data-centric methods across many different domains with highly varied applications.
> 2. Scalability: By focusing on tabular data, we are able to provide a more comprehensive benchmark in terms of the number of seeds and datasets due to the additional computational resources required to train and apply models for image or text generation/classification.
> 3. Likely generalization: The methods for data-centric processing we have used in this paper are all directly applicable to other modalities, due to their only requirement being a trained classification model. As the mechanism of the data-centric methods is to identify wrongly labelled samples, we would expect the removal of these to have a similar effect regardless of the modality of the data. Additionally, we metrics we are using are not specifically related to tabular data but are applicable across all domains.
>
> We agree that this limitation is not adequately reflected in the title of the paper. In accordance, we have changed the title to be: “Reimagining Synthetic Tabular Data Generation through Data-Centric AI: A Comprehensive Benchmark”. Additionally, we have moved the Limitations section to the main manuscript and included a section on other modalities.
>
> ## (B) In-depth analysis
> By analysing 11 different datasets across several domains of tabular data, as well as assessing datasets with added label noise,  we have aimed to cover sufficient ground for our results to generalize to, at least, the tabular modality. We agree that a more in-depth analysis of the results would strengthen the findings. We now provide a comprehensive analysis of the distributions of the findings, along with statistical tests of the effect of the data-centric processing strategies in the newly added Appendix C.9 and Appendix D. Further, we have added a section showing the effect of postprocessing the real data in Appendix C.11. Please see below for an overview of the additional analyses and sections (also posted as a reply to reviewer 1).
>
> ### (B.1) Main benchmark datasets
>
> #### (B.1.1) Variability
> One reason for the large confidence intervals reported Table 2 is the fact that we are averaging across 11 different datasets. To decompose the variability, we have 1) plotted the distribution of scores across all the datasets and generative models in Appendix C.9.1, and 2) modelled the effect of the pre/postprocessing strategies using linear models that control for the effect of the datasets and generative models (see A.1.2 in this response and Appendix D).
>
> The main takeaways from Appendix C.9 are that the difficulty of the datasets is highly variable and that the variability of model performance regardless of pre/postprocessing strategy seems to be related to model performance. Notably, the models demonstrating strong performance, such as TabDDPM, exhibit comparatively lower variability across different seeds. In contrast, models with poorer performance, like Bayesian networks and normalizing flows, display higher variability across seeds.
>
> The performance metrics exhibit higher variance for the tasks of feature selection and model ranking compared to classification. This discrepancy can likely be attributed to the inherent smoothness of the performance metrics. The model ranking score is established by accurately ranking the eight supervised classification models listed in Appendix B.3. Similarly, the feature selection score is determined by correctly ranking the metrics for feature importance across all features within the dataset – a quantity that naturally varies by dataset.
>
> In contrast, classification performance relies on AUROC values computed on the test datasets, which encompasses a relatively large number of samples (1,141 in the smallest dataset). Due to the dissimilarity in the number of samples, models, and features, the metrics for model and feature ranking are inevitably more susceptible to larger variance. This occurs because disparities in a single element can lead to more substantial effects on the performance metrics in comparison to the impact on classification.

---

> > ### Author Response · Authors · 2023-08-12
> >
> > #### (B.1.2) Significance
> > To statistically assess the effect of the pre- and postprocessing strategies, we constructed three linear models (one for each task: classification, feature selection, model selection). The models were designed to control for both dataset and generative model effects while accounting for potential interaction effects between datasets and generative models, as the different datasets might have characteristics that make them more or less difficult for specific generative models. The models had the following formula:
> >
> > metric ~ dataset * generative_model + preprocessing_strategy + postprocessing_strategy
> >
> > Where “metric” represents the performance metric corresponding to the task, i.e., AUROC for classification or Spearman's Rank Correlation for feature and model selection.
> >
> > Appendix D shows the full output of the linear models along with a summary and interpretation. As expected, the variability between datasets, and between interactions between datasets and generative models, is quite large. However, of most interest, is the effect of the pre/processing strategies which is shown in Table 15 in the Appendix. Notably, the only two significant effects of the processing strategies are postprocessing for the classification task, and preprocessing for the model selection task, although their estimates are rather small.
> >
> >
> > When contrasted with the results from the label noise experiments (see A.2.2 in the response below or Appendix D), this might suggest that the datasets are too “clean” to reap benefits from the data-centric processing. As mentioned in the paper, the data used in the main experiment come from the “Tabular benchmark numerical classification” benchmark suite from [1]. To facilitate ease of modelling, the benchmark datasets have been processed heavily, with e.g. removal of categorical features and missing values, and balancing of the classes. This preprocessing could be suspected to remove a large proportion of the noise in the datasets and thereby mitigate the effects of further processing with data-centric methods.
> >
> > [1] Grinsztajn, L., Oyallon, E., & Varoquaux, G. (2022). Why do tree-based models still outperform deep learning on typical tabular data?. Advances in Neural Information Processing Systems, 35, 507-520.
> >
> > ### (B.2) Label noise
> >
> > #### (B.2.1) Variability
> > To investigate the variability in the label noise experiments, we have added two additional tables in Appendix C.9.2 for the label noise experiment and created the same visualizations of the distribution of scores in Appendix C.9.1 for the label noise experiment in Appendix C.9.2. The tables show 1) the same as Table 1, but aggregated across levels of label noise instead of datasets, 2) performance by level of label noise summarized over all generative models. The table that does not summarize across either generative model or dataset id takes up 126 rows, and is therefore not included. Instead, we refer to section A.2.2 in this response or Appendix D for an analysing of the interaction between levels of label noise and the effect of data-centric processing..
> >
> > #### (B.2.2) Significance
> > To statistically assess the effect of the pre- and postprocessing strategies with different amounts of label noise, we constructed three linear models (one for each task: classification, feature selection, model selection). The models were designed to control for label noise and generative model effects, and had the following formula:
> >
> > metric ~ prop_label_noise + generative_model + preprocessing_strategy + postprocessing_strategy
> >
> > Where “metric” represents the performance metric corresponding to the task, i.e., AUROC for classification or Spearman's Rank Correlation for feature and model selection.
> >
> > The main effect of pre- and postprocessing is shown in Appendix Table 16. The effect of the processing strategies is significant in all cases except postprocessing for feature selection.
> >
> > To assess whether the effectiveness of the pre- and postprocessing strategies is modulated by the level of label noise in the data, we constructed a linear model equivalent to the one described above, but with an interaction effect between the proportion of label noise and the pre- and postprocessing strategies. Akaike’s Information Criteria indicated that the simple model (without interactions) was a better fit, and only a few of the interaction effects were significant, although there was a slight tendency towards an inverted V shape effect of the processing strategies, i.e. larger benefit with moderate levels of label noise. The output of these models is reported in Tables 23-25.

---

> > > ### Author Response · Authors · 2023-08-12
> > >
> > > ## (C) Documentation
> > > Thank you for bringing the issue in the documentation to our attention. We have corrected the README to refer to the correct repository. Additionally, we have added an overview of the tree structure of the repository in the README with explanations of the content of each folder and links to implementations of Data-IQ and Data Maps. We have used the Python package `cleanlab` for the Cleanlab implementation.
> > >
> > > Unfortunately, the anonymous repository that we’re using does not allow for cloning of the repository. Instead, you can download a zipped version of the repository from the following link: https://we.tl/t-dGfVYiYKV6
> > >
> > > Thank you for notifying us of the issue.

---

> ### Author Response · Authors · 2023-08-28
> **Author Follow-Up**
>
> Dear Reviewer g5vc
>
> We are grateful for your time invested into the review process and appreciate your suggestions which have helped us to improve the paper!  Given the limited time left in the response phase, we wanted to check whether our responses and updated paper have addressed your comments. We are still eager to do our utmost to address any additional comments.
>
> Thank you!
>
> Paper 351 Authors

---

### Official Review · Reviewer_P9Bf · 2023-07-22
**Interesting experiments integrating data quality metrics into synthetic data generation and quality evaluation**

**Rating:** 7
**Confidence:** 4

**Strengths:**

These results will be useful for tasks requiring data augmentation, e.g., applications in which large annotated datasets are difficult to obtain, e.g., cybersecurity challenges.



**Additional Feedback:**

The paper is missing a good discussion on the techniques for partitioning data instances into easy/hard/ambiguous categories.


**Clarity:**

The paper is clear and readable without significant effort.

The abbreviations used in Figure 1 should have been introduced earlier; they are found a page later.


**Correctness:**

The claims seem to be correct.


**Documentation:**

I believe this aspect is satisfactory.


**Ethics:**

I do not have ethical concerns with this framework


**Limitations:**

I think the limitations should be in the Main Paper, not the Appendix.


**Opportunities For Improvement:**

I think the inverse KL-divergence may not be the best statistical fidelity measure to use. But it helps the author(s) make their point.
How do you set the threshold for determining the easy or hard instances in Cleanlab? Is there a sound theoretical basis for selecting the threshold? I did glance at the empirical approach in Appendix C.3



**Relation To Prior Work:**

I think the paper should compare with relevant work from the literature such as this one: https://dl.acm.org/doi/pdf/10.1145/3603709

**Summary And Contributions:**

This research provides fresh insights into integrating data-centric AI techniques into synthetic data
 generation. A framework is proposed to evaluate the integration of data profiles for creating
 more representative synthetic data. It is shown that statistical fidelity (e.g., inverse KL-divergence) alone is insufficient for assessing synthetic data’s utility, as it may overlook important nuances impacting downstream tasks. Incorporating data-centric methods consistently improves the utility of synthetic data across varying levels of label noise.

---

> ### Author Response · Authors · 2023-08-12
>
> We appreciate your time and effort in reviewing our paper and thank you for the encouraging review.
>
> ### Limitations
> We agree that the limitations section would be better suited to the main paper as its integral for interpreting the results. We have moved it to the end end of Results and expanded it with a mention of other data modalities (images, text).
>
> ### Statistical Fidelity
> You are right that other measures of statistical fidelity can be useful to report for comparison. We have added Table 12 to Appendix C.10, which also shows Wasserstein distance and Maximum Mean Discrepancy across all conditions.
>
> ### Determining the data-centric threshold
> Determining the correct data-centric threshold is crucial for obtaining the right balance between false positives and false negatives. As you mention, our procedure for determining the threshold for the data-centric methods is described in Appendix C.3. The procedure and theoretical basis for determining the label quality score for Cleanlab is quite extensive and is thoroughly described in [Northcutt et al., 2021]( https://jair.org/index.php/jair/article/view/12125/26676), section 3.1. We have added a link to the appropriate section of the Cleanlab paper to Appendix C.3 to guide interested readers.

---

### Official Review · Reviewer_yA6T · 2023-07-23
**Review for "Reimagining Synthetic Data Generation through Data-Centric AI: A Comprehensive Benchmark"**

**Rating:** 7
**Confidence:** 3

**Strengths:**

* Very strong motivation for an important issue that is of broad importance. Evaluating the quality of generated data is crucial and an active research direction that is appropriate for the datasets & benchmarks track.
* The inclusion of relative model ranking and feature importance for each classification model provides an insightful additional point of comparison beyond just accuracy, and allows for deeper understanding of the strengths and weaknesses of the generated data.
* The experimental manipulations around both label noise and different pre/post-processing steps are important for understanding data-level interventions that affect model performance.

**Additional Feedback:**

A very relevant question to this work would be to ask what _mixture_ of real and synthetic data is optimal, rather than just which model produces the most valid synthetic data. It’s possible that mixtures of real and synthetic data will perform better than even just real data alone.

**Clarity:**

**Main ideas (content-related)**: The main ideas and rationale for most choices made is clearly conveyed.

**Writing style (not content-related)**: The writing style often lacks clarity. In particular, there are a lot of sentences that seem to refer back to some previously mentioned idea or entity, but it’s not clear to me which idea/entity is being referenced. In terms of organization, many crucial details weren’t where I expected them to be. For example, the use of classification models in evaluation was discussed three or four separate times, with some overlap in content, but only in the later ‘experiments’ section did the authors state _which_ classification models they’re using.

**Correctness:**

As noted in the “Opportunities For Improvement” section, it is hard to assess the correctness of the claims given that the findings are not very robust and the authors have not used significance testing.

**Documentation:**

The appendix and supplementary materials provide appropriate documentation.

**Ethics:**

No ethical concerns.

**Limitations:**

The authors provide limitations in the appendix. Discussion is minimal and only focuses on the idea of expanding future work to include privacy preservation.

**Opportunities For Improvement:**

**Major**
* Most results do not appear to be robust, but this is not addressed. To be clear, it’s not an issue that results are inconclusive, but the text presents the results from table 2 as indicating clearly that “data-centric methods improve the utility of synthetic data” (line 288), but in every single comparison, the 95% confidence interval includes the value 0 (which indicates no difference from baseline, since the authors report this as a difference score). The same is the case for nearly all of the other metrics (model selection, feature selection, statistical fidelity), often with incredibly large ranges on the confidence intervals – bayesian networks, in particular, seem unstable with respect to feature selection across model runs. The source of this variability should be explored and better understood in order to interpret the results.
* Given the variability shown in Table 2, the authors should have added these confidence intervals to Figures 3 and 4. Though this can be somewhat reconstructed for Figure 3, it’s impossible to assess the reliability of the results across different levels of label noise (figure 4) without this information. I don’t doubt the presence of an overall trend, but the authors should either use a statistical test to support their claims or at least show the way that levels of label noise affect the variance in their results.
* It would be helpful to also apply the post-processing strategies to real data for comparison. If this step affects results with synthetic data, it should also have similar effects if the post-processing is applied to real data. Same for the label noise experiments, it would be helpful to see results against real data as a point of comparison.

**Minor**
* The takeaway from figure 1 seems like it’s intended to be a proof of concept for the paper, but it was unclear what the takeaway was. The paragraph from lines 36-46 seemed to assume the reader already knew a lot of information that was never provided, like what the different models are, what size of a difference in the metrics represented on the y-axes are meaningful (or even whether higher values should be preferred, compared to real data), and relevant details of the dataset (the authors just give a name, but no details of what kind of dataset or even a reference). The authors then state that these findings correlate with performance on downstream tasks, but I couldn’t find in the results where they directly compare these data profile metrics with classification accuracy (which is what I thought was being reported here).
* The argument for using tabular data could be improved. I agree with the authors that this is an important area to focus on, and that there are broad implications to improving data here, but that’s also true of other areas of generative data. What are the data requirements specific to tabular data that makes it appropriate for this research question? How are the metrics that are being used in this paper reflecting those requirements? The authors state that they expect their approach to generalize, but I have a hard time following how this kind of structured data generation would generalize to language generation tasks or image generation, for example.
* The authors are only using a subset of a larger benchmark. Though they describe and motivate their process for selecting from within just the ‘numerical features’ part of the benchmark, it’s unclear why they left out any datasets with categorical features.
* Just as a general comment, it would help to provide an example of the type of data being generated by the statistical models, along with an example of real data. It’s hard to get a picture of what the resulting dataset looks like without a concrete example.

**Relation To Prior Work:**

The authors sufficiently discuss relevant previous work on generating synthetic tabular data and evaluating it.

**Missing references**
* Line 39: Missing reference for data profiles that were “recently introduced in the data-centric AI literature”. Since “data profiles” is central to this paper, I’m very surprised that I can’t figure out what specific “data profiles” work they’re referencing and building off of (the paragraph that starts in line 104 makes me think it’s the authors’ term, but earlier discussion made it sound like an established term I’m just unfamiliar with – this can be clarified much sooner in the paper, I think).
* Line 45: Missing reference for the Adult dataset. (I’d also recommend giving a quick 1-sentence explanation of the dataset/task).

**Summary And Contributions:**

This paper applies data-processing steps to synthetic tabular data generated by a range of different models to investigate the effects of various design choices in creating a synthetic dataset. They find that though synthetic data is inferior to real data, design choices like which generative model is being used and which pre and post processing steps are applied can impact the utility of the data that is generated.

---

> ### Author Response · Authors · 2023-08-12
>
> Thank you for your thoughtful and constructive comments and suggestions! Please find our answers to each of your points below:
>
> * (A) Variability and significance
> * (B) Post-processing real data
> * (C) Arguments for tabular data
> * (D) Example data
> * (E) Data subset
> * (F) Figure 1
> * (G) Additional points
>
>
> ## (A) Variability and significance
> We agree that providing more information on the variability and robustness of the results is beneficial. Appendix C has now been expanded with a new section on the distribution of scores (C.9), and we have added a new section to the Appendix (D) that investigates the statistical significance of the results. Below, we report the main findings from the two new sections.
>
>
> ### (A.1) Main benchmark datasets
>
> #### (A.1.1) Variability
> One reason for the large confidence intervals reported Table 2 is the fact that we are averaging across 11 different datasets. To decompose the variability, we have 1) plotted the distribution of scores across all the datasets and generative models in Appendix C.9.1, and 2) modelled the effect of the pre/postprocessing strategies using linear models that control for the effect of the datasets and generative models (see A.1.2 in this response and Appendix D).
>
> The main takeaways from Appendix C.9 are that the difficulty of the datasets is highly variable and that the variability of model performance regardless of pre/postprocessing strategy seems to be related to model performance. Notably, the models demonstrating strong performance, such as TabDDPM, exhibit comparatively lower variability across different seeds. In contrast, models with poorer performance, like Bayesian networks and normalizing flows, display higher variability across seeds.
>
> As you have pointed out, the performance metrics exhibit higher variance for the tasks of feature selection and model ranking compared to classification. This discrepancy can likely be attributed to the inherent smoothness of the performance metrics. The model ranking score is established by accurately ranking the eight supervised classification models listed in Appendix B.3. Similarly, the feature selection score is determined by correctly ranking the metrics for feature importance across all features within the dataset – a quantity that naturally varies by dataset.
>
> In contrast, classification performance relies on AUROC values computed on the test datasets, which encompasses a relatively large number of samples (1,141 in the smallest dataset). Due to the dissimilarity in the number of samples, models, and features, the metrics for model and feature ranking are inevitably more susceptible to larger variance. This occurs because disparities in a single element can lead to more substantial effects on the performance metrics in comparison to the impact on classification.
>
> Regarding the high variance of Bayesian networks, this is due to the properties of the model. Bayesian networks are simply a graphical representation of conditional independencies, which is challenging with continuous variables. It is common to assume that the random variables in a Bayesian network are discrete since many Bayesian network learning algorithms are unable to efficiently handle continuous variables. While assumptions make it possible this might lead to suboptimal results compared to other methods. Indeed, as shown in Figures 14-16 in the Appendix, Bayesian networks fail to fit three of the datasets.

---

> > ### Author Response · Authors · 2023-08-12
> >
> > #### (A.1.2) Significance
> >
> > To statistically assess the effect of the pre- and postprocessing strategies, we constructed three linear models (one for each task: classification, feature selection, model selection). The models were designed to control for both dataset and generative model effects while accounting for potential interaction effects between datasets and generative models, as the different datasets might have characteristics that make them more or less difficult for specific generative models. The models had the following formula:
> >
> > metric ~ dataset * generative_model + preprocessing_strategy + postprocessing_strategy
> >
> > Where “metric” represents the performance metric corresponding to the task, i.e., AUROC for classification or Spearman's Rank Correlation for feature and model selection.
> >
> > Appendix D shows the full output of the linear models along with a summary and interpretation. As expected, the variability between datasets, and between interactions between datasets and generative models, is quite large. However, of most interest, is the effect of the pre/processing strategies which is shown in Table 15 in the Appendix. Notably, the only two significant effects of the processing strategies are postprocessing for the classification task, and preprocessing for the model selection task, although their estimates are rather small.
> >
> >
> > When contrasted with the results from the label noise experiments (see A.2.2 in the response below or Appendix D), this might suggest that the datasets are too “clean” to reap benefits from the data-centric processing. As mentioned in the paper, the data used in the main experiment come from the “Tabular benchmark numerical classification” benchmark suite from [1]. To facilitate ease of modelling, the benchmark datasets have been processed heavily, with e.g. removal of categorical features and missing values, and balancing of the classes. This preprocessing could be suspected to remove a large proportion of the noise in the datasets and thereby mitigate the effects of further processing with data-centric methods.
> >
> > [1] Grinsztajn, L., Oyallon, E., & Varoquaux, G. (2022). Why do tree-based models still outperform deep learning on typical tabular data?. Advances in Neural Information Processing Systems, 35, 507-520.
> >
> > ### (A.2) Label noise
> >
> > #### (A.2.1) Variability
> > To investigate the variability in the label noise experiments, we have added two additional tables in Appendix C.9.2 for the label noise experiment and created the same visualizations of the distribution of scores in Appendix C.9.1 for the label noise experiment in Appendix C.9.2. The tables show 1) the same as Table 1, but aggregated across levels of label noise instead of datasets, 2) performance by level of label noise summarized over all generative models. The table that does not summarize across either generative model or dataset id takes up 126 rows, and is therefore not included. Instead, we refer to section A.2.2 in this response or Appendix D for an analysing of the interaction between levels of label noise and the effect of data-centric processing.
> >
> > #### (A.2.2) Significance
> > To statistically assess the effect of the pre- and postprocessing strategies with different amounts of label noise, we constructed three linear models (one for each task: classification, feature selection, model selection). The models were designed to control for label noise and generative model effects, and had the following formula:
> >
> > metric ~ prop_label_noise + generative_model + preprocessing_strategy + postprocessing_strategy
> >
> > Where “metric” represents the performance metric corresponding to the task, i.e., AUROC for classification or Spearman's Rank Correlation for feature and model selection.
> >
> > The main effect of pre- and postprocessing is shown in Appendix Table 16. The effect of the processing strategies is significant in all cases except postprocessing for feature selection.
> >
> > To assess whether the effectiveness of the pre- and postprocessing strategies is modulated by the level of label noise in the data, we constructed a linear model equivalent to the one described above, but with an interaction effect between the proportion of label noise and the pre- and postprocessing strategies. Akaike’s Information Criteria indicated that the simple model (without interactions) was a better fit, and only a few of the interaction effects were significant, although there was a slight tendency towards an inverted V shape effect of the processing strategies, i.e. larger benefit with moderate levels of label noise. The output of these models is reported in Tables 23-25.

---

> > > ### Author Response · Authors · 2023-08-12
> > >
> > > ## (B) Post-processing real data
> > >
> > > We agree that reporting results for postprocessing of the real data provides valuable insights for the performance metrics and as a comparison for the generative models. We have added a subsection to Appendix C (C.11) that investigates this.
> > >
> > > Post-processing the real data reduces to applying a data-centric method (e.g. cleanlab) to the dataset before training supervised classification models on the data. Table 14 has been added to Appendix C.11, and shows the unnormalized performance of all the generative models (averaged over all datasets), with postprocessing of the original data included. Additionally, Table 15 shows the same, but for the data with added label noise.
> > >
> > >
> > > ## (C) Arguments for tabular data
> > > The decision to concentrate on tabular data generation was driven by several factors that we believe make our work both valuable and applicable to a broader context of machine learning research:
> > >
> > > 1. Data Diversity: Tabular data is ubiquitous across a wide range of domains, including finance, healthcare, e-commerce, and more. By focusing on tabular data we are able to evaluate the data-centric methods across many different domains with highly varied applications.
> > > 2. Scalability: By focusing on tabular data, we are able to provide a more comprehensive benchmark in terms of the number of seeds and datasets due to the additional computational resources required to train and apply models for image or text generation/classification.
> > > 3. Likely generalization: The methods for data-centric processing we have used in this paper are all directly applicable to other modalities, due to their only requirement being a trained classification model. As the mechanism of the data-centric methods is to identify wrongly labelled samples, we would expect the removal of these to have a similar effect regardless of the modality of the data. Additionally, the metrics we are using are not specifically related to tabular data but are applicable across all domains.
> > >
> > > ## (D) Example data
> > >
> > > Examples of the dataset can be found online via the links in Appendix Table 4. We agree that it’s useful to inspect and compare real and synthetic data, however, this is substantially harder to do without reducing to a metric of e.g. statistical fidelity for tabular data than for image/text data. For instance, it’s easy to inspect whether a generated image of a cat actually looks like a cat, compared to identifying whether the combinations of values in the columns of a dataset of e.g. eye-movements (dataset-id 361070) resemble reality. For this reason, we provide a measure of statistical fidelity as a proxy for how well the generated data matches the real data.
> > >
> > > ## (E) Data subset
> > > The reason for not including the benchmark with categorical features was initially due to wanting to include DAG learning (causal structure) as part of the benchmark (e.g. DAGMA and NO-TEARS) which do not work for categorical data. However, they were found to be extremely unstable, even for different splits of the real data, and were therefore excluded. As such, there is no theoretical limitation on the generative models that preclude using categorical features, but rather a limitation imposed by the initial plans for the project and subsequent time- and resource constraints.
> > >
> > > ## (F) Figure 1
> > > We apologize for the lack of clarity in Figure 1. We have extended the figure legend and re-ordered the panels to hopefully be more clear.
> > >
> > > ## (G) Additional points
> > > Thank you for pointing out the missed references. We have added a reference to the adult dataset at all mentions and made it more clear that the term “data profiles” is our own. We agree entirely that investigating which mixture of real and synthetic data is optimal is a worthy pursuit for future research - especially in cases of data scarcity or highly unbalanced data.

---

> > ### Comment · Reviewer_yA6T · 2023-08-28
> > **Appreciate the thorough reply**
> >
> > I appreciate the detailed reply and additional analyses provided by the authors. I've read through the updates and much of what was added to the appendix. While I still feel that the results are not especially robust, the authors have done an effective job of providing enough additional analyses to better understand the results and variance in their experiments. These changes have improved the paper, and so I will raise my score to reflect that.

---

> > > ### Author Response · Authors · 2023-08-28
> > >
> > > Dear Reviewer yA6T
> > >
> > > We are glad our additional analysis and updates have helped and would like to thank you for raising your score!
> > >
> > > Thanks again for your time and suggestions!
> > >
> > > Regards
> > >
> > > Paper 351 Authors

---

### Author Response · Authors · 2023-08-12

We wish to thank all the reviewers for the excellent feedback they have provided. We have now submitted revised versions of the main manuscript and the Appendix, which we believe have been substantially improved thanks to the comments from the reviewers.

The major changes include:

**Main manuscript**

* Limitations has been moved to the main manuscript and expanded with a section on other modalities (text, images).
* Figure 1 has been made more clear.
* Comments on the significance of the findings have been added to the Results section.

**Appendix**

Multiple new sections have been added to the Appendix:
* C.9: Distributions of scores. This section contains an investigation of the distribution of the performance scores across both the main benchmark data and the data with added label noise.
* C.10: Other Measures of Statistical Fidelity. This section contains a table with additional measures of statistical fidelity (Wasserstein distance and Maximum Mean Discrepancy).
* C.11: Postprocessing the real data: This section shows the effect of postprocessing (i.e. removing hard examples) from the original data and its effect on the performance of the three tasks.
* Appendix D: Statistical tests. This new part of the Appendix adds detailed information on statistical tests of the significance and magnitude of the effects of the data-centric processing on the main benchmark data and the data with added label noise.

The individual points raised by the reviewers are attended to in the comments of each review.

---

### Author Response · Authors · 2023-08-24

Dear reviewers,

We once again wish to thank you for your feedback and insightful suggestions.

We are glad the reviewers deemed the “motivation” (R-g5vc,R-yA6T) of the paper to address a “crucial and practical challenge” (R-yA6T) of “broad importance” (R-g5vc) and that our benchmarking findings “provides fresh insights” (R-P9Bf). Further, the “experimental manipulations around both label noise and different pre/post-processing steps are important for understanding data-level interventions that affect model performance” (R-P9Bf) and that our benchmarking framework  “will potentially benefit the community of synthetic data generation” (R-yA6T).

To further improve understanding of our work, we have addressed the following additional aspects:

**Main manuscript updates**:
- For each benchmarking question we have added practical guidance based on our benchmarking findings. This is to provide actionable takeaways to practitioners on how to better generate and use synthetic tabular data on the basis of our benchmarking findings. We hope that this helps to provide additional insights into our results from a practical perspective.
- Expanded the Limitations and Future work section.
- Changed the title to “Reimagining Synthetic **Tabular** Data Generation through Data-Centric AI: A Comprehensive Benchmark”.

We also wish to reiterate that we focus on the benchmarking of synthetic data generation and the role of data-centric AI, particularly for the tabular data domain, given its importance across a wide variety of industries. In this spirit, we have looked at 11 tabular datasets with diverse characteristics (higher than the median average of 4 datasets on the D&B track [R1]).

Of course, there is applicability of our framework to other modalities like text and images, however, we wish to articulate the distinct characteristics of tabular data which permit us to go beyond just classification performance and/or model selection. The tabular domain also has the important aspect of defined features which differs from the images and text. Hence, feature selection in tabular data may not directly apply to images or text. This highlights that while the “idea” and “framework” introduced in our paper can be applied to other modalities, ideally, we desire custom benchmarks for each data type. This is a potential avenue for future research which we mention in Section 5.1.

We hope that this helps to further address the reviewers' concerns. In the limited time remaining, we are still eager to do our utmost to address any concerns!

With gratitude

The Authors of paper 351

[R1] I.Guyon. “The Data-Centric Era: How ML is Becoming an Experimental Science”, NeurIPS 2022 (Invited Keynote)

---

### Decision · Program_Chairs · 2023-09-22

**Decision:**

Accept (Poster)

**Comment:**

The paper addresses the important problem of how data-centric AI methods can be used to carefully build synthetic data and evaluates the performance of  five state-of-the-art models for tabular data generation on eleven distinct tabular datasets.  Practical recommendations for integrating data-centric insights are offered for the synthetic data generation process, with a specific focus on classification performance, model selection, and feature selection. The authors find that though synthetic data is inferior to real data, careful design choices of the generative model and pre and post processing steps can impact the utility of the data for the learning task.

Pros: Addresses an important issue, Good illustration on how relative model ranking and feature importance for each classification model provides insights beyond accuracy and sheds light on generated data strengths/weaknesses, data level interventions such as label noise and pre/post processing steps allow for more detailed performance characterization.
Cons: Most results do not appear to be robust. The paper was oversold in the first version as a generic approach that can be applied for all data modalities. This was then changed to only tabular data. Thus the scope of applicability of the work is limited.  Besides this, the type of insights offered in this paper is well known to statisticians who practice careful stratification of data samples during model estimation.

After discussion between the area chair, reviewers, the rating of the reviewer g5vc was increased to 5 (even though it is not reflected in the final review yet). Thus the overall average rating is: 19/3 = 6.33.    The paper is a borderline accept and may be considered for a poster.